# Dynamic Programming for Epistemic Uncertainty in Markov Decision Processes

**Axel Benyamine** [1 2]   **Julien Grand-Clément** [3]   **Marek Petrik** [4]   **Michael I. Jordan** [2 5]   **Alain Durmus** [1]

## Abstract

In this paper, we propose a general theory of ambiguity-averse MDPs, which treats the uncertain transition probabilities as random variables and evaluates a policy via a risk measure applied to its random return. This ambiguity-averse MDP framework unifies several models of MDPs with epistemic uncertainty for specific choices of risk measures. We extend the concepts of value functions and Bellman operators to our setting. Based on these objects, we establish the consequences of dynamic programming principles in this framework (existence of stationary policies, value and policy iteration algorithms), and we completely characterize law-invariant risk measures compatible with dynamic programming. Our work draws connections among several variants of MDP models and fully delineates what is possible under the dynamic programming paradigm and which risk measures require leaving it.

## 1. Introduction

Markov Decision Processes (MDPs) are a model of sequential decision-making in which an agent repeatedly interacts with a system to optimize rewards over a (possibly infinite) horizon (Puterman, 2014). It is well-documented that the performance of a policy, which dictates the choice of actions given the current state of the system, may severely deteriorate when the model parameters (such as the transition probabilities across states) are incorrect (Delage & Mannor, 2010). This *epistemic* uncertainty may occur in applications with high stakes, e.g., healthcare (Goh et al., 2018; Steimle et al., 2021) and vehicle routing (Miao et al., 2017).

The issue of model errors has been addressed in several ways in the literature. Robust MDPs (RMDPs) posit that the true parameters belong to an uncertainty set and optimize for the worst-case parameter realization (Iyengar, 2005; Nilim & El Ghaoui, 2005; Wiesemann et al., 2013); the uncertainty set represents all plausible parameter realizations and can be estimated from a dataset. RMDPs are among the most well-studied MDP models with uncertain kernels, and under appropriate rectangularity assumptions, they can be solved efficiently via dynamic programming (Wiesemann et al., 2013; Grand-Clément et al., 2024). Several other frameworks for MDPs with uncertain parameters exist, e.g., multi-model MDPs (Steimle et al., 2021; Su & Petrik, 2023) that optimize over the average of the return across plausible parameters, models based on value at risk (percentile optimization) (Delage & Mannor, 2010; Petrik & Russel, 2019; Behzadian et al., 2021), conditional value at risk (CVaR) (Lobo et al., 2020; Lin et al., 2022) and the entropic risk measure (ERM) (Russel et al., 2020).

Overall, several "uncertain MDP" models have been introduced independently, each with its own ad hoc analysis. For some models, computing an optimal policy is tractable via dynamic programming (RMDPs), for others, computing an optimal policy is NP-hard (e.g., for multi-model MDPs and percentile optimization), and for others, the answers to these questions are unknown (e.g., models based on CVaR and ERM). One may wonder whether there is a common perspective unifying these models, and, crucially from an application standpoint, whether one could design new models with favorable tractable properties (beyond RMDPs).

In this work, we propose a unifying model for MDPs with uncertain parameters. Our **contributions** are as follows.

- We propose *ambiguity-averse MDPs*, a principled approach to epistemic uncertainty in MDPs, where the uncertain transition probabilities are modeled as random variables, and the return is estimated using a risk measure capturing the agent's risk preferences. This model unifies most of the literature on MDPs with parameter uncertainty. We then introduce the generalization of the dynamic programming principle for ambiguity-averse MDPs, which states that ambiguity-averse value functions are fixed points of some Bellman operator.

- For a broad class of risk measures satisfying a set of

[1]CMAP, CNRS, École polytechnique, Institut Polytechnique de Paris [2]Inria Paris, Ecole Normale Supérieure, PSL Research University [3]ISOM Department, HEC Paris [4]University of New Hampshire [5]Univeristy of California, Berkeley. Correspondence to: Axel Benyamine <firstname.lastname@polytechnique.edu>.

*Proceedings of the 43$^{rd}$ International Conference on Machine Learning*, Seoul, South Korea. PMLR 306, 2026. Copyright 2026 by the author(s).

axioms, we prove that the fundamental properties of the Bellman operators (monotonicity and contraction) are preserved. We then establish that dynamic programming implies key structural and computational guarantees, most notably, the existence of stationary optimal policies and the validity of value and policy iteration algorithms.

- Finally, we fully characterize the risk measures for which ambiguity-averse MDPs satisfy dynamic programming. Surprisingly, we show that, under a law-invariance and monotonicity assumption, or under a continuity assumption, dynamic programming forces the objective to collapse to only robust, optimistic, or risk-neutral MDPs (we refer to Theorem 4.4 and Theorem 4.5 for formal statements).

Overall, our work provides a unified perspective on parameter uncertainty in MDPs within the dynamic programming framework. On the one hand, dynamic programming enables several desirable tractability properties, however it also severely limits the choice of risk models (among law-invariant risk measures). Our work emphasizes that designing further tractable ambiguity-averse MDP instances requires either going beyond dynamic programming and allowing, for instance, state-augmentation (Chow et al., 2015), or solutions based on direct gradient descent of the returns (Wang & Zou, 2022), or sticking to dynamic programming but going beyond law-invariance (e.g., focusing on nested risk measures (Ruszczyński, 2010)).

In the rest of the paper, we first provide background on MDPs and risk measures, then introduce the framework of ambiguity-averse MDPs, and finally state our main results on the algorithmic implications of dynamic programming and characterize risk measures compatible with dynamic programming. All the proofs are detailed in the appendices and we provide a literature review in Appendix C.

**Notations** Random variables are denoted by a capital letter and a tilde, e.g., $\tilde{X}$ or $\tilde{S}$. We denote by $L_c^\infty(B)$ the set of bounded random variables with convex support (see for instance Definition A.1) that are almost surely in $B$. The simplex over a set $\mathcal{S}$ is denoted by $\Delta(\mathcal{S})$. For two vectors $v, w \in \mathbb{R}^\mathcal{S}$, the inequality $v \leqslant w$ is component-wise: $v(s) \leqslant w(s), \forall\, s \in \mathcal{S}$.

## 2. Preliminaries

We first introduce (nominal) MDPs and risk measures.

### 2.1. Nominal MDPs

A (discounted) MDP is a tuple $\mathsf{M}_P = (\mathcal{S}, \mathcal{A}, r, P, \gamma, \mu)$, where $\mathcal{S}, \mathcal{A}$ are finite state and action spaces, $r \in \mathbb{R}^{\mathcal{S} \times \mathcal{A} \times \mathcal{S}}$

represents the instantaneous rewards; i.e., $r(s, a, s') \in \mathbb{R}$ is the reward when taking action $a$ in state $s$ and moving to state $s'$, and $P$ is a (fixed) transition probability kernel in $\Delta(\mathcal{S})^{\mathcal{S} \times \mathcal{A}}$. The vector $\mu \in \Delta(\mathcal{S})$ is the initial probability distribution, and the scalar $\gamma \in [0, 1)$ is a discount factor. A history-dependent policy maps any finite history $(s_0, a_0, s_1, ..., s_t)$ up to period $t \in \mathbb{N}$ to a distribution over actions in $\Delta(\mathcal{A})$. A stationary policy can be identified with a map $\mathcal{S} \to \Delta(\mathcal{A})$. The set of history-dependent (resp. stationary) policies is denoted by $\Pi_\mathsf{H}$ (resp. $\Pi_\mathsf{S}$). The goal of the decision-maker is to find an optimal policy that maximizes the expected discounted return; i.e., to solve

$$\sup_{\pi \in \Pi_\mathsf{H}} \mathbb{E}_\mu^{\pi, P} \left( \sum_{t=0}^\infty \gamma^t r(\tilde{S}_t, \tilde{A}_t, \tilde{S}_{t+1}) \right) , \qquad (1)$$

where $(\tilde{S}_t, \tilde{A}_t)$ is the (random) state-action pair visited at time $t$ and $\mathbb{E}_\mu^{\pi, P}$ is the expectation with the measure induced by $\pi, P, \mu$ over the trajectories $(\tilde{S}_t, \tilde{A}_t)_{t \in \mathbb{N}}$. Note that the discounted return is equal to $\mu^\top V^{\pi, P}$ with $V^{\pi, P} \in \mathbb{R}^\mathcal{S}$ the *value function* which characterizes the return obtained starting from each state and is defined as follows:

$$V^{\pi, P}(s) = \mathbb{E}_s^{\pi, P} \left( \sum_{t=0}^\infty \gamma^t r(\tilde{S}_t, \tilde{A}_t, \tilde{S}_{t+1}) \right) , \qquad (2)$$

for $\pi \in \Pi_\mathsf{H}$ and $s \in \mathcal{S}$. Given some transitions $P$ and a stationary policy $\pi \in \Pi_\mathsf{S}$, the associated Bellman operator is denoted as $T^{\pi, P} : \mathbb{R}^\mathcal{S} \to \mathbb{R}^\mathcal{S}$ with, for $v \in \mathbb{R}^\mathcal{S}, s \in \mathcal{S}$,

$$T^{\pi, P} v(s) \qquad\qquad\qquad\qquad\qquad (3)$$
$$= \sum_{a \in \mathcal{A}} \pi(s, a) \sum_{s' \in \mathcal{S}} P(s, a, s') \left( r(s, a, s') + \gamma v(s') \right) .$$

$T^{\pi, P}$ is a contraction for the $\ell_\infty$-norm, and its unique fixed-point is the value function $V^{\pi, P}$ (Puterman, 2014):

$$V^{\pi, P} = T^{\pi, P} V^{\pi, P} . \qquad\qquad (4)$$

Moreover, the optimal value function $V^P \in \mathbb{R}^\mathcal{S}$ defined as $V^P(s) = \sup_{\pi \in \Pi_\mathsf{H}} V^{\pi, P}(s)$ for $s \in \mathcal{S}$, satisfies the following Bellman optimality equations:

$$V^P(s) = \max_{\pi \in \Pi_\mathsf{S}} T^{\pi, P} V^P(s), \forall\, s \in \mathcal{S} . \qquad (5)$$

A stationary optimal policy can be computed by choosing a policy that attains the $\max$ in the right-hand side of (5). The fact that value functions and optimal value functions satisfy fixed-point equations as in (4) and (5) is usually referred to as the *dynamic programming principle* (Wang et al., 2023; Grand-Clément et al., 2024) and plays a central role in the favorable tractability properties of nominal MDPs. In particular, computing an optimal policy can be done efficiently by value iteration, policy iteration, linear programming, and

gradient descent (see, e.g., Puterman, 2014). One of the main objectives of our work is to understand, in the case where the transition probabilities are uncertain, when a similar dynamic programming principle holds. To this end, we rely on risk measures that we now introduce.

## 2.2. Risk measures

A risk measure is a map from random variables to $\mathbb{R}$. Examples include the expectation: $\rho(\tilde{X}) = \mathbb{E}(\tilde{X})$ and the Value-at-Risk ($\mathsf{VaR}_\alpha$) defined as the upper $(1-\alpha)$-quantile of $\tilde{X}$: $\mathsf{VaR}_\alpha(\tilde{X}) = \inf\{x \in \mathbb{R} \mid \mathbb{P}(\tilde{X} \leqslant x) > 1 - \alpha\}$ for some $\alpha \in (0,1)$, as well as the *essential infimum* (ess inf) and the *essential supremum* (ess sup), defined as

$$\operatorname{ess\,inf}(\tilde{X}) = \sup\{b \in \mathbb{R} \mid \mathbb{P}(\tilde{X} < b) = 0\}$$
$$\operatorname{ess\,sup}(\tilde{X}) = \inf\{b \in \mathbb{R} \mid \mathbb{P}(b < \tilde{X}) = 0\}$$

which coincide with the infimum and supremum of the support of $\tilde{X}$ (when $\tilde{X}$ is bounded, see Lemma A.2). Recent work on risk-averse MDPs has also focused on the Conditional Value-at-Risk ($\mathsf{CVaR}$) (Chow et al., 2015; Godbout & Durand, 2025), the Entropic Risk Measure ($\mathsf{ERM}$) and the Entropic Value-at-Risk ($\mathsf{EVaR}$) (Hau et al., 2023b). The terminology "Conditional Value-at-Risk" should not be confused with the conditional risk mappings used in time-consistent dynamic risk models, e.g. (Ruszczyński, 2010). In this paper, $\mathsf{CVaR}$ is a static risk measure (also called Average Value-at-Risk, Tail Value-at-Risk, or superquantile) applied to a single random variable. Classical analysis of risk measures focuses on a small number of important properties. We introduce here the ones that play a role in our framework:

- A risk measure $\rho$ is **monotone** if $\rho(\tilde{X}) \geqslant \rho(\tilde{Y})$ for any random variables $\tilde{X}, \tilde{Y}$ such that $\tilde{X} \geqslant \tilde{Y}$ a.s.

- A risk measure $\rho$ is **translation-invariant** if $\rho(\tilde{X} + c) = \rho(\tilde{X}) + c$, for any $c \in \mathbb{R}$ and random variable $\tilde{X}$.

- A risk measure $\rho$ is **law-invariant** if it only depends on the law of the random variables, i.e. if $\rho(\tilde{X}) = \rho(\tilde{Y})$ for any random variables $\tilde{X}, \tilde{Y}$ that have same law. We provide an example and a discussion on risk measures that are not law-invariant in Example A.3.

Most standard risk measures, including $\mathbb{E}$, ess inf, ess sup, $\mathsf{VaR}$, $\mathsf{CVaR}$, $\mathsf{ERM}$ and $\mathsf{EVaR}$, satisfy these properties. Non-law-invariant risk measures exhibit less interpretable behaviours, see Example A.3 for an example. We refer to Chapter 4 in Föllmer & Schied (2016) for an introduction to risk measures and to Appendix A for further details.

## 3. Ambiguity-averse MDPs

### 3.1. Definition, objectives and examples

In this section, we introduce the framework of *ambiguity-averse MDPs* to model parameter uncertainty in MDPs. The main idea is to consider the transition probabilities as a random variable $\tilde{P}$ with distribution $\nu$, so that the return $\mu^\top V^{\pi, \tilde{P}}$ is itself a random variable and the decision-maker optimizes for $\pi \mapsto \rho(\mu^\top V^{\pi, \tilde{P}})$, where $\rho$ is a risk measure. We focus on uncertain transition probabilities and known rewards (this is the most difficult case, already for RMDPs (Grand-Clément et al., 2024)).

**Definition 3.1.** A (discounted) *ambiguity-averse MDP* is a tuple $\mathsf{M}_\nu = (\mathcal{S}, \mathcal{A}, r, \nu, \gamma, \mu)$ where $\mathcal{S}, \mathcal{A}, r, \gamma, \mu$ are defined exactly as in nominal MDPs (see Section 2) and $\nu$ is a probability distribution over the set $\Delta(\mathcal{S})^{\mathcal{S} \times \mathcal{A}}$ of feasible transition probabilities.

The distribution $\nu$ models the likelihood of each realization of the transition kernels in $\Delta(\mathcal{S})^{\mathcal{S} \times \mathcal{A}}$. Uncertainty in transition probabilities is frequent in high-stakes applications where model parametrization is difficult, e.g., in healthcare (Goh et al., 2018; Steimle et al., 2021). Since $\Delta(\mathcal{S})^{\mathcal{S} \times \mathcal{A}}$ is bounded for fixed $(\mathcal{S}, \mathcal{A})$, any measure $\nu$ in an ambiguity-averse MDP necessarily has compact support. The distribution $\nu$ naturally induces a distribution for the sequence $(\tilde{P}_t)_{t \in \mathbb{N}}$ of random kernels for each decision period. To fully specify the dynamics induced by an ambiguity-averse MDP instance, we consider two types of uncertainty:

**Definition 3.2.** Let $\mathsf{M}_\nu$ be an ambiguity-averse MDP. The (random) transition kernels $(\tilde{P}_t)_{t \in \mathbb{N}}$ are **static** if $\tilde{P}_t = \tilde{P}_0$ for any $t \in \mathbb{N}$, with $\tilde{P}_0 \sim \nu$. The transition kernels are **resampled** if $(\tilde{P}_t)_{t \in \mathbb{N}}$ is i.i.d. with distribution $\nu$.

We refer to Appendix B for a rigorous formalization of these notions. In principle, one can extend the model of resampled kernels to independent variables, not necessarily identically distributed, i.e., to the case where $\tilde{P}_t \sim \nu_t$ for each $t \in \mathbb{N}$ and some distributions $(\nu_t)_{t \in \mathbb{N}}$. We consider only the i.i.d. setting here to simplify the exposition of our main results, and because this model reduces to i.i.d. kernels for distributionally robust MDPs; see Appendix B.1. We note that value functions $V^{\pi, \tilde{P}}$ naturally extend to the case where $\tilde{P}$ represents a sequence of kernels (we slightly overload notation here; see Appendix B). We note that considering static or resampled kernels leads to very different computational implications. For instance, for multi-model MDPs, it is known that static kernels lead to NP-hard problems (Steimle et al., 2021) while with resampled kernels, multi-model MDPs can be solved efficiently by dynamic programming.

We use the following notion of product structure.

**Definition 3.3.** A distribution $\nu$ on $\Delta(\mathcal{S})^{\mathcal{S} \times \mathcal{A}}$ is said to have a *product structure* if for any $\tilde{P} \sim \nu$, the family of random

variables $\{\tilde{P}(s,\cdot,\cdot) \in \Delta(\mathcal{S})^{\mathcal{A}} : s \in \mathcal{S}\}$ is independent.

We say that $\mathsf{M}_\nu$ has a product structure when $\nu$ does. In this case, the support of $\nu$, which is a subset of the set of kernels $\Delta(\mathcal{S})^{\mathcal{S} \times \mathcal{A}}$, has the classical *s-rectangularity property* widely studied for robust MDPs (Wiesemann et al., 2013) and necessary for the existence of Bellman equations in the context of robust MDPs (Grand-Clément et al., 2024), see Appendix C for details.

We will use the following simple example to illustrate the model of ambiguity-averse MDPs.

**Example 3.4.** Figure 1 provides a simple example of an ambiguity-averse MDP. There is only a single action and

*Figure 1.* Ambiguity-averse MDP with two states and one action. The edges $(s, a, s')$ are labeled with pairs $(P(s, a, s'), r(s, a, s'))$.

the state End is absorbing. The transitions from state Start are modeled with a random variable $\tilde{X}$ with a uniform distribution on $[0, 1]$. Here, $\nu$ is the distribution of the random matrix $\begin{pmatrix} \tilde{X} & 1 - \tilde{X} \\ 0 & 1 \end{pmatrix}$. In the case of resampled kernels, $\tilde{X}$ is resampled i.i.d. at each period $t \in \mathbb{N}$. In the case of static kernels, there is a single sample of $\tilde{X}$ at time $t = 0$, so that the kernel $\tilde{P}_t$ at time $t \in \mathbb{N}$ is $\tilde{P}_t = \begin{pmatrix} \tilde{X} & 1 - \tilde{X} \\ 0 & 1 \end{pmatrix}$.

Given an ambiguity-averse MDP $\mathsf{M}_\nu$, and a risk measure $\rho$, the objective of the decision maker is to solve:

$$\sup_{\pi \in \Pi_{\mathsf{H}}} \rho\left(\mathbb{E}_\mu^{\pi,\tilde{P}}\left(\sum_{t=0}^{\infty} \gamma^t r(\tilde{S}_t, \tilde{A}_t, \tilde{S}_{t+1})\right)\right), \qquad (6)$$

where the inner expectation $\mathbb{E}_\mu^{\pi,\tilde{P}}$ is over random trajectories $(\tilde{S}_t, \tilde{A}_t)_{t \in \mathbb{N}}$ and the risk measure $\rho$ is over the random kernel sequence $\tilde{P} = (\tilde{P}_t)_{t \in \mathbb{N}}$. Note that the objective function in (6) can be formulated as $\pi \mapsto \rho\left(\mu^\top V^{\pi,\tilde{P}}\right)$, with nominal value functions defined in (2). Similarly to the case of nominal MDPs, we define the **ambiguity-averse value function** $V^{\pi,\nu,\rho} \in \mathbb{R}^{\mathcal{S}}$ under policy $\pi \in \Pi_{\mathsf{H}}$ as:

$$V^{\pi,\nu,\rho}(s) = \rho\left(V^{\pi,\tilde{P}}(s)\right), \quad \forall s \in \mathcal{S},$$

where we recall that the nominal value function is defined in Equation (2). We recall that the randomness (for the risk measure $\rho$) is only in $\tilde{P}$. The **optimal ambiguity-averse value function** $V^{\nu,\rho} \in \mathbb{R}^{\mathcal{S}}$ is

$$V^{\nu,\rho}(s) = \sup_{\pi \in \Pi_{\mathsf{H}}} V^{\pi,\nu,\rho}(s), \quad \forall s \in \mathcal{S}.$$

**Examples and related work** We briefly overview some important examples of ambiguity-averse MDPs here; see Appendix C for a rigorous presentation.

- **Nominal MDPs** with (fixed) kernel $P$ correspond to static kernels with $\nu$ a Dirac distribution supported on $P$, and $\rho(\tilde{X}) = \mathbb{E}^\nu(\tilde{X})$, so that we recover the classical nominal MDP objective: $\rho(\mu^\top V^{\pi,\tilde{P}}) = \mu^\top V^{\pi,P}$.

- **Robust MDPs** with compact convex uncertainty set $\mathcal{P}$ correspond to static kernels, with $\nu$ supported on $\mathcal{P}$, and $\rho(\tilde{X}) = \operatorname{ess\,inf}(\tilde{X})$, so that we recover the classical RMDP objective (Wiesemann et al., 2013): $\rho(\mu^\top V^{\pi,\tilde{P}}) = \inf_{P \in \mathcal{P}} \mu^\top V^{\pi,P}$. The case of robust MDPs with Markovian kernels can be modeled with resampled kernels, see Appendix C. We also cover **optimistic MDPs** for $\rho(\tilde{X}) = \operatorname{ess\,sup}(\tilde{X})$, see Lemma A.2 for the connection between $\operatorname{ess\,inf}, \operatorname{ess\,sup}, \inf$, and $\sup$.

- **Multi-model MDPs** (MMDP) with a distribution $\nu$ over the plausible realization of the transition kernels correspond to $\rho(\tilde{X}) = \mathbb{E}^\nu\left(\tilde{X}\right)$. Most of the literature focus on the case of static kernels, for which we recover the classical MMDP objective (Steimle et al., 2021): $\rho(\mu^\top V^{\pi,\tilde{P}}) = \mathbb{E}^\nu(\mu^\top V^{\pi,\tilde{P}})$. Multiple-environment MDPs (MEMDPs) with *prior semantics* can be represented in the same way, while the *adversarial semantics* of MEMDPs are closer to static robust MDPs with a possibly non-convex set of environments (Raskin & Sankur, 2014; Chatterjee et al., 2020; Bordais & Raskin, 2026), see Appendix C for more detail.

- **Percentile Optimization** with distribution $\nu$ and risk level $\alpha \in (0, 1)$ corresponds to static kernels, with $\rho(\tilde{X}) = \mathsf{VaR}_\alpha(\tilde{X})$, so that we recover the popular percentile optimization objective (Delage & Mannor, 2010; Lobo et al., 2023): $\rho(\mu^\top V^{\pi,\tilde{P}}) = \mathsf{VaR}_\alpha(\mu^\top V^{\pi,\tilde{P}})$. Similarly, we can recover related frameworks that use risk measures other than $\mathsf{VaR}$, referred to as soft-robustness (Lobo et al., 2020) or Bayes risk-aversion (Lin et al., 2022), by choosing the appropriate risk measures.

We refer to Table 1 for a summary of these reformulations. Given this unification, a natural question that arises is whether we can compute an optimal ambiguity-averse policy in general; i.e., whether we can solve (6) efficiently. Given the central role of Bellman operators and Bellman equations in solving nominal and robust MDPs, we next extend these notions to ambiguity-averse MDPs.

**Remark 3.5. Distributionally robust MDPs** (DRMDPs) require an *ambiguity set* of plausible distributions and not just a single distribution $\nu$ as in Definition 3.1. To model DRMDPs as ambiguity-averse MDPs, one would need the formalism of *ambiguous probability space* (Delage et al.,

*Table 1.* Summary and properties of the main variants of MDP models studied in this paper. These models are reformulated as ambiguity-averse MDPs with a risk measure $\rho(\tilde{X})$. For nominal MDPs, we note "$\delta_P$" for a Dirac measure supported on the set $\{P\}$. For robust MDPs and optimistic MDPs, we assume a rectangular convex compact uncertainty set. The column "DP" indicates whether Condition 1 and Condition 2 hold. The column "$\pi^\star \in \Pi_S$" indicates that an optimal policy can be chosen stationary. The column "Trac." indicates whether the problem is tractable: for this column, "$\times$" refers to NP-hardness, "$\checkmark$" refers to problems that can be solved in polynomial time (when the discount factor is fixed), and "?" refers to unknown status.

| Name | Objective | Risk measure $\rho(\tilde{X})$ | Static Kernels | | | Resampled Kernels | | |
|---|---|---|---|---|---|---|---|---|
| | | | DP | $\pi^\star \in \Pi_S$ | Trac. | DP | $\pi^\star \in \Pi_S$ | Trac. |
| Nominal MDP | $\mu^\top V^{\pi,P}$ | $\mathbb{E}^\nu(\tilde{X})$ with $\nu = \delta_P$ | $\checkmark$ | $\checkmark$ | $\checkmark$ | $\checkmark$ | $\checkmark$ | $\checkmark$ |
| Robust MDP | $\inf_{P\in\mathcal{P}} \mu^\top V^{\pi,P}$ | $\operatorname{ess\,inf}(\tilde{X})$ | $\checkmark$ | $\checkmark$ | $\checkmark$ | $\checkmark$ | $\checkmark$ | $\checkmark$ |
| Optimistic MDP | $\sup_{P\in\mathcal{P}} \mu^\top V^{\pi,P}$ | $\operatorname{ess\,sup}(\tilde{X})$ | $\checkmark$ | $\checkmark$ | $\checkmark$ | $\checkmark$ | $\checkmark$ | $\checkmark$ |
| Multi-model MDP | $\mathbb{E}^\nu(\mu^\top V^{\pi,\tilde{P}})$ | $\mathbb{E}^\nu(\tilde{X})$ | $\times$ | ? | $\times$ | $\checkmark$ | $\checkmark$ | $\checkmark$ |
| Percentile optimization | $\mathsf{VaR}_\alpha(\mu^\top V^{\pi,\tilde{P}})$ | $\mathsf{VaR}_\alpha(\tilde{X})$ | $\times$ | $\times$ | $\times$ | $\times$ | $\times$ | $\times$ |

2019), requiring considerably more notation and formalization. This extension is left as an interesting next step.

**Remark 3.6.** We compare here the ambiguity-averse MDP framework to *risk-averse* MDPs (see, e.g., Ruszczyński, 2010; Hau et al., 2023a), which considers a single, known transition kernel $P$ and hedges against the variability of the return over the random trajectories (sometimes called *aleatoric risk*) with a risk measure by solving

$$\sup_{\pi \in \Pi_H} \rho \left( \sum_{t=0}^{+\infty} \gamma^t r(\tilde{S}_t, \tilde{A}_t, \tilde{S}_{t+1}) \right) .$$

In contrast, our model (6) considers the expected return over the trajectories, and hedges against the uncertainty in parameters (sometimes called *epistemic risk*) with a risk measure; see Appendix C.2 for more comparisons.

### 3.2. Dynamic programming

For a risk measure $\rho$, we define the **ambiguity-averse Bellman operator** $T^{\pi,\nu,\rho} : \mathbb{R}^S \to \mathbb{R}^S$ associated with a stationary policy $\pi \in \Pi_S$ and a random transition kernel following law $\nu$ with identical marginals over time (static or resampled) as, for $v \in \mathbb{R}^S$,

$$T^{\pi,\nu,\rho}v(s) = \rho\left(T^{\pi,\tilde{P}}v(s)\right), \quad \forall s \in \mathcal{S}, \qquad (7)$$

where $\tilde{P} \sim \nu$ on the right-hand side of (7) and $T^{\pi,\tilde{P}}$ is the standard Bellman operator for nominal MDPs as defined in (3). We then define the **optimal ambiguity-averse Bellman operator** $T^{\nu,\rho} : \mathbb{R}^S \to \mathbb{R}^S$ as, for $v \in \mathbb{R}^S$,

$$T^{\nu,\rho}v(s) = \sup_{\pi \in \Pi_S} T^{\pi,\nu,\rho}v(s), \quad \forall s \in \mathcal{S} . \qquad (8)$$

The operators $T^{\pi,\nu,\rho}, T^{\nu,\rho}$ play the same role as the Bellman operators $T^{\pi,P}$ and $T^P$ (introduced in Section 2) in the theory of nominal MDPs and inherit their properties under some conditions on the risk measure $\rho$, as we show next.

**Proposition 3.7.** *Let $\rho$ be a monotone, translation-invariant risk measure. Then we have the following properties for the operator $T \in \{T^{\pi,\nu,\rho}, T^{\nu,\rho}\}$:*

1. *Monotonicity: for any $v, w \in \mathbb{R}^S$ such that $v \leqslant w$ then, $Tv \leqslant Tw$.*

2. *Translation-invariance: for any $v \in \mathbb{R}^S$ and $c \in \mathbb{R}$, denoting $1_S = (1, ..., 1) \in \mathbb{R}^S$, we have $T(v + c \cdot 1_S) = Tv + c \cdot 1_S$.*

3. *Contraction: $T$ is a contraction for the $\ell_\infty$-norm.*

*Finally, we have the following* attainability property*: for any $v \in \mathbb{R}^S$, there exists $\pi \in \Pi_S$ such that $T^{\nu,\rho}v = T^{\pi,\nu,\rho}v$.*

We now introduce the **dynamic programming principles** for ambiguity-averse MDPs, which state that value functions are fixed points of Bellman operators. We write $\mathcal{M}_c$ for the class of all "convex, product" ambiguity-averse MDPs, i.e. the set of all $M_\nu = (\mathcal{S}, \mathcal{A}, r, \nu, \gamma, \mu)$ with any $\mathcal{S}, \mathcal{A}, r, \gamma, \mu$, and any $\nu$ with convex support and product structure.

**Condition 1.** (Bellman Equations for Policy Evaluation). We say that a risk measure $\rho$ satisfies this condition, if $V^{\pi,\nu,\rho}$ is a fixed point of $T^{\pi,\nu,\rho}$ for any instance $M_\nu \in \mathcal{M}_c$ and stationary policy $\pi \in \Pi_S$, i.e.,

$$V^{\pi,\nu,\rho} = T^{\pi,\nu,\rho}V^{\pi,\nu,\rho}, \quad \forall M_\nu \in \mathcal{M}_c, \forall \pi \in \Pi_S .$$

This condition can be satisfied either for all $M_\nu$ with *static* kernels, or for all $M_\nu$ with *resampled* kernels.

**Condition 2.** (Bellman Optimality Equations). We say that a risk measure $\rho$ satisfies this condition, if $V^{\nu,\rho}$ is a fixed point of $T^{\nu,\rho}$ for any instance $M_\nu \in \mathcal{M}_c$, i.e.,

$$V^{\nu,\rho} = T^{\nu,\rho}V^{\nu,\rho}, \quad \forall M_\nu \in \mathcal{M}_c .$$

This condition can be satisfied either for all $M_\nu$ with *static* kernels, or for all $M_\nu$ with *resampled* kernels.

These conditions are related to the dynamic programming principles introduced recently for robust MDPs (Wang et al.,

2023; Grand-Clément et al., 2024) and to dynamic consistency of risk measures (Kupper & Schachermayer, 2009; Pflug & Pichler, 2016). In particular, Conditions 1 and 2 are natural extensions of the Bellman equation for evaluating the value function of a policy, given in (4) and Bellman optimality equation (5) for nominal MDPs. Conditions 1 and 2 are also known to hold for rectangular Robust MDPs (see, e.g., Eq. (11) and Eq. (25) in Wiesemann et al., 2013). We emphasize that it is not obvious that these conditions hold, e.g., it is known that they fail for multi-model MDPs with static kernels (Steimle et al., 2021) (we provide a simple example in Example 3.8). Note that Conditions 1 and 2 do not imply that the operators have a *unique* fixed point, only that the value functions are in the sets of fixed points. Note that for robust MDPs to satisfy Conditions 1 and 2, the product structure is a necessary condition (as recently shown in Grand-Clément et al. (2024)), and without the convexity requirement on the support of $\nu$, even s-rectangular robust MDPs may fail to satisfy Conditions 1 and 2 (see Table 3 and Table 4 in Wang et al. (2023)). Adding the convex support and the product structure in the conditions only makes them more general than, for instance, requiring that Conditions 1 and 2 hold for all general ambiguity-averse MDPs (with no assumption on $\nu$).

We refer to Table 1 for a summary of the dynamic programming properties of several MDP variants. We also illustrate Conditions 1 and 2 in the next simple example.

**Example 3.8** (Continued from Example 3.4)**.** We show that Conditions 1 and 2 may fail to hold for the MDP instance from Figure 1 when $\rho = \mathbb{E}^\nu$ for static kernels. To do so, we compute $V^{\pi,\nu,\mathbb{E}^\nu}$ and show that it differs from $T^{\pi,\nu,\mathbb{E}^\nu} V^{\pi,\nu,\mathbb{E}^\nu}$. For $\gamma = 1/2$ we have

$$V^{\pi,\nu,\mathbb{E}^\nu} = \begin{pmatrix} \mathbb{E}\left(\sum_{t\geqslant 0}(1/2)^t \tilde{X}^t\right) \\ \mathbb{E}(0) \end{pmatrix} = \begin{pmatrix} 2\log(2) \\ 0 \end{pmatrix}$$

$$T^{\pi,\nu,\mathbb{E}^\nu} V^{\pi,\nu,\mathbb{E}^\nu} = \begin{pmatrix} 1 + \log(2)/2 \\ 0 \end{pmatrix} \neq V^{\pi,\nu,\mathbb{E}^\nu} .$$

Therefore, $\mathbb{E}$ violates Condition 1 for static kernels. It also violates Condition 2 for static kernels because the MDP has only one policy; see Appendix E for more detail.

## 4. Main Results

This section focuses on ambiguity-averse MDPs under the lens of dynamic programming. We first analyze the properties of optimal policies and derive algorithms to compute them. We then provide a complete characterization of law-invariant risk measures that satisfy Conditions 1 and 2.

### 4.1. Computing optimal policies

Dynamic programming has several useful practical and computational consequences, as it reduces policy evaluation and

policy optimization to fixed-point computations. In fact, when Conditions 1 and 2 both hold, an optimal policy can be chosen to be stationary and can be recovered from the optimal value function, as we show next.

**Theorem 4.1.** *Assume that $\rho$ is monotone and translation-invariant, and that Conditions 1 and 2 hold for static or resampled kernels. Let $\mathsf{M}_\nu \in \mathcal{M}_c$, with static or resampled kernels. Then there exists a* stationary *policy $\pi^\star \in \Pi_\mathsf{S}$ that is optimal for starting from any state:*

$$\exists\, \pi^\star \in \Pi_\mathsf{S}, \forall\, s \in \mathcal{S},\ \sup_{\pi \in \Pi_\mathsf{H}} \rho\left(V^{\pi,\tilde{P}}(s)\right) = \rho\left(V^{\pi^\star,\tilde{P}}(s)\right).$$

*Moreover, $T^{\nu,\rho}$ is a contraction and any policy $\pi \in \Pi_\mathsf{S}$ such that $T^{\nu,\rho}V^{\nu,\rho} = T^{\pi,\nu,\rho}V^{\nu,\rho}$ is optimal.*

Theorem 4.1 generalizes existing results in the RMDP literature (e.g., Theorem 4 in Wiesemann et al. (2013)). Note that when dynamic programming holds for *both* static *and* resampled kernels, the stationary optimal policy defined in Theorem 4.1 is optimal for both models of kernels. Under the assumptions of Theorem 4.1, an optimal policy can be computed efficiently; e.g., by value iteration and policy iteration, which both converge at a linear rate as we show next.

**Proposition 4.2.** *Assume that $\rho$ is monotone and translation-invariant, and that Conditions 1 and 2 hold for static or resampled kernels. Let $\mathsf{M}_\nu \in \mathcal{M}_c$, with static or resampled kernels.*

1. *(Value iteration) Let $(\pi^n)_{n\geqslant 0}$ be the sequence of policies generated by Algorithm 1. Then for all $n \in \mathbb{N}$,*

$$\|V^{\pi^n,\nu,\rho} - V^{\nu,\rho}\|_\infty \leqslant \frac{2\gamma^n}{1-\gamma}\|V^{\pi^0,\nu,\rho} - V^{\nu,\rho}\|_\infty .$$

2. *(Policy iteration) Let $(\pi^n)_{n\geqslant 0}$ be the sequence of policies generated by Algorithm 2. Then for all $n \in \mathbb{N}$, $V^{\pi^n,\nu,\rho} \leqslant V^{\pi^{n+1},\nu,\rho}$ and*

$$\|V^{\pi^n,\nu,\rho} - V^{\nu,\rho}\|_\infty \leqslant \gamma^n \|V^{\pi^0,\nu,\rho} - V^{\nu,\rho}\|_\infty .$$

We also derive a convex program for computing $V^{\nu,\rho}$ when $\rho$ satisfies a convexity assumption; see Appendix G. Deriving modern gradient techniques may require more assumptions on the risk measure $\rho$ (see Li et al. (2022b) for the case of robust MDPs, where the analysis is not an easy consequence of existing results for nominal MDPs). We note that the main bottleneck in value and policy iterations lies in evaluating the Bellman operators $T^{\nu,\rho}$ and $T^{\pi,\nu,\rho}$, as well as the fixed point $V^{\pi^n,\nu,\rho}$ in Algorithm 2. The specific computation to do this depends on the risk measure $\rho$ and the distribution $\nu$, see Appendix G for some examples.

### 4.2. Dynamic programming compatibility

We now characterize the risk measures for which Conditions 1 and 2 hold. We focus solely on *law-invariant* risk

**Algorithm 1** Ambiguity-averse Value Iteration

1: Initialize $\pi^0 \in \Pi_S, V^0 = V^{\pi^0, \nu, \rho}$.
2: **for** $n \geqslant 0$ **do**
3:     Compute $V^{n+1} = T^{\nu, \rho} V^n$
4:     Choose $\pi^{n+1}$ such that $T^{\nu, \rho} V^n = T^{\pi^{n+1}, \nu, \rho} V^n$
5: **end for**

---

**Algorithm 2** Ambiguity-averse Policy Iteration

1: Initialize $\pi^0 \in \Pi_S$.
2: **for** $n \geqslant 0$ **do**
3:     Compute $V^{\pi^n, \nu, \rho}$
4:     Choose $\pi^{n+1} \in \Pi_S$ such that

$$T^{\nu, \rho} V^{\pi^n, \nu, \rho} = T^{\pi^{n+1}, \nu, \rho} V^{\pi^n, \nu, \rho} \,,$$

and $\pi^{n+1} = \pi^n$ if possible
5:     **stop** if $\pi^{n+1} = \pi^n$.
6: **end for**
7: **return** $\pi^n$

---

measures here, which cover most of the common interpretable risk measures. We first establish some important implications of dynamic programming, namely, a law-invariant risk measure that satisfies either Condition 1 or Condition 2 must be the identity on constant random variables, positively homogeneous and additive for independent variables.

**Proposition 4.3.** *Let $\rho$ be a risk measure that is law-invariant and non-constant on $L_c^\infty(\mathbb{R})$. If $\rho$ satisfies Condition 1 or Condition 2 in the case of either static or resampled kernels, then the following statements hold:*

1. *$\rho$ on constant random variables is the identity:*
   $$\forall x \in \mathbb{R}, \quad \rho(x) = x \,.$$

2. *$\rho$ is positive homogeneous on $L_c^\infty(\mathbb{R})$:*
   $$\forall \tilde{X} \in L_c^\infty(\mathbb{R}), \forall \alpha \geqslant 0, \quad \rho(\alpha \tilde{X}) = \alpha \rho(\tilde{X}) \,.$$

3. *$\rho$ is additive independent on $L_c^\infty(\mathbb{R})$:*
   *for any independent[1] $\tilde{X}, \tilde{Y} \in L_c^\infty(\mathbb{R})$ we have,*
   $$\rho(\tilde{X} + \tilde{Y}) = \rho(\tilde{X}) + \rho(\tilde{Y}) \,.$$

We derive these results using a single MDP instance where the geometry of the transitions enforces algebraic constraints on the risk measure, see Appendix H.1. Combined with Conditions 1 and 2, transitions to a next state imply positive homogeneity, and transitions to two different next states imply additivity for independent variables. Note that item 1 and item 3 in Proposition 4.3 imply translation-invariance. Proposition 4.3 already rules out several of the standard risk measures, e.g., positive homogeneity rules out ERM and additivity rules out CVaR. To completely characterize law-invariant risk measures satisfying Conditions 1 and 2,

---

[1] A rigorous reformulation is provided in Remark B.4

---

we rely on an additional assumption (monotonicity) to invoke representation theorems from the literature on statistics (Mu et al., 2024), or to a stronger continuity notion ($W^1$-continuity).

Our first characterization requires the assumption that $\rho$ is monotone. We then use the results in Mu et al. (2024) to obtain that $\rho$ can be decomposed as a mixture of entropic risk measures; i.e., $\rho$ must have a representation of the form

$$\rho(\tilde{X}) = \int_{\overline{\mathbb{R}}} \mathsf{ERM}_a(\tilde{X}) d\mu(a), \quad \forall \, \tilde{X} \in L_c^\infty(\mathbb{R}) \,, \quad (9)$$

for some measure $\mu$ with support on the extended real line $\overline{\mathbb{R}} = \mathbb{R} \cup \{+\infty, -\infty\}$. A key technical step in our proof is bridging the gap between our properties on variables with convex support and the domain of the representation theorem. We do so by extending $\rho$ to all bounded random variables via a noise-addition technique that convexifies supports while preserving the algebraic structure imposed by Proposition 4.3. Positive homogeneity then implies, through a dilatation argument on the mixing measure $\mu$, that $\mu$ can only be non-zero at $-\infty$, 0 and $+\infty$. We further prove that $\rho$ must be multiplicative for non-negative random variables; i.e., for any independent $\tilde{X}, \tilde{Z} \in L_c^\infty(\mathbb{R}_+), \rho(\tilde{Z}\tilde{X}) = \rho(\tilde{Z})\rho(\tilde{X})$. Substituting independent uniform random variables into this multiplicativity identity yields quadratic polynomial equalities in the endpoints of the intervals. These equalities force $\mu$ to be a Dirac mass at $-\infty$, 0, or $+\infty$, leading to the following theorem.

**Theorem 4.4.** *Let $\rho$ be a risk measure that is law-invariant and non-constant on $L_c^\infty(\mathbb{R})$. Assume additionally that $\rho$ is monotone on $L_c^\infty(\mathbb{R})$.*

*For static kernels, the following are equivalent:*

1. *$\rho$ satisfies Condition 1.*

2. *$\rho$ satisfies Condition 2.*

3. *$\rho$ is the essential infimum or the essential supremum on $L_c^\infty(\mathbb{R})$.*

*For resampled kernels, the following are equivalent:*

1. *$\rho$ satisfies Condition 1.*

2. *$\rho$ satisfies Condition 2.*

3. *$\rho$ is the essential infimum, the essential supremum, or the expectation on $L_c^\infty(\mathbb{R})$.*

The proof of Theorem 4.4 is given in Appendix H. We note that the fact that the essential infimum and essential supremum satisfy Conditions 1 and 2 for the static and resampled kernels is already known from the literature on robust MDPs and optimistic MDPs (Wiesemann et al., 2013; Grand-Clément et al., 2023). The fact that expectation also satisfies Conditions 1 and 2 for resampled kernels is a consequence of seminal results on distributionally robust MDPs

(see, e.g., Section 4 in Xu & Mannor (2012)). Therefore, our main contribution in Theorem 4.4 is to prove that *only* these risk measures yield dynamic programming equations, for the class of monotone, law-invariant risk measures.

For our second characterization, we use a continuity assumption (specifically $W^1$-continuity, a form of "continuity in quantiles"; see Appendix H.3). Additivity and positive homogeneity give that $\rho(\tilde{X}) = \rho\left(\frac{1}{n}\sum_{i=1}^{n}\tilde{X}_i\right)$, for $\tilde{X} \in L_c^{\infty}(\mathbb{R})$ and $\tilde{X}_i$ some i.i.d. copies of $\tilde{X}$. Using $W^1$-continuity and the Law of Large Numbers, we get that it is necessary that $\rho$ coincides with the expectation, and we conclude by distinguishing between static kernels (where expectation doesn't satisfy dynamic programming) and resampled kernels (where expectation satisfies dynamic programming). This approach yields a more restrictive class of risk measures than Theorem 4.4 (since ess inf and ess sup are not $W^1$-continuous).

**Theorem 4.5.** *Let $\rho$ be a risk measure that is law-invariant and non-constant on $L_c^{\infty}(\mathbb{R})$. Assume additionally that $\rho$ is $W^1$-continuous on $L_c^{\infty}(\mathbb{R})$.*

*For static kernels, there is no such $\rho$ satisfying Condition 1 or Condition 2.*

*For resampled kernels, the only $\rho$ satisfying Condition 1 or Condition 2 is the expectation on $L_c^{\infty}(\mathbb{R})$.*

### 4.3. Discussion

**Main takeaways** We have established that dynamic programming implies several useful properties, including the existence of stationary optimal policies and the extension of classical algorithms to the ambiguity-averse framework. We have also shown that, in the class of monotone law-invariant risk measures, only ess inf, ess sup and $\mathbb{E}$ can satisfy dynamic programming equations. The risk measures ess inf and ess sup correspond to robust MDPs (Nilim & El Ghaoui, 2005; Iyengar, 2005). Our impossibility results explain why no new model compatible with dynamic programming has emerged in the wake of those seminal papers, despite substantial interest in other alternatives involving VaR or CVaR. Assuming stronger continuity properties (e.g., $W^1$-continuity) only further restricts the class of risk measures compatible with dynamic programming.

Our work also clarifies the computational role of how uncertain kernels change over time. Static kernels induce significantly harder optimization problems than resampled kernels, mirroring the corresponding distinction for multi-model MDPs. In fact, we note that our results closely align with the separations between NP-hard problems that require history-dependent policies (multi-model MDPs and percentile optimization with static kernels, e.g., Delage & Mannor (2010); Steimle et al. (2021)) and models that can

be solved efficiently and for which optimal stationary policies exist (e.g., robust and optimistic MDPs for both static and resampled kernels, see Wiesemann et al. (2013); Grand-Clément et al. (2023)). It would be interesting to understand if Conditions 1 and 2 are "equivalent" (in some sense) to the existence of optimal stationary policies. We also note that, under the assumptions of Theorem 4.4 or of Theorem 4.5, we establish that dynamic programming is valid for policy evaluation (Condition 1) if and only if it is valid for policy optimization (Condition 2). This is surprising since Condition 1 imposes a fixed-point equation for the value functions of all stationary policies, whereas Condition 2 imposes a fixed-point equation only for the optimal value function.

**Limitations and assumptions** The translation-invariance and monotonicity assumptions are quite natural when the objective is to maximize returns, and are widely used in the risk measure literature. As discussed in Section 3.2, we could have written Conditions 1 and 2 without the restriction to instances with convex support and product structure, but (a) this would actually lead to more restrictive conditions, since the fixed point equations would need to hold for a larger set of instances, and (b) robust MDPs and optimistic MDPs would not satisfy these more restrictive conditions (Wang et al., 2023; Grand-Clément et al., 2024). So in a sense, our formulations of dynamic programming as in Conditions 1 and 2 are the "most general formulations" for which we can recover robust and optimistic MDPs. Finally, we show in Appendix I that relaxed versions of Conditions 1 and 2, where the risk measure is allowed to be different in the objective function and the Bellman operators, are equivalent to our formulation of Conditions 1 and 2.

**Extensions to more general setups.** The impossibility side of our results extends immediately to any proposed framework that asks Conditions 1 and 2 to hold for classes of instances where finite state and action sets are a special case, e.g. to MDPs with measure state and action sets. Positive results in general spaces would require additional measurable-selection and continuity assumptions to make the Bellman operators and risk functionals well-posed. Average- or total-reward criteria would also require a different technical setup: one must first specify the class of instances for which the criterion is well-defined. Rewriting and analyzing Conditions 1 and 2 in those setups is a promising direction.

**Beyond dynamic programming** Our results do not preclude that one can solve the ambiguity-averse MDP problem (6) via other types of algorithms than dynamic programming over the set of states. In particular, while our results imply that attractive *law-invariant* risk criteria (such as VaR or CVaR) do not satisfy dynamic programming for value functions indexed solely by states, it is possible that one could obtain DP equations on *augmented* state sets as in risk-

averse MDPs (Chow et al., 2015; Hau et al., 2023a). Another option would be to drop the law-invariance assumption and take inspiration from nested risk measures (Ruszczyński, 2010) to recover DP equations over the set of states (at the price of less interpretable risk criteria). Generalizing the dynamic principles to *collections of risk measures* in the same spirit as Bellman closedness in distributional reinforcement learning (Rowland et al., 2019) is also an interesting next step. Finally, it is worth noting that direct optimization of the objective function (6) (e.g., via gradient descent) is possible as soon as one can compute gradient estimates, although the potential absence of optimal *stationary* policies may be an issue for practical implementation.

## 5. Conclusion

We have introduced a unified theory of epistemic uncertainty in MDPs, in which the transition probabilities are treated as random variables. This ambiguity-averse MDP framework subsumes several existing models depending on the choice of risk measures in the objective functions. We define the ambiguity-averse equivalent of value functions and derive the fundamental properties of Bellman operators. Our main results show both the "advantages" of dynamic programming (stationary optimality and classical algorithmic machinery), and its modeling implications, as we provide a comprehensive characterization of law-invariant risk measures compatible with dynamic programming. This provides useful takeaways for applications, where we can choose to stick with robust/optimistic formulations for dynamic programming tractability, or where we must anticipate the need for more advanced optimization methods (e.g., augmented state sets or nested risk measures).

It is worth investigating in more detail the connections with other variants, such as risk-averse MDPs, regularized MDPs, and distributionally robust MDPs. Studying ambiguity-averse MDPs with other objectives (e.g., total or average return, Blackwell optimality) or with non-law-invariant risk measures is also a promising research direction. Deriving policy-gradient methods or value function approximation methods for ambiguity-averse MDPs with general risk measures is also an interesting next step.

## Impact Statement

This paper advances machine learning theory. As with any theoretical advance, there are many potential indirect societal consequences, but they are difficult to predict or contemplate.

## Acknowledgements

Axel Benyamine, Michael I. Jordan and Alain Durmus are funded by the European Union (ERC-2022-SYG-OCEAN-101071601). Views and opinions expressed are however those of the author(s) only and do not necessarily reflect those of the European Union or the European Research Council Executive Agency. Neither the European Union nor the granting authority can be held responsible for them. Julien Grand-Clément and Alain Durmus were supported by Hi! Paris and Agence Nationale de la Recherche (Grant 11-LABX-0047). Alain Durmus a reçu un financement de la Fondation de l'École polytechnique dans le cadre de sa campagne Servir la science. Alain Durmus is supported by the France 2030 program with the reference ANR-25-PEIA-0001 (THEOREM project).

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

# A. Risk measures

In this section, we define general risk measures rigorously, present some common examples, and outline some of the most studied properties of risk measures. We follow the lines of Chapter 4 in Föllmer & Schied (2016) and Appendix A in Hau et al. (2023b) here.

**Definitions.** Let $(\Omega, \mathcal{F}, \mathbb{P})$ be a probability space. The set $L(\Omega, \mathcal{F}, \mathbb{P}; \mathbb{R})$ is the set of real random variables, i.e. of $\mathcal{F}$-measurable functions from $\Omega$ to $\mathbb{R}$. A risk measure $\rho$ is a map from the set $L(\Omega, \mathcal{F}, \mathbb{P}; \mathbb{R})$ to $\mathbb{R}$.

**Definition A.1** (Support of a Random Variable and Support of a Measure)**.**
The **support** of a random variable $\tilde{X}$ is denoted $\mathrm{supp}(\tilde{X})$, and it is defined as the smallest closed set $C \subseteq \mathbb{R}$ such that $\mathbb{P}(\tilde{X} \in C) = 1$.
More generally, for a probability measure $\nu$ on $\mathbb{R}^n$, the **support** $\mathrm{supp}(\nu)$ of $\nu$ is the smallest closed set $C \subseteq \mathbb{R}^n$ such that $\nu(C) = 1$, or equivalently, the complement of the largest open set having $\nu$-measure zero.
We say that a measure $\nu$ has convex support if its support $\mathrm{supp}(\nu)$ is convex, i.e., for any $\lambda \in [0,1]$ and $P, P' \in \mathrm{supp}(\nu) \subseteq \mathbb{R}^n$, we have $\lambda P + (1-\lambda)P' \in \mathrm{supp}(\nu)$. Accordingly, we say that a random variable $\tilde{X}$ has convex support if its support $\mathrm{supp}(\tilde{X})$ is convex.

Since by definition $\mathrm{supp}(\tilde{X})$ is closed, we have that $\mathrm{supp}(\tilde{X})$ is compact if and only if the random variable $\tilde{X}$ is a.s. bounded. Similarly, the support of a measure $\nu$ is compact if and only if there exists a bounded set with full $\nu$-measure.

**Properties.** Risk measures satisfying the monotonicity and translation-invariance properties given in Section 2.2 are sometimes called **monetary**. A risk measure is said to be **concave** if it is monetary and it satisfies, for any random variables $\tilde{X}, \tilde{Y}$ and scalar $\lambda \in [0,1]$, the following inequality:

$$\rho(\lambda \tilde{X} + (1-\lambda)\tilde{Y}) \geqslant \lambda \rho(\tilde{X}) + (1-\lambda)\rho(\tilde{Y}) \,.$$

A **coherent** risk measure is a concave risk measure that is positively homogeneous, i.e., that satisfies the following condition:

$$\forall \, \alpha \geqslant 0, \ \forall \tilde{X}, \quad \rho(\alpha \tilde{X}) = \alpha \rho(\tilde{X}) \,.$$

Finally, a risk measure $\rho$ is **law-invariant** if it only depends on the law of its argument, i.e. if $\rho(\tilde{X}) = \rho(\tilde{Y})$ when $\mathcal{L}(\tilde{X}) = \mathcal{L}(\tilde{Y})$, where $\mathcal{L}(\tilde{X})$ is the law of the random variable $\tilde{X}$. Note that we will generalize this notion for the ambiguity-averse MDP framework in Appendix B and Section 3.1.

**Examples.** All the examples discussed below are law-invariant.

- The expectation $\tilde{X} \mapsto \mathbb{E}(\tilde{X})$ is a simple example of a coherent risk measure. As a side note, the variance $\tilde{X} \mapsto \mathbb{E}\left((\tilde{X} - \mathbb{E}(\tilde{X}))^2\right)$ is not a monetary risk measure (it is easy to see that it is not monotone nor translation-invariant).

- The essential supremum $\tilde{X} \mapsto \mathrm{ess}\sup(\tilde{X})$ and essential infimum $\tilde{X} \mapsto \mathrm{ess}\inf(\tilde{X})$ are defined as

$$\mathrm{ess}\inf(\tilde{X}) = \sup\{b \in \mathbb{R} \mid \mathbb{P}(\tilde{X} < b) = 0\}$$
$$\mathrm{ess}\sup(\tilde{X}) = \inf\{b \in \mathbb{R} \mid \mathbb{P}(b < \tilde{X}) = 0\}$$

Both $\mathrm{ess}\inf$ and $\mathrm{ess}\sup$ are monotone and positively homogeneous, but only $\mathrm{ess}\inf$ is concave (hence coherent). Additionally, $\mathrm{ess}\inf$ and $\mathrm{ess}\sup$ coincide with the infimum and supremum of the variables, under some simple assumptions on the support of the random variables, as stated more formally below.

**Lemma A.2.** *Let $\tilde{X}$ be an a.s. bounded random variable. Then $\mathrm{ess}\inf$ and $\mathrm{ess}\sup$ coincide with the infimum and the supremum of the support of $\tilde{X}$:*

$$\mathrm{ess}\inf(\tilde{X}) = \inf\mathrm{supp}(\tilde{X})$$
$$\mathrm{ess}\sup(\tilde{X}) = \sup\mathrm{supp}(\tilde{X}) \,.$$

*Proof of Lemma A.2.* Recall that by definition (Definition A.1), the support of $\tilde{X}$ is the smallest closed set $C$ such that $\mathbb{P}(\tilde{X} \in C) = 1$. We provide the proof for the essential supremum, the proof for the essential infimum follows exactly the same line. As noted in Definition A.1, $\mathrm{supp}(\tilde{X})$ is compact if and only if $\tilde{X}$ is a.s. bounded.

Let $M = \sup \mathrm{supp}(\tilde{X})$. Since $\mathrm{supp}(\tilde{X})$ is compact, we get that $M \in \mathrm{supp}(\tilde{X})$. By definition we get that $\mathbb{P}(\tilde{X} > M) = 0$ so that $\mathrm{ess}\sup(\tilde{X}) \leqslant M$. Additionally, since $M \in \mathrm{supp}(\tilde{X})$, any neighborhood of $M$ has positive mass, i.e. for every $\epsilon > 0$ we have $\mathbb{P}(\tilde{X} > M - \epsilon) > 0$. We conclude that $M - \epsilon \leqslant \mathrm{ess}\sup(\tilde{X})$ for any $\epsilon > 0$, i.e. we conclude that $M \leqslant \mathrm{ess}\sup(\tilde{X})$. Therefore we have shown that $M = \mathrm{ess}\sup(\tilde{X})$, which concludes the proof. $\square$

- The Value-at-Risk ($\mathsf{VaR}_\alpha$) is defined as the upper $(1 - \alpha)$-quantile:

$$\mathsf{VaR}_\alpha(\tilde{X}) = \inf\{x \in \mathbb{R} \mid \mathbb{P}(\tilde{X} \leqslant x) > 1 - \alpha\}$$

for some $\alpha \in (0, 1)$, and the Conditional Value-at-Risk ($\mathsf{CVaR}_\alpha$) is the expectation of the worst $(1 - \alpha)$-fraction of $\tilde{X}$. $\mathsf{CVaR}$ can be computed via the following formula:

$$\mathsf{CVaR}_\alpha(\tilde{X}) = \sup_{\xi \in \mathbb{R}} \xi - \frac{1}{1-\alpha}\mathbb{E}(\max\{\xi - \tilde{X}, 0\}) . \tag{10}$$

$\mathsf{CVaR}$ is a concave risk measure, and positively homogeneous, thus $\mathsf{CVaR}$ is coherent. In contrast, $\mathsf{VaR}$ is only monetary. Note we can extend $\mathsf{CVaR}$ by continuity, as $\mathsf{CVaR}_0(\tilde{X}) = \mathbb{E}(\tilde{X}), \mathsf{CVaR}_1(\tilde{X}) = \mathrm{ess}\inf(\tilde{X})$.

- The Entropic Risk Measure ($\mathsf{ERM}$) at risk level $\beta \in \mathbb{R}$ is defined as

$$\mathsf{ERM}_\beta(\tilde{X}) = \frac{1}{\beta}\log(\mathbb{E}(\exp(\beta \cdot \tilde{X})))$$

extended by continuity as $\mathsf{ERM}_0(\tilde{X}) = \mathbb{E}(\tilde{X}) \mathsf{ERM}_{+\infty}(\tilde{X}) = \mathrm{ess}\sup(\tilde{X}), \mathsf{ERM}_{-\infty}(\tilde{X}) = \mathrm{ess}\inf(\tilde{X})$. $\mathsf{ERM}$ is convex for $\beta > 0$ and concave for $\beta \leqslant 0$. The Entropic Value-at-Risk ($\mathsf{EVaR}$) is defined as

$$\mathsf{EVaR}_\alpha(\tilde{X}) = \sup_{\beta > 0} \mathsf{ERM}_\beta(\tilde{X}) + \frac{\log(1-\alpha)}{\beta} .$$

We note that $\mathsf{ERM}$ is concave but not coherent, while $\mathsf{EVaR}$ is a coherent risk measure. We refer to Hau et al. (2023b) for recent advances in using $\mathsf{ERM}$ and $\mathsf{EVaR}$ in risk-averse MDPs.

For the sake of readability, we summarize the properties of some common risk measures in Table 2. Note that most of this

*Table 2.* Properties of some common risk measures. The term "Monetary" means that the risk measure is monotone and translation-invariant. The column "$W^1$-$C^0$" refers to the continuity in the $W^1$-sense defined in Appendix H.3 (see Definition H.7).

| Risk measure | Notation | Risk properties | | | | |
|---|---|---|---|---|---|---|
| | | Coherent | Concave | Monetary | Law invariant | $W^1$-$C^0$ |
| Expectation | $\mathbb{E}(\tilde{X})$ | ✓ | ✓ | ✓ | ✓ | ✓ |
| Essential Supremum | $\mathrm{ess}\sup(\tilde{X})$ | ✗ | ✗ | ✓ | ✓ | ✗ |
| Essential Infimum | $\mathrm{ess}\inf(\tilde{X})$ | ✓ | ✓ | ✓ | ✓ | ✗ |
| Value-at-Risk | $\mathsf{VaR}_\alpha(\tilde{X})$ | ✗ | ✗ | ✓ | ✓ | ✗ |
| Cond. Value-at-Risk | $\mathsf{CVaR}_\alpha(\tilde{X})$ | ✓ | ✓ | ✓ | ✓ | ✓ |
| Entropic Risk Measure | $\mathsf{ERM}_\beta(\tilde{X})$ | ✗ | ✓ (for $\beta \leqslant 0$) | ✓ | ✓ | ✓ |
| Entropic Value-at-Risk | $\mathsf{EVaR}_\alpha(\tilde{X})$ | ✓ | ✓ | ✓ | ✓ | ✓ |

work focuses on risk measures that are law-invariant. For the sake of completeness, we provide a simple example of a non-law-invariant risk measure below.

**Example A.3** (A Non-Law-Invariant Risk Measure)**.**
Consider a probability space $(\Omega, \mathcal{F}, \mathbb{P})$ where $\Omega = \{\omega_1, \omega_2\}$ with $\mathbb{P}(\{\omega_i\}) = 1/2$ for $i \in \{1, 2\}$. Define a risk measure $\rho$ as

follows: for a random variable $\tilde{X} : \Omega \to \mathbb{R}$, let $\rho(\tilde{X}) = \tilde{X}(\omega_1) + \mathbb{E}[\tilde{X}]$, i.e., the risk measure depends explicitly on the value at a specific outcome $\omega = \omega_1$ in addition to the expectation.

This risk measure is *not law-invariant*. To see this, consider two random variables:

$$\tilde{X}(\omega) = \begin{cases} 10 & \text{if } \omega = \omega_1 \\ 0 & \text{if } \omega = \omega_2 \end{cases}$$

$$\tilde{Y}(\omega) = \begin{cases} 0 & \text{if } \omega = \omega_1 \\ 10 & \text{if } \omega = \omega_2 \end{cases}$$

Both $\tilde{X}$ and $\tilde{Y}$ have the same law: each takes the value 10 with probability $1/2$ and the value 0 with probability $1/2$.

However, we have:

$$\rho(\tilde{X}) = \tilde{X}(\omega_1) + \mathbb{E}[\tilde{X}] = 10 + 10/2 = 15$$
$$\rho(\tilde{Y}) = \tilde{Y}(\omega_1) + \mathbb{E}[\tilde{Y}] = 0 + 10/2 = 5$$

Since $\rho(\tilde{X}) \neq \rho(\tilde{Y})$ despite the fact that $\tilde{X}, \tilde{Y}$ have the same law, the risk measure $\rho$ is not law-invariant. The dependence on the specific outcome $\tilde{X}(\omega_1)$ violates law-invariance because the risk measure distinguishes between different realizations of the same distribution.

In the context of ambiguity-averse MDPs, non-law-invariant risk measures would depend on the specific sample path or realization of the uncertain transition kernel, rather than only on its distribution. This makes them less interpretable and difficult to work with, as they require tracking which particular realization of the uncertainty occurred, rather than just its statistical properties.

**Remark A.4.** We conclude by noting that risk measures are sometimes introduced for *minimizing costs* (e.g. this is the point of view adopted in Shapiro et al. (2021)), instead of *maximizing rewards* as in the present paper. These two settings are equivalent, in the sense that if we have a risk measure $\phi$ for minimizing costs, then we can consider the risk measure for maximizing reward $\rho(\tilde{X}) := -\phi(-\tilde{X})$, which induces the same preferences over the random variables $\tilde{X}$ modeling rewards than the risk measure $\phi$ over random variables $-\tilde{X}$ modeling costs. It is worth noting that, in the cost minimization framework, the role of concavity is replaced by that of convexity and translation invariance becomes $\phi(\tilde{X} + c) = \phi(\tilde{X}) - c$ for $\tilde{X}$ a random variable and $c \in \mathbb{R}$.

## B. Randomization and Risk Measure for Ambiguity-Averse MDPs

In this section, we provide a rigorous foundation for the ambiguity-averse MDPs introduced in Section 3. We decide to keep this construction in the appendices to simplify the exposition of our results in the main body. Overall, our goal is to provide a rigorous framework for the objective functions of the form $\pi \mapsto \rho(\mu^\top V^{\pi, \tilde{P}})$ that appear in the ambiguity-averse MDP optimization problem (6). Recall that by definition, a risk measure is a map from the set of real random variables $L(\Omega, \mathcal{F}, \mathbb{P})$. However, to rigorously model ambiguity-averse MDPs, we need one probability space $(\Omega_\mathsf{M}, \mathcal{F}_\mathsf{M}, \mathbb{P}_\mathsf{M})$ per instance $\mathsf{M}$ of ambiguity-averse MDP, and in principle, there is a distinct risk measure for each ambiguity-averse MDP instance (since the set of arguments $(\Omega_\mathsf{M}, \mathcal{F}_\mathsf{M}, \mathbb{P}_\mathsf{M})$ depends on the instance). We therefore extend the notion of risk measure (defined in Appendix A for a single probability space) to risk measures for ambiguity-averse MDPs (Definition B.2), which can take as arguments random variables induced by different ambiguity-averse MDP instances (i.e., induced by different probability spaces).

**Notation for sets of ambiguity-averse MDPs**    Recall that we denote by $\mathcal{M}_c$ the class of all "convex, product" ambiguity-averse MDPs, i.e. the set of all $\mathsf{M}_\nu = (\mathcal{S}, \mathcal{A}, r, \nu, \gamma, \mu)$ with any $\mathcal{S}, \mathcal{A}, r, \gamma, \mu$, and any $\nu$ with convex support and product structure. We also denote by $\mathcal{M}$ the set of all ambiguity-averse MDPs, i.e. the set of all $\mathsf{M}_\nu = (\mathcal{S}, \mathcal{A}, r, \nu, \gamma, \mu)$ with any $\mathcal{S}, \mathcal{A}, r, \gamma, \mu$ and any $\nu$ (not necessarily with convex support or product structure).

**Probability space for an instance.**    We first define a probability space $(\Omega, \mathcal{F}, \mathbb{P})$ for each ambiguity-averse MDP.

We provide a slightly more rigorous definition of an ambiguity-averse MDP, which allows us to define a distribution over *sequences* of transition probabilities directly.

**Definition B.1.** A (discounted) *ambiguity-averse MDP* is a tuple $\mathsf{M}_{\hat{\nu}} = (\mathcal{S}, \mathcal{A}, r, \hat{\nu}, \gamma, \mu)$ where $\mathcal{S}, \mathcal{A}, r, \gamma, \mu$ are defined exactly as in nominal MDPs (see Section 2) and $\hat{\nu}$ is a probability distribution over the set $(\Delta(\mathcal{S})^{\mathcal{S} \times \mathcal{A}})^{\mathbb{N}}$ of feasible sequences of transition probabilities.

When $\hat{\nu}$ is equal to $\nu^{\otimes \mathbb{N}}$ for a certain measure $\nu \in \Delta(\mathcal{S})^{\mathcal{S} \times \mathcal{A}}$, $\mathsf{M}_{\hat{\nu}}$ corresponds to an ambiguity-averse MDP with resampled kernels according to $\nu$. Here $\nu^{\otimes \mathbb{N}}$ is the i.i.d. distribution over the path space $(\Delta(\mathcal{S})^{\mathcal{S} \times \mathcal{A}})^{\mathbb{N}}$. When $\hat{\nu}$ is equal to the distribution of $(\tilde{P}_t)_{t \in \mathbb{N}}$ such that $\tilde{P}_0 \sim \nu$ and $\tilde{P}_t = \tilde{P}_0$ for any $t$, $\mathsf{M}_{\hat{\nu}}$ corresponds to an ambiguity-averse MDP with static kernel.

For any ambiguity-averse MDP $\mathsf{M}_{\hat{\nu}} = (\mathcal{S}, \mathcal{A}, r, \hat{\nu}, \gamma, \mu) \in \mathcal{M}$, we define the uncertainty in the model transition kernel sequences via a probability space $(\Omega_{\mathsf{M}}, \mathcal{F}_{\mathsf{M}}, \mathbb{P}_{\mathsf{M}})$, where:

- $(\Omega_{\mathsf{M}}, \mathcal{F}_{\mathsf{M}})$ is a measurable space with $\Omega_{\mathsf{M}} = (\Delta(\mathcal{S})^{\mathcal{S} \times \mathcal{A}})^{\mathbb{N}}$ the set of transition kernel sequences and $\mathcal{F}_{\mathsf{M}}$ the canonical $\sigma$-field $\mathcal{B}(\Omega_{\mathsf{M}}) = \mathcal{B}(\Delta(\mathcal{S})^{\mathcal{S} \times \mathcal{A}})^{\otimes \mathbb{N}}$,

- $\mathbb{P}_{\mathsf{M}} = \hat{\nu}$ is a probability measure on $(\Omega_{\mathsf{M}}, \mathcal{F}_{\mathsf{M}})$.

Without loss of generality, we consider $(\tilde{P}_t)_{t \in \mathbb{N}}$ to be the canonical process $\tilde{P}_t : (p_s)_{s \in \mathbb{N}} \mapsto p_t$. In particular, it follows law $\hat{\nu}$ in the sense of Definition B.1.

When $\hat{\nu}$ is non-atomic (i.e., it has no set of positive measure that contains no subsets of smaller positive measure), the probability space $(\Omega_{\mathsf{M}}, \mathcal{F}_{\mathsf{M}}, \mathbb{P}_{\mathsf{M}})$ is **nonatomic** (i.e., it supports a random variable uniformly distributed on $[0, 1]$) and therefore the space $L^\infty(\Omega_{\mathsf{M}}, \mathcal{F}_{\mathsf{M}}, \mathbb{P}_{\mathsf{M}}; \mathbb{R})$ is rich enough to contain real bounded random variables with arbitrary distributions, see for instance Delage et al. (2019).
Note that by Definition 3.1, the set $\mathcal{M}_c$ of all ambiguity-averse MDPs such that the measure has convex support and product structure contains ambiguity-averse MDPs with non-atomic measure in both static kernels or resampled kernels cases (say for example the ambiguity-averse MDP of Example E.1 in both static and resampled cases).

Formally, denoting by $\mathfrak{D}$ the set of all distributions with bounded support (i.e., $\mathfrak{D}$ is the set of non-decreasing right-continuous functions $F : \mathbb{R} \to [0, 1]$ that attain both 0 and 1), for any $F \in \mathfrak{D}$ there exists $\mathsf{M}_\nu \in \mathcal{M}_c$ and $\tilde{X} \in L^\infty(\Omega_{\mathsf{M}}, \mathcal{F}_{\mathsf{M}}, \mathbb{P}_{\mathsf{M}}; \mathbb{R})$ such that $F_{\tilde{X}} = F$, where $F_{\tilde{X}}$ is the distribution function of $\tilde{X}$ under $\mathbb{P}_{\mathsf{M}_\nu}$ defined through $F_{\tilde{X}}(x) = \mathbb{P}_{\mathsf{M}}(\tilde{X} \leqslant x)$.

**Notation for sets of random variables**  Throughout this work, we denote by $L^\infty(B) = \cup_{\mathsf{M} \in \mathcal{M}} L^\infty(\Omega_{\mathsf{M}}, \mathcal{F}_{\mathsf{M}}, \mathbb{P}_{\mathsf{M}}; B)$ the set of all bounded real random variables with support in $B \subseteq \mathbb{R}$ across all these probability spaces, and by $L_c^\infty(B) = \cup_{\mathsf{M} \in \mathcal{M}} L_c^\infty(\Omega_{\mathsf{M}}, \mathcal{F}_{\mathsf{M}}, \mathbb{P}_{\mathsf{M}}; B)$ its analog restricted to random variables with convex supports in $B$.
Note that, by nonatomicity of some probability spaces $(\Omega_{\mathsf{M}}, \mathcal{F}_{\mathsf{M}}, \mathbb{P}_{\mathsf{M}})$ (for example the one of Example E.1 whether we consider static or resampled kernels), the set $L^\infty(\mathbb{R})$ (resp. $L_c^\infty(\mathbb{R})$) is rich enough to contain real bounded random variables (resp. bounded real random variables with convex support) with arbitrary distributions.

**Risk measures**  We now define the notion of risk measure for ambiguity-averse MDPs:

**Definition B.2** (Risk Measure for Ambiguity-Averse MDPs)**.** A **risk measure for ambiguity-averse MDPs** is a family $\rho = (\rho_{\mathsf{M}})_{\mathsf{M} \in \mathcal{M}}$ where each $\rho_{\mathsf{M}}$ is a risk measure on the corresponding probability space:

$$\forall \mathsf{M} \in \mathcal{M}, \quad \rho_{\mathsf{M}} : L^\infty(\Omega_{\mathsf{M}}, \mathcal{F}_{\mathsf{M}}, \mathbb{P}_{\mathsf{M}}; \mathbb{R}) \to \mathbb{R} \,.$$

We also define the following properties of a risk measure $\rho$. We say that a risk measure $\rho$ is:

- **law-invariant** on $L^\infty(\mathbb{R})$ (resp. $L_c^\infty(\mathbb{R})$) if for any $\mathsf{M}, \mathsf{M}' \in \mathcal{M}$ and $\tilde{X}, \tilde{X}' \in L^\infty(\Omega_{\mathsf{M}}, \mathcal{F}_{\mathsf{M}}, \mathbb{P}_{\mathsf{M}}; \mathbb{R}) \times L^\infty(\Omega_{\mathsf{M}'}, \mathcal{F}_{\mathsf{M}'}, \mathbb{P}_{\mathsf{M}'}; \mathbb{R})$ (resp. $L_c^\infty(\Omega_{\mathsf{M}}, \mathcal{F}_{\mathsf{M}}, \mathbb{P}_{\mathsf{M}}; \mathbb{R}) \times L_c^\infty(\Omega_{\mathsf{M}'}, \mathcal{F}_{\mathsf{M}'}, \mathbb{P}_{\mathsf{M}'}; \mathbb{R}))$,

$$F_{\tilde{X}} = F_{\tilde{X}'} \implies \rho_{\mathsf{M}}(\tilde{X}) = \rho_{\mathsf{M}'}(\tilde{X}') \,.$$

- **non-constant** on $L^\infty(\mathbb{R})$ (resp. $L_c^\infty(\mathbb{R})$) if there exists $\mathsf{M}, \mathsf{M}' \in \mathcal{M}$ and $\tilde{X}, \tilde{X}' \in L^\infty(\Omega_{\mathsf{M}}, \mathcal{F}_{\mathsf{M}}, \mathbb{P}_{\mathsf{M}}; \mathbb{R}) \times L^\infty(\Omega_{\mathsf{M}'}, \mathcal{F}_{\mathsf{M}'}, \mathbb{P}_{\mathsf{M}'}; \mathbb{R})$ (resp. $L_c^\infty(\Omega_{\mathsf{M}}, \mathcal{F}_{\mathsf{M}}, \mathbb{P}_{\mathsf{M}}; \mathbb{R}) \times L_c^\infty(\Omega_{\mathsf{M}'}, \mathcal{F}_{\mathsf{M}'}, \mathbb{P}_{\mathsf{M}'}; \mathbb{R}))$ such that

$$\rho_{\mathsf{M}}(\tilde{X}) \neq \rho_{\mathsf{M}'}(\tilde{X}') \,.$$

- **monotone** on $L^\infty(\mathbb{R})$ (resp. $L_c^\infty(\mathbb{R})$) if for any $\mathsf{M} \in \mathcal{M}$ and $\tilde{X}, \tilde{Y} \in L^\infty(\Omega_{\mathsf{M}'}, \mathcal{F}_{\mathsf{M}'}, \mathbb{P}_{\mathsf{M}'}; \mathbb{R})$ (resp. $L_c^\infty(\Omega_{\mathsf{M}'}, \mathcal{F}_{\mathsf{M}'}, \mathbb{P}_{\mathsf{M}'}; \mathbb{R})$) such that $\tilde{X} \geqslant \tilde{Y}$ a.s.,

$$\rho_{\mathsf{M}}(\tilde{X}) \geqslant \rho_{\mathsf{M}}(\tilde{Y}) \,.$$

For ease of notation, we drop the subscript $\mathsf{M}$ when considering risk measures for ambiguity-averse MDPs that are law-invariant on $L^\infty(\mathbb{R})$ (resp. $L_c^\infty(\mathbb{R})$) and note $\rho_{\mathsf{M}}(\tilde{X})$ as $\rho(\tilde{X})$ for $\tilde{X} \in L^\infty(\Omega_{\mathsf{M}}, \mathcal{F}_{\mathsf{M}}, \mathbb{P}_{\mathsf{M}}; \mathbb{R})$ (resp. $\tilde{X} \in L_c^\infty(\Omega_{\mathsf{M}}, \mathcal{F}_{\mathsf{M}}, \mathbb{P}_{\mathsf{M}}; \mathbb{R})$).

As expected, we also have the following characterization of law-invariant risk measures for ambiguity-averse MDPs:

**Proposition B.3** (Existence and Uniqueness of $\varrho$). *If $\rho$ is a risk measure for ambiguity-averse MDPs that is law invariant on $L^\infty(\mathbb{R})$, then there exists a unique functional $\varrho : \mathfrak{D} \to \mathbb{R}$ such that:*

$$\rho_{\mathsf{M}}(\tilde{X}) = \varrho(F_{\tilde{X}}), \quad \forall \mathsf{M} \in \mathcal{M}, \forall \tilde{X} \in L^\infty(\Omega_{\mathsf{M}}, \mathcal{F}_{\mathsf{M}}, \mathbb{P}_{\mathsf{M}}; \mathbb{R}) \,.$$

The proof directly follows from Proposition 1 in Delage et al. (2019).

*Proof.* Denoting by $\mathsf{M}_0$ the MDP of Example E.1 (whether we consider static or resampled kernels), we have that $(\Omega_{\mathsf{M}_0}, \mathcal{F}_{\mathsf{M}_0}, \mathbb{P}_{\mathsf{M}_0})$ is nonatomic, so by Proposition 1 in Delage et al. (2019) there exists a unique functional $\varrho : \mathfrak{D} \to \mathbb{R}$ such that for any $\tilde{X} \in L^\infty(\Omega_{\mathsf{M}_0}, \mathcal{F}_{\mathsf{M}_0}, \mathbb{P}_{\mathsf{M}_0}; \mathbb{R})$:

$$\rho_{\mathsf{M}_0}(\tilde{X}) = \varrho(F_{\tilde{X}}).$$

Now, for any $\mathsf{M} \in \mathcal{M}$ and $\tilde{X} \in L^\infty(\Omega_{\mathsf{M}}, \mathcal{F}_{\mathsf{M}}, \mathbb{P}_{\mathsf{M}}; \mathbb{R})$, because $(\Omega_{\mathsf{M}_0}, \mathcal{F}_{\mathsf{M}_0}, \mathbb{P}_{\mathsf{M}_0})$ is nonatomic, there exists $\tilde{X}_0 \in L^\infty(\Omega_{\mathsf{M}_0}, \mathcal{F}_{\mathsf{M}_0}, \mathbb{P}_{\mathsf{M}_0}; \mathbb{R})$ such that $F_{\tilde{X}} = F_{\tilde{X}_0}$. Therefore, by law invariance of $\rho$:

$$\rho_{\mathsf{M}}(\tilde{X}) = \rho_{\mathsf{M}_0}(\tilde{X}_0) = \varrho(F_{\tilde{X}_0}) = \varrho(F_{\tilde{X}}) \,.$$

Finally, if $\varrho, \varrho'$ are such that $\forall \mathsf{M} \in \mathcal{M}, \forall \tilde{X} \in L^\infty(\Omega_{\mathsf{M}}, \mathcal{F}_{\mathsf{M}}, \mathbb{P}_{\mathsf{M}}; \mathbb{R}), \quad \varrho'(F_{\tilde{X}}) = \rho_{\mathsf{M}}(\tilde{X}) = \varrho(F_{\tilde{X}})$, then, because $(\Omega_{\mathsf{M}_0}, \mathcal{F}_{\mathsf{M}_0}, \mathbb{P}_{\mathsf{M}_0})$ is nonatomic, for any $F \in \mathfrak{D}$ there exists $\tilde{X} \in L^\infty(\Omega_{\mathsf{M}_0}, \mathcal{F}_{\mathsf{M}_0}, \mathbb{P}_{\mathsf{M}_0}; \mathbb{R})$ such that $F_{\tilde{X}} = F$, and therefore $\varrho'(F) = \varrho(F)$. This shows uniqueness of $\varrho$. $\square$

**Remark B.4.** When adding or multiplying random variables $\tilde{X}$ and $\tilde{Y}$, it is important to note that the risk measure $\rho$ is only defined on bounded random variables living in one of the probability spaces $((\Omega_{\mathsf{M}}, \mathcal{F}_{\mathsf{M}}, \mathbb{P}_{\mathsf{M}}))_{\mathsf{M} \in \mathcal{M}}$. Consequently, to make sense of expressions like $\rho(\tilde{X} + \tilde{Y})$ or $\rho(\tilde{X} \cdot \tilde{Y})$, the random variables $\tilde{X}$ and $\tilde{Y}$ must be defined on the same probability space $(\Omega_{\mathsf{M}}, \mathcal{F}_{\mathsf{M}}, \mathbb{P}_{\mathsf{M}})$ for some ambiguity-averse MDP $\mathsf{M} \in \mathcal{M}$. When we state properties like additivity or multiplicativity for independent random variables, we implicitly assume they are defined on the same probability space and are independent with respect to the probability measure on that space. For instance, a rigorous formulation of Statement 3 of Proposition 4.3 would be:

Additive independence on $L_c^\infty$:

For any $\mathsf{M} \in \mathcal{M}$ and independent $\tilde{X}, \tilde{Y} \in L_c^\infty(\Omega_{\mathsf{M}}, \mathcal{F}_{\mathsf{M}}, \mathbb{P}_{\mathsf{M}}; \mathbb{R})$, we have $\rho(\tilde{X} + \tilde{Y}) = \rho(\tilde{X}) + \rho(\tilde{Y})$.

### B.1. More details on resampled kernels

In this section, we provide more details on the resampled kernels setup.

**Value functions for sequences of kernels** For $\pi \in \Pi_{\mathsf{H}}$ and $s \in \mathcal{S}$, we naturally extend the definition of the value function for nominal MDPs (see Equation (2)) to the case of a non-random transition kernel sequence $(P_t)_{t \in \mathbb{N}}$ that changes over time. In this setup, the value function $V^{\pi, (P_t)_{t \in \mathbb{N}}} \in \mathbb{R}^{\mathcal{S}}$ depends on $\pi$ and $(P_t)_{t \in \mathbb{N}}$ and can be written as, for $s \in \mathcal{S}$,

$$V^{\pi, (P_t)_{t \in \mathbb{N}}}(s) = \mathbb{E}_s^{\pi, (P_t)_{t \in \mathbb{N}}} \Big( \sum_{t=0}^{\infty} \gamma^t r(\tilde{S}_t, \tilde{A}_t, \tilde{S}_{t+1}) \Big) \,.$$

where $\mathbb{E}_s^{\pi, (P_t)_{t \in \mathbb{N}}}$ is the expectation over the random trajectories $(\tilde{S}_t, \tilde{A}_t)_{t \in \mathbb{N}}$ starting in state $s$, with the distribution over trajectories induced by $\pi$ and $(P_t)_{t \in \mathbb{N}}$. In the paper, we choose to overload the notation $V^{\pi, \tilde{P}}$ when $\tilde{P} = (\tilde{P}_t)_{t \in \mathbb{N}}$ for the sake of simplicity (instead of writing $V^{\pi, (\tilde{P}_t)_{t \in \mathbb{N}}}$).

**Dynamic programming for non-i.i.d. kernels** We provide here a brief discussion of the case of resampled kernels with non-i.i.d. kernels. In particular, assume that $\tilde{P}_t \sim \nu_t$ for each period $t \in \mathbb{N}$ with each $\nu_t$ having a product structure, $\tilde{P}_t \perp\!\!\!\perp \tilde{P}'_t$ for each $t \neq t'$, but where the law $\nu_t$ may be different at each period. To recover a dynamic programming equation, one needs to index value functions with a time index $t \in \mathbb{N}$, a situation similar to the case of finite horizon MDPs (see Chapter 4 in Puterman (2014)). For distributionally robust MDPs, Xu & Mannor (2012), the case of non-i.i.d. resampled kernels is called *nonstationary model* and the case of i.i.d. resampled kernels is called *stationary model*, see beginning of Section 4 in Xu & Mannor (2012); the term *stationary/nonstationary* refers to the to the stationarity of the sequence $(\nu_t)_{t \geqslant 0}$ which is a stationary sequence of $\nu_t = \nu_{t'}, \forall\, t, t' \in \mathbb{N}$. A central result shown by Xu & Mannor (2012) is the case of **i.i.d. resampled kernels and non-i.i.d. resampled kernels are equivalent** in the sense that they yield the same optimal value functions and optimal policies (see Theorem 4.1 and Theorem 4.2 in Xu & Mannor (2012)). This diminishes the practical interest of the non-i.i.d. case, and this is one of the main reasons why we left a more formal analysis of this non-i.i.d. framework for future work.

# C. Connection with previous models

In this section, we provide rigorous reformulations of several of the common variants of MDPs as ambiguity-averse MDPs. We also provide a literature review, discussing the connection of ambiguity-averse MDPs with risk-averse MDPs.

## C.1. Reformulations and related work on MDPs with uncertain parameters

In this appendix we provide a rigorous reformulation of some of the classical variants of MDP models as ambiguity-averse MDPs. We summarize these reformulations in Table 1.

**Nominal MDPs** Consider a nominal MDP with transition probability $P \in \Delta(\mathcal{S})^{\mathcal{S} \times \mathcal{A}}$. We note that in the standard MDP model, the transition probabilities are fixed over time. Therefore, the transition kernel sequence is static. Consider a Dirac measure $\nu$ that puts all its mass at $P : \nu = \delta_P$ (we denote by $\delta_x$ the distribution that puts a mass of $1$ on an element $x$). Then $\rho(\mu^\top V^{\pi, \tilde{P}}) = \rho(\mu^\top V^{\pi, P})$. We additionally choose $\rho = \mathbb{E}^\nu$. Because $\mu^\top V^{\pi, P}$ is a constant, we get that $\rho(\mu^\top V^{\pi, P}) = \mu^\top V^{\pi, P}$ and we recover the standard objective for nominal MDPs. This shows the following proposition.

**Proposition C.1.** *Any nominal MDP with transition kernel $P$ can be reformulated as an ambiguity averse MDP with $\nu = \delta_P$ and $\rho(\tilde{X}) = \mathbb{E}^\nu(\tilde{X})$.*

**Robust MDPs and Optimistic MDPs** There are two main frameworks for robust MDPs (Iyengar, 2005; Nilim & El Ghaoui, 2005; Wiesemann et al., 2013). An uncertainty set $\mathcal{P} \subseteq \Delta(\mathcal{S})^{\mathcal{S} \times \mathcal{A}}$ is *s-rectangular* if it satisfies the following Cartesian product property: $\mathcal{P} = \times_{s \in \mathcal{S}} \mathcal{P}_s$ for some $\mathcal{P}_s \in \Delta(\mathcal{S})^{\mathcal{A}}$ for each state $s \in \mathcal{S}$. Note that if $\nu$ is a distribution over $\Delta(\mathcal{S})^{\mathcal{S} \times \mathcal{A}}$ with product structure, then its support $\text{supp}(\nu) \subseteq \Delta(\mathcal{S})^{\mathcal{S} \times \mathcal{A}}$ is a compact, s-rectangular uncertainty set.

We first focus on the model introduced in the seminal paper on s-rectangular robust MDPs (Wiesemann et al., 2013), which analyzes

$$\sup_{\pi \in \Pi_{\text{H}}} \inf_{P \in \mathcal{P}} \mu^\top V^{\pi, P} \tag{11}$$

for some convex compact s-rectangular uncertainty set $\mathcal{P}$. Note that here the transition probabilities $P$ are fixed over time, so that we are in the static kernel case as described in Definition 3.2. This model can be recovered by choosing a distribution $\nu$ supported on $\mathcal{P}$ for the distribution of the random variable $\tilde{P}$, and $\mathcal{P}$ such that $\text{supp}(\nu) = \mathcal{P}$, and $\rho(\tilde{X}) = \text{ess}\inf(\tilde{X})$. In this case, we indeed have, following Lemma A.2, that

$$\text{ess}\inf \mu^\top V^{\pi, \tilde{P}} = \inf_{P \in \text{supp}(\nu)} \mu^\top V^{\pi, P} = \inf_{P \in \mathcal{P}} \mu^\top V^{\pi, P}$$

so that we have shown the following proposition.

**Proposition C.2.** *Consider a robust MDP as in (11) with convex compact s-rectangular uncertainty set $\mathcal{P}$. We can reformulate any robust MDPs as in (11) with an ambiguity-averse MDP with product structure and static kernels with distribution $\nu$ supported on $\mathcal{P}$ and $\rho(\tilde{X}) = \text{ess}\inf(\tilde{X})$.*

A second popular model of RMDPs allows the transition kernel to change over time and focuses on solving

$$\sup_{\pi \in \Pi_{\text{H}}} \inf_{P \in \mathcal{P}^{\mathbb{N}}} \mu^\top V^{\pi, P} \tag{12}$$

where $\mathcal{P}^{\mathbb{N}}$ is the set of sequences of kernels: $\mathcal{P}^{\mathbb{N}} = \{(P_t)_{t \in \mathbb{N}} \mid P_t \in \mathcal{P}, \forall\, t \in \mathbb{N}\}$ for some uncertainty set $\mathcal{P} \subseteq \Delta(\mathcal{S})^{\mathcal{S} \times \mathcal{A}}$. Note that Proposition 2.2 in Grand-Clément et al. (2023) shows that (11) and (12) coincide (Proposition 2.2 in Grand-Clément et al. (2023) in fact shows a stronger statement for history-dependent transition probabilities, but these include as a special case Markovian transition probabilities, so we indeed have that the two optimization problems (11) and (12) coincide). We can reformulate (12) as an ambiguity-averse MDP with the same setup as for (11) but where we allow for resampled kernels. Indeed, writing $\nu^{\otimes \mathbb{N}}$ for the law of the i.i.d. sequence of kernels $(\tilde{P}_t)_{t \in \mathbb{N}}$ where $\tilde{P}_t \sim \nu$ for each period $t \in \mathbb{N}$, we get that $\mathrm{supp}(\nu^{\otimes \mathbb{N}}) = \mathrm{supp}(\nu)^{\mathbb{N}}$, so that we obtain that for resampled kernels

$$\mathrm{ess\,inf}\left(\mu^{\top} V^{\pi, \tilde{P}}\right) = \inf_{(P_t)_{t \in \mathbb{N}} \in \mathrm{supp}(\nu^{\otimes \mathbb{N}})} \mu^{\top} V^{\pi, P} = \inf_{(P_t)_{t \in \mathbb{N}} \in \mathrm{supp}(\nu)^{\mathbb{N}}} \mu^{\top} V^{\pi, P} = \inf_{(P_t)_{t \in \mathbb{N}} \in \mathcal{P}^{\mathbb{N}}} \mu^{\top} V^{\pi, P}$$

for $\mathcal{P} = \mathrm{supp}(\nu)$. We have shown the following proposition.

**Proposition C.3.** *Consider a robust MDP as in* (12) *with convex compact s-rectangular uncertainty set $\mathcal{P}$. We can reformulate any robust MDPs as in* (12) *with an ambiguity-averse MDP with product structure and resampled kernels with distribution $\nu$ supported on $\mathcal{P}$ for i.i.d. kernels and $\rho(\tilde{X}) = \mathrm{ess\,inf}(\tilde{X})$.*

We also note that robust MDPs with non-rectangular uncertainty sets have been studied in the literature, e.g. the r-rectangular model (Goyal & Grand-Clément, 2023) or model based on $\ell_1$-norm between the kernels (Kumar et al., 2025), but dynamic programming does not hold for these models *in all generality*, i.e. for all possible choices of instantaneous rewards. In particular, it is shown in Grand-Clément et al. (2024) that s-rectangularity (implied by the product structure for the distribution $\nu$ over kernels) is a necessary condition for dynamic programming to hold, and r-rectangular models can only satisfy weaker forms of dynamic programming (i.e. they require strong assumptions, e.g. that the rewards do not depend on the next state, see Grand-Clément et al. (2024)).

Finally, we note that the case of optimistic MDPs, which focus on $\sup_{\pi \in \Pi_{\mathsf{H}}} \sup_{P \in \mathcal{P}} \mu^{\top} V^{\pi, P}$ can be reformulated analogously as for RMDPs by choosing a distribution $\nu$ over the transition probabilities in $\Delta(\mathcal{S})^{\mathcal{S} \times \mathcal{A}}$ with $\mathrm{supp}(\nu) = \mathcal{P}$ and $\rho = \mathrm{ess\,sup}$. As noted in Iyengar (2005); Goh et al. (2018) and proved in Givan et al. (2000), the results for robust MDPs extend to optimistic MDPs and in particular Conditions 1 and 2 hold for $\rho = \mathrm{ess\,sup}$ in both the static and resampled kernel cases.

**Multi-model MDPs** Multi-model MDPs (MMDPs) consider an optimization problem of the form

$$\sup_{\pi \in \Pi_{\mathsf{H}}} \mathbb{E}^{\nu}(\mu^{\top} V^{\pi, \tilde{P}}). \tag{13}$$

This model has received many different names in the literature. For instance, Le Tallec (2007); Steimle et al. (2021) focus on this model in the case of *finite-horizon* MDPs, and Le Tallec (2007) also treats the case of discounted return (*infinite-horizon* MDPs). The multi-model MDP framework has been reintroduced and studied independently over the years, and is also called *soft-robust MDPs* (Derman et al., 2018; Lobo et al., 2020), *concurrent MDPs* (Buchholz & Scheftelowitsch, 2019; 2020), and *Multiple-Environment MDPs (MEMDPs)* (Raskin & Sankur, 2014). More precisely, MEMDPs with prior semantics draw the true environment once from a fixed prior and are therefore equivalent to static MMDPs with $\rho = \mathbb{E}^{\nu}$ (Chatterjee et al., 2020; Bordais & Raskin, 2026). The adversarial semantics of MEMDPs instead ask for guarantees against every possible environment; this is the same as a static robust MDP with a finite, generally non-convex uncertainty set. The optimization problem (13) is known to be NP-hard for static kernels, even in the case of supports of cardinality two, e.g. Le Tallec (2007) proved that the following problem is NP-hard

$$\sup_{\pi \in \Pi_{\mathsf{H}}} \frac{1}{2} \mu^{\top} V^{\pi, P_1} + \frac{1}{2} \mu^{\top} V^{\pi, P_2}.$$

The same result for discounted return (infinite-horizon MDPs) is present in (Buchholz & Scheftelowitsch, 2019). The following reformulation of MMDPs as ambiguity-averse MDPs is straightforward.

**Proposition C.4.** *Consider a multi-model MDP as in* (13)*, with static or resampled kernels. Then we can model it as an ambiguity-averse MDP $\mathsf{M}_{\nu}$ with the distribution $\nu$ using $\rho(\tilde{X}) = \mathbb{E}^{\nu}\left(\tilde{X}\right)$.*

We note that multi-model MDPs also arise as a special case of distributionally robust MDPs (see Appendix B.1 and Equation 15 below) where there is only a single possible distribution $\nu$ (and not a full set of possible probability distributions

$\mathcal{V}$). Given the seminal results for distributionally robust MDPs as in Section 4 in Xu & Mannor (2012), we know that multi-model MDPs with *resampled* kernels satisfy Conditions 1 and 2. It is known that multi-model MDPs with *static* kernels do not satisfy Conditions 1 and 2, and for the sake of completeness, we reprove this in Example 3.8.

**Percentile optimization**  Percentile optimization (Mannor et al., 2004; Delage & Mannor, 2010; Petrik & Russel, 2019; Zhang et al., 2024) optimizes for the Value-at-Risk (VaR) at a certain risk level $\alpha \in (0, 1)$, i.e. it optimizes

$$\sup_{\pi \in \Pi_H} \mathsf{VaR}_\alpha \left( \mu^\top V^{\pi, \tilde{P}} \right). \tag{14}$$

where $\tilde{P}$ is random and follows a distribution $\nu$ (typically assumed uniform). Note that it is NP-hard to compute an optimal policy for general distributions (corollary 1 in Delage & Mannor (2010)). The following reformulation is straightforward.

**Proposition C.5.** *Consider a percentile optimization problem as in* (14). *Then it can be formulated as an ambiguity-averse MDP with $\rho(\tilde{X}) = \mathsf{VaR}_\alpha(\tilde{X})$ and a distribution $\nu$ for static kernels.*

**Other recent work**  The authors in Lobo et al. (2020) consider optimizing for a convex combination of expectation and the Conditional Value-at-Risk (CVaR), i.e. optimizing $\pi \mapsto (1 - \lambda)\mathbb{E}(\mu^\top V^{\pi, \tilde{P}}) + \lambda\mathsf{CVaR}(\mu^\top V^{\pi, \tilde{P}})$ for some $\lambda \in [0, 1]$. This optimization problem can be cast as an ambiguity-averse MDP using the risk measure $\rho(\tilde{X}) = (1 - \lambda)\mathbb{E}(\tilde{X}) + \lambda\mathsf{CVaR}(\tilde{X})$.

## C.2. Other related work

**Robust MDPs with uncertain rewards**  An alternative robust MDP model considers that the rewards $r \in \mathbb{R}^{\mathcal{S} \times \mathcal{A} \times \mathcal{S}}$ are uncertain while the transitions are known and fixed, or that both rewards and transitions are uncertain, e.g. Givan et al. (2000); Eysenbach & Levine (2019); Brekelmans et al. (2022); Ashlag et al. (2026); it is also possible to analyze distributionally robust MDPs with both uncertain rewards and transitions, as in the seminal papers (Xu & Mannor, 2012). RMDPs with uncertain rewards represent a simpler optimization problem compared to RMDPs with uncertain transitions, because value functions are linear in the rewards (this follows from the linearity of the expectation operator) but depend in a non-linear fashion on the transition probabilities. In our paper, we decide to focus solely on the case of uncertain kernels for the sake of conciseness, noting that our framework could be extended to also model uncertain rewards, at the cost of heavier notations.

**Distributionally robust MDPs**  DRMDPs are introduced in Xu & Mannor (2012) as the following optimization problem:

$$\sup_{\pi \in \Pi_H} \inf_{\nu \in \mathcal{V}} \mathbb{E}^\nu(\mu^\top V^{\pi, \tilde{P}}) \tag{15}$$

where $\mathcal{V}$ is a set of probability distribution over the sequences of transition probabilities $(P_t)_{t \in \mathbb{N}}$. For ambiguity-averse MDPs to cover this setting, we would need to define them with sets of distributions $\mathcal{V}$ instead of a single distribution $\nu$. Note that DRMDPs also consider the case of uncertain transition kernels that can change over time in a non-i.i.d. fashion, so that we would need a different set $\mathcal{V}_t$ of possible distributions $\nu_t$ for the transition probabilities at each period $t$. This complicates the formulation of the optimization problem (6), which requires to define risk measures with arguments to have multiple distributions (one per choice of $\nu \in \mathcal{V}$). In principle, this can be done with the notion of an ambiguous probability space (see Delage et al. (2019)), but it appears beyond the scope of this paper.

**Necessary and sufficient conditions for dynamic programming**  There are numerous papers obtaining necessary and sufficient conditions for the existence of dynamic programming in the context of MDPs. The authors in Grand-Clément et al. (2024) focus on robust MDPs with an uncertainty set $\mathcal{P}$, and show that a dynamic programming equation holds for policy evaluation if and only if $\mathcal{P}$ is an s-rectangular uncertainty set. The authors in Marthe et al. (2023) focus on risk-averse MDPs with a finite horizon (in the framework of distributional reinforcement learning) and show that for a Bellman optimality equation to hold it is necessary that the risk measure is $\mathsf{ERM}_\beta$ for some $\beta \in \mathbb{R}$; the sufficiency is proved in Howard & Matheson (1972). The authors in Rowland et al. (2019) focus on distributional reinforcement learning and define *Bellman closedness* for a set of statistics, as the property that these statistics can be jointly optimized by dynamic programming (e.g., this is the case for the set consisting of the expectation and the variance (Sobel, 1982)). The authors in Rowland et al. (2019) show that the only sets of Bellman closed statistics are sets of moments (see Theorem 4.3 in Rowland et al. (2019)).

**Risk-averse MDPs** In risk-averse MDPs, the transition probabilities $P$ and the rewards $r$ are known, and the risk measures are with respect to the distribution over the trajectories induced by pairs of policies and transition kernels, i.e., risk-averse MDPs aim at solving

$$\sup_{\pi \in \Pi_\mathsf{H}} \rho \left( \sum_{t=0}^{+\infty} \gamma^t r(\tilde{S}_t, \tilde{A}_t, \tilde{S}_{t+1}) \right) . \tag{16}$$

It is also common to focus on the case of finite-horizon MDPs, i.e., to optimize, for a given horizon $T \in \mathbb{N}$, the following objective:

$$\sup_{\pi \in \Pi_\mathsf{H}} \rho \left( \sum_{t=0}^{H} r(\tilde{S}_t, \tilde{A}_t, \tilde{S}_{t+1}) \right) . \tag{17}$$

The existence of dynamic programming equations in risk-averse MDPs has been studied extensively, often using the terms "dynamic consistency" or "time consistency" for risk measures. Informally, a risk measure is time consistent when the risk preferences do not change throughout the execution (Roorda et al., 2005; Iancu et al., 2015). Time consistency is typically a weaker requirement than our Conditions 1 and 2 because it is sufficient that it holds only for $\gamma = 1$.

Several works from the risk-averse MDP literature study necessary and sufficient conditions for the time-consistency of risk measures. Kupper & Schachermayer (2009) shows that only law-invariant, time-consistent, and relevant risk measures are the entropic risk measure, which subsumes the expectation operator. Relevance is a technical condition that excludes risk measures that are insensitive to the $\mathrm{ess\,inf}$ of the random variable. Similar results have been derived in later work, including the restriction for law-invariant coherent risk measures (Shapiro, 2012), and for law-invariant and $W^1$-continuous (see Appendix H.3) risk measures (Marthe et al., 2023).

Despite some superficial similarity to our results, the literature on the time consistency of risk-averse MDPs does not apply to ambiguity-averse MDPs. There is a fundamental difference in the setup of the two problems. In risk-averse MDPs, the risk aversion with respect to randomness is in the returns of the uncertain histories (or trajectories). As a result, we derive that the properties of ambiguity-averse MDPs stem from the interaction between the risk measure, the expectation operator, and the discount factor.

Interestingly, one can use dynamic programming to compute policies for risk-averse MDPs using one of the following two approaches:

- *Dropping the law-invariance condition: nested risk measures.* If one drops the law-invariance requirement for the risk measure, it is possible to recover a dynamic programming equation over the set of states using nested risk measures (Ruszczyński, 2010) (also called Markov or iterated risk measures). Intuitively, nested risk measure replacing the objective $\rho \left( \sum_{t=0}^{+\infty} \gamma^t r(\tilde{S}_t, \tilde{A}_t, \tilde{S}_{t+1}) \right)$ by applying the risk measure $\rho$ repeatedly, i.e., informally it corresponds to optimizing

$$\rho \left( r(\tilde{S}_0, \tilde{A}_0, \tilde{S}_1) + \rho \left( \gamma r(\tilde{S}_1, \tilde{A}_1, \tilde{S}_1) + \rho \left( \gamma r(\tilde{S}_2, \tilde{A}_2, \tilde{S}_2) + .... \right) \right) \right) . \tag{18}$$

  From the above equation, one can construct a dynamic programming equation and derive value iteration algorithms (Ruszczyński, 2010) to compute optimal Markov policies. However, nested risk measures are not law-invariant, and their values can depend on the order in which rewards are accrued, complicating their interpretation. That is, the quantity in (18) differs from the original objective $\rho \left( \sum_{t=0}^{+\infty} \gamma^t r(\tilde{S}_t, \tilde{A}_t, \tilde{S}_{t+1}) \right)$ (Iancu et al., 2015).

- *Extended state space.* For most common risk measures, including VaR, CVaR, EVaR, ERM, one can formulate dynamic programming equations after augmenting the state space with an additional variable. The particular augmentation depends on the risk measure. For example, to optimize discounted risk-averse objectives with ERM and EVaR, one can augment the state space with the current time step, yielding Markov-optimal policies (Hau et al., 2023b). When the objective is VaR, one can construct a dynamic program that augments the state space with the risk level $\alpha \in [0, 1]$ (Li et al., 2022a; Hau et al., 2023a; 2025), resulting in a history-dependent policy. When the objective is a CVaR, one can construct a dynamic program by augmenting the state space with a real variable that keeps track of the accumulated rewards (Bäuerle & Ott, 2011) and compute an optimal history-dependent policy. When *evaluating* a fixed policy with the CVaR objective, one can construct a dynamic programming that augments the state with the risk level $\alpha \in [0, 1]$, similarly to VaR. However, the risk-level augmentation fails for CVaR when *optimizing* a policy (Hau et al., 2023a; Godbout & Durand, 2025).

**Bayes-adaptive MDPs** Epistemic uncertainty over MDPs parameters has been studied extensively in the context of Bayes-adaptive MDPs (Guez et al., 2012). Most research on Bayes-adaptive MDPs has focused on reinforcement learning algorithms that compute history-dependent policies for the risk-neutral objective. For the discussion of the differences with the risk-averse setting, we refer the interested reader to Lin et al. (2022, section 2). While classical Bayesian MDPs often couple parameter uncertainty with trajectory observations through posterior updates (Lin et al., 2022; Shapiro et al., 2025), the ambiguity-averse objective first computes the nominal expected return for every kernel (or kernel sequence) and then applies the risk measure across possible kernels. This is why policies in our framework can differ qualitatively from Bayesian policies that average or learn parameter estimates from realized trajectories.

## D. Proof of Proposition 3.7

*Proof of Proposition 3.7.* 1. Let $v, w \in \mathbb{R}^{\mathcal{S}}$ with $v \leqslant w$. Since $T^{\pi,P}$ is a monotone operator for any realization of the transition probabilities $P$, that $T^{\pi,\tilde{P}}v \leqslant T^{\pi,\tilde{P}}w$ almost surely. Since $\rho$ is a monotone risk measure, we obtain that

$$\rho\left(T^{\pi,\tilde{P}}v\right) \leqslant \rho\left(T^{\pi,\tilde{P}}w\right), \tag{19}$$

which proves that $T^{\pi,\nu,\rho}$ is a monotone operator for each $\pi \in \Pi_{\mathsf{S}}$. The fact that $T^{\nu,\rho}$ is a monotone operator follows directly from taking the supremum over $\pi \in \Pi_{\mathsf{S}}$ on both sides of (19).

2. Let $v \in \mathbb{R}^{\mathcal{S}}, c \in \mathbb{R}$. We have

$$T^{\pi,\nu,\rho}(v + c \cdot 1_{\mathcal{S}}) = \rho(T^{\pi,\tilde{P}}(v + c \cdot 1_{\mathcal{S}})) = \rho(T^{\pi,\tilde{P}}v + c \cdot 1_{\mathcal{S}}) = \rho(T^{\pi,\tilde{P}}v) + c \cdot 1_{\mathcal{S}} = T^{\pi,\nu,\rho}v + c \cdot 1_{\mathcal{S}},$$

where the first equality is by definition of the operator $T^{\pi,\nu,\rho}$, the second equality is because $T^{\pi,\tilde{P}}$ is translation-invariant for any realization of $\tilde{P}$, the third equality is from the translation-invariance of $\rho$, and the last equality is again by definition of ours operators. This proves that $T^{\pi,\nu,\rho}$ is translation-invariant. The proof for $T^{\nu,\rho}$ follows the same line, using additionally that $\max_{s \in \mathcal{S}}\{w(s) + c\} = \max_{s \in \mathcal{S}}\{w(s)\} + c$ for any $w \in \mathbb{R}^{\mathcal{S}}$ and $c \in \mathbb{R}$.

3. A direct calculation shows that the composition of a non-expansive risk measure and a contraction remains a contraction. Lemma 4.3 in Föllmer & Schied (2016) shows that a monotone, cash-invariant risk measure is non-expansive (this is also a consequence of Crandall-Tartar theorem (Crandall & Tartar, 1980)), in the sense that

$$|\rho(\tilde{X}) - \rho(\tilde{Y})| \leqslant \operatorname{ess\,sup} |\tilde{X} - \tilde{Y}|, \quad \forall \tilde{X}, \tilde{Y}. \tag{20}$$

Note that in (Föllmer & Schied, 2016) this result is written an $\ell_\infty$-norm for random variables, but for the sake of clarity we choose to use the notation $\|\cdot\|_\infty$ only for functions in $\mathbb{R}^{\mathcal{S}}$ in this paper. For the sake of completeness, we provide a concise proof of (20) here. Let $\delta = \operatorname{ess\,sup} |\tilde{X} - \tilde{Y}|$. By definition, $|\tilde{X} - \tilde{Y}| \leqslant \delta$ a.s. so that $\tilde{Y} - \delta \leqslant \tilde{X} \leqslant \tilde{Y} + \delta$ a.s.. Using the monotonicity of $\rho$ we get that $\rho(\tilde{Y} - \delta) \leqslant \rho(\tilde{X}) \leqslant \rho(\tilde{Y} + \delta)$. Using translation invariance we get that $\rho(\tilde{Y}) - \delta \leqslant \rho(\tilde{X}) \leqslant \rho(\tilde{Y}) + \delta$ so that $|\rho(\tilde{X}) - \rho(\tilde{Y})| \leqslant \delta$, which concludes the proof of (20).

Now we have

$$\begin{aligned}
\|T^{\pi,\nu,\rho}v - T^{\pi,\nu,\rho}w\|_\infty &= \max_{s \in \mathcal{S}} |\rho(T^{\pi,\tilde{P}}v(s)) - \rho(T^{\pi,\tilde{P}}w(s))| \\
&\leqslant \max_{s \in \mathcal{S}} \operatorname{ess\,sup} |T^{\pi,\tilde{P}}v(s) - T^{\pi,\tilde{P}}w(s)| \\
&\leqslant \max_{s \in \mathcal{S}} \gamma \|v - w\|_\infty \\
&\leqslant \gamma \|v - w\|_\infty
\end{aligned}$$

where the first line is from the definition of $T^{\pi,\nu,\rho}$ and of the $\ell_\infty$-norm for vectors, the second line is from the non-expansiveness of $\rho$ as in (20), and the third line is from the fact for each kernel $P$ the operator $T^{\pi,P}$ (for nominal MDPs) is a contraction. Therefore, we have shown

$$\forall v, w \in \mathbb{R}^{\mathcal{S}}, \|T^{\pi,\nu,\rho}v - T^{\pi,\nu,\rho}w\|_\infty \leqslant \gamma \|v - w\|_\infty. \tag{21}$$

The contraction property of $T^{\nu,\rho}$ follows directly from taking the supremum over $\pi \in \Pi_{\mathsf{S}}$ in the right-hand side of (21).

We next prove the attainability property.

We show that for any $v \in \mathbb{R}^{\mathcal{S}}$, we can find $\pi \in \Pi_{\mathsf{S}}$ such that $T^{\nu,\rho}v = T^{\pi,\nu,\rho}v$, i.e. such that $\sup_{\pi \in \Pi_{\mathsf{S}}} T^{\pi,\nu,\rho}v(s) = \max_{\pi \in \Pi_{\mathsf{S}}} T^{\pi,\nu,\rho}v(s)$ for each $s \in \mathcal{S}$. To prove this, it suffices to show that $\pi \to T^{\pi,\nu,\rho}v(s)$ is continuous for $\pi \in \Pi_{\mathsf{S}}$ (note that $\Pi_{\mathsf{S}} = \Delta(\mathcal{A})^{\mathcal{S}}$ is a compact set and that $T^{\pi,\nu,\rho}v(s)$ only depends on the s-th component of $\pi$). To prove that $\pi \to T^{\pi,\nu,\rho}v(s)$ is a continuous function, we use the definition of the Bellman operator: $T^{\pi,\nu,\rho}v(s) = \rho(T^{\pi,\tilde{P}}v(s))$. Let $\pi \in \Pi_{\mathsf{S}}$ and consider a sequence $(\pi_n)_{n \in \mathbb{N}}$ of stationary policies such that $\lim_{n \to +\infty} \pi_n = \pi$. Then

$$|T^{\pi_n,\nu,\rho}v(s) - T^{\pi,\nu,\rho}v(s)| = |\rho(T^{\pi_n,\tilde{P}}v(s)) - \rho(T^{\pi,\tilde{P}}v(s))|$$
$$\leqslant \operatorname{ess\,sup} |T^{\pi_n,\tilde{P}}v(s) - T^{\pi,\tilde{P}}v(s)|$$

where the first equality uses the definition of the Bellman operator and the inequality uses the non-expansivity of $\rho$ (Equation (20), similarly as in the proof for the contraction property of the operators in the third point above). Now from the definition of the operator $T^{\pi,\tilde{P}}$ we get that for any $P \in \Delta(\mathcal{S})^{\mathcal{S} \times \mathcal{A}}$ we have

$$|T^{\pi_n,\tilde{P}}v(s) - T^{\pi,\tilde{P}}v(s)| \leqslant C \cdot \|\pi_n - \pi\|_1$$

where $C := \max_{s,a,s'} |r(s,a,s') + \gamma \max_{s \in \mathcal{S}} |v(s)|$. Overall we obtained that

$$|T^{\pi_n,\nu,\rho}v(s) - T^{\pi,\nu,\rho}v(s)| \leqslant C \cdot \|\pi_n - \pi\|_1$$

and it is therefore straightforward to conclude that $\lim_{n \to +\infty} |T^{\pi_n,\nu,\rho}v(s) - T^{\pi,\nu,\rho}v(s)| = 0$ when $\pi_n \to \pi$, i.e. we have shown that $\pi \mapsto T^{\pi,\nu,\rho}v(s)$ is a continuous function on the compact set $\Pi_{\mathsf{S}}$. By Weierstrass theorem, we conclude that $\sup_{\pi \in \Pi_{\mathsf{S}}} T^{\pi,\nu,\rho}v(s) = \max_{\pi \in \Pi_{\mathsf{S}}} T^{\pi,\nu,\rho}v(s)$. $\qquad\square$

## E. More details on Example 3.4-3.8

We provide a very simple example where Conditions 1 and 2 fail to hold below.

**Example E.1.** We consider, for $\rho = \mathbb{E}^{\nu}$, the static ambiguity-averse MDP of Figure 2 where $\tilde{X} \sim \text{Unif}([0,1])$.

*Figure 2.* Ambiguity-averse MDP with $\mathcal{S} = \{\mathsf{Start}, \mathsf{End}\}$, $\mathcal{A} = \{1\}$. The edges $(s, a, s')$ are labeled with pairs $(P(s,a,s'), r(s,a,s'))$.

Explicitly, we have $\mathcal{S} = \{\mathsf{Start}, \mathsf{End}\}$, $\mathcal{A} = \{1\}$, discount factor $\gamma \in [0,1)$ and reward function $r(\mathsf{Start}, 1, s') = 1$ and $r(\mathsf{End}, 1, s') = 0$ for any $s' \in \mathcal{S}$. Here, $\nu$ is the distribution of the random matrix $\begin{pmatrix} \tilde{X} & 1 - \tilde{X} \\ 0 & 1 \end{pmatrix}$. We consider here the case of static kernels: there is a single sample of $\tilde{X}$ at time $t = 0$, so that the kernel $\tilde{P}_t$ at time $t$ is

$$\tilde{P}_t = \begin{pmatrix} \tilde{X} & 1 - \tilde{X} \\ 0 & 1 \end{pmatrix}, \forall t \in \mathbb{N}\,.$$

Explicitly, at any time step, from state $\mathsf{Start}$, the system transitions to state $\mathsf{Start}$ with probability $\tilde{X}$ and to state $\mathsf{End}$ with probability $1 - \tilde{X}$, and from state $\mathsf{End}$, the system stays in state $\mathsf{End}$ with probability 1.

Because the transitions are only random at $\mathsf{Start}$, the transition kernel law $\nu$ of this ambiguity-averse MDP has a product structure over spaces and identical marginals over time (because static), and because the uniform distribution has convex support, the overall transition kernel law has convex support.

We consider the unique stationary policy $\pi$ that always takes the single action possible. The random nominal value function under policy $\pi$ and transition kernel sequence $\tilde{P}$ is given by:

$$V^{\pi,\tilde{P}} = \begin{pmatrix} \sum_{t \geqslant 0} \gamma^t \tilde{X}^t \\ 0 \end{pmatrix} = \begin{pmatrix} \frac{1}{1 - \gamma \tilde{X}} \\ 0 \end{pmatrix}.$$

Therefore, the ambiguity-averse value function under policy $\pi$ is:

$$V^{\pi,\nu,\mathbb{E}^\nu} = \begin{pmatrix} \mathbb{E}\left(\frac{1}{1-\gamma\tilde{X}}\right) \\ 0 \end{pmatrix} = \begin{pmatrix} \frac{-\log(1-\gamma)}{\gamma} \\ 0 \end{pmatrix}.$$

On the other hand, the ambiguity-averse Bellman operator $T^{\pi,\nu,\mathbb{E}^\nu}$ is given by:

$$\forall v \in \mathbb{R}^2, \qquad T^{\pi,\nu,\mathbb{E}^\nu}\begin{pmatrix} v(\mathsf{Start}) \\ v(\mathsf{End}) \end{pmatrix} = \begin{pmatrix} \mathbb{E}\left(1 + \tilde{X}\gamma v(\mathsf{Start}) + (1-\tilde{X})\gamma v(\mathsf{End})\right) \\ \mathbb{E}\left(0 \cdot \gamma v(\mathsf{Start}) + 1 \cdot \gamma v(\mathsf{End})\right) \end{pmatrix} = \begin{pmatrix} 1 + \frac{\gamma}{2} v(\mathsf{Start}) + \frac{\gamma}{2} v(\mathsf{End}) \\ \gamma v(\mathsf{End}) \end{pmatrix},$$

and in particular, we have

$$T^{\pi,\nu,\mathbb{E}^\nu} V^{\pi,\nu,\mathbb{E}^\nu} = \begin{pmatrix} 1 - \frac{\log(1-\gamma)}{2} \\ 0 \end{pmatrix}.$$

Finally, we have $T^{\pi,\nu,\mathbb{E}^\nu} V^{\pi,\nu,\mathbb{E}^\nu} = V^{\pi,\nu,\rho} \iff -\log(1-\gamma) = \frac{2\gamma}{2-\gamma}$, so because the second assertion does not hold for every $\gamma \in [0,1)$, there exists $\mathsf{M} \in \mathcal{M}_c$ and $\pi \in \Pi_S$ such that $T^{\pi,\nu,\mathbb{E}^\nu} V^{\pi,\nu,\mathbb{E}^\nu} \neq V^{\pi,\nu,\mathbb{E}^\nu}$. That is, the expectation does not satisfy Condition 1 for static kernels.

Because there is a single policy, $T^{\pi,\nu,\mathbb{E}^\nu} = T^{\nu,\mathbb{E}^\nu}$ and $V^{\pi,\nu,\mathbb{E}^\nu} = V^{\nu,\mathbb{E}^\nu}$, so we also have that the expectation doesn't satisfy Condition 2 for static kernels.

# F. Proof of Theorem 4.1

*Proof of Theorem 4.1.* Under the assumption that $\rho$ is monotone and translation invariant, from Proposition 3.7, both $T^{\pi,\nu,\rho}$ (for each $\pi \in \Pi_S$) and $T^{\nu,\rho}$ are monotone contractions.

We first prove that there exists $\pi^\star \in \Pi_S$ such that $V^{\nu,\rho} = V^{\pi^\star,\nu,\rho}$. Indeed, for $s \in \mathcal{S}$ we have

$$V^{\nu,\rho}(s) = T^{\nu,\rho} V^{\nu,\rho} = \max_{\pi \in \Pi_S} T^{\pi,\nu,\rho} V^{\nu,\rho}(s) = T^{\pi^\star,\nu,\rho} V^{\nu,\rho}(s),$$

where the first equality is from Condition 2, the second equality follows from the definition of the operator $T^{\nu,\rho}$ as in (8), and the last equation is from defining the policy $\pi^\star \in \Pi_S$ attaining the $\arg\max$ in $\max_{\pi \in \Pi_S} T^{\pi,\nu,\rho} V^{\nu,\rho}(s)$ for each $s \in \mathcal{S}$ (note that $\pi^\star$ exists from the attainability property stated in Proposition 3.7) and from the definition of the operator $T^{\pi,\nu,\rho}$ as in (7). This shows that $V^{\nu,\rho}$ is a fixed point of the contraction operator $T^{\pi^\star,\nu,\rho}$. Therefore $V^{\nu,\rho}$ coincides with the fixed point of $T^{\pi^\star,\nu,\rho}$, which is equal to $V^{\pi^\star,\nu,\rho}$ from Condition 1.

We have

$$\sup_{\pi \in \Pi_H} \rho\left(V^{\pi,\tilde{P}}(s)\right) = V^{\nu,\rho}(s) = V^{\pi^\star,\nu,\rho}(s) \leqslant \max_{\pi \in \Pi_S} \rho\left(V^{\pi,\tilde{P}}(s)\right),$$

where the first equality is by definition of $V^{\nu,\rho}$, the second equality was proved in the previous paragraph, and the last equality is by taking the maximum over $\Pi_S$ (since $\pi^\star \in \Pi_S$). Since $\Pi_S \subseteq \Pi_H$, in general we also always have

$$\max_{\pi \in \Pi_S} \rho\left(V^{\pi,\tilde{P}}(s)\right) \leqslant \sup_{\pi \in \Pi_H} \rho\left(V^{\pi,\tilde{P}}(s)\right),$$

from which we conclude that

$$\sup_{\pi \in \Pi_H} \rho\left(V^{\pi,\tilde{P}}(s)\right) = \max_{\pi \in \Pi_S} \rho\left(V^{\pi,\tilde{P}}(s)\right) = \rho\left(V^{\pi^\star,\tilde{P}}(s)\right),$$

i.e., we conclude that $\pi^\star$ is a stationary optimal policy starting from any state.

The fact that $T^{\nu,\rho}$ is a contraction follows from Proposition 3.7. Additionally, we have defined $\pi^\star$ as any policy $\pi \in \Pi_S$ such that $T^{\nu,\rho} V^{\nu,\rho} = T^{\pi,\nu,\rho} V^{\nu,\rho}$, which concludes the proof. $\qquad\square$

The next corollary shows that when Conditions 1 and 2 hold for *both* static kernels and resampled kernels, then one can choose an optimal policy that is stationary and optimal for both models of kernels. This is stated in the next corollary.

**Corollary F.1.** *Assume that $\rho$ is monotone, translation-invariant. Assume that for static* and *resampled kernels, Conditions 1 and 2 hold.*

*Then there exists a* stationary *policy that is optimal starting from any state in both the model with static kernels and resampled kernels.*

*Proof of Corollary F.1.* Note that under the assumptions of Corollary F.1, the operator $T^{\nu,\rho}$ is a contraction, hence it has a unique fixed point. Additionally, this operator only depends on the marginal at time $t = 0$ (in evaluating $T^{\nu,\rho}v$ for some $v \in \mathbb{R}^{\mathcal{S}}$, the transition kernels are sampled only once before a continuation value of $v(s')$ is obtained at state $s' \in \mathcal{S}$), so that the operator $T^{\nu,\rho}$ is the same function in the case of static kernels and in the case of resampled kernels. Because Condition 2 holds for both the static and resampled kernels, the optimal value functions for both models of kernels are fixed point of the contraction operator $T^{\nu,\rho}$. Since a contraction has a unique fixed point, we get that the optimal value functions for both kernel models coincide, and we denote this common optimal value function as $V^{\star}$. From theorem 4.1 we know that in both kernel models (static or resampled) we can find a stationary optimal policy as any $\pi \in \Pi_{\mathsf{S}}$ such that $T^{\nu,\rho}V^{\star} = T^{\pi,\nu,\rho}V^{\star}$, which concludes the proof. $\qquad\square$

# G. Proof of Proposition 4.2

The proof of Proposition 4.2 follows from standard arguments in the MDP literature, e.g., Chapter 6 in Puterman (2014). The arguments rely entirely on the monotonicity, attainability, and contraction properties of the Bellman operators.

*Proof of Proposition 4.2.* 1. We start by proving the results for value iteration.

Let $V^n = T^{\nu,\rho}V^{n-1}$ for $n \geqslant 1$ and $V^0 \in \mathbb{R}^{\mathcal{S}}$. By induction we get that $V^n = (T^{\nu,\rho})^n V^0$ for any $n \geqslant 0$, and since $T^{\nu,\rho}$ is a contraction, we have that $(T^{\nu,\rho})^n V^0$ converges to the fixed point of the operator $T^{\nu,\rho}$ as $n \to +\infty$, i.e. we get that $\lim_{n \to +\infty} (T^{\nu,\rho})^n V^0 = V^{\nu,\rho}$. Moreover,

$$\|V^{\nu,\rho} - V^n\|_\infty = \|T^{\nu,\rho}V^{\nu,\rho} - T^{\nu,\rho}V^{n-1}\|_\infty \leqslant \gamma\|V^{\nu,\rho} - V^{n-1}\|_\infty,$$

where the equality follows by definition of the sequence $(V^n)_{n \in \mathbb{N}}$ and because $V^{\nu,\rho}$ is a fixed point of $T^{\nu,\rho}$ and the inequality is because $T^{\nu,\rho}$ is a contraction under the assumption of Proposition 4.2. We conclude by induction that $\|V^{\nu,\rho} - V^n\|_\infty \leqslant \gamma^n\|V^{\nu,\rho} - V^0\|_\infty$ for any $n \in \mathbb{N}$.

There remains to bound $\|V^{\pi^n,\nu,\rho} - V^{\nu,\rho}\|_\infty$ based on our bound on $\|V^{\nu,\rho} - V^n\|_\infty$. Note that

$$
\begin{aligned}
\|V^{\nu,\rho} - V^{\pi^n,\nu,\rho}\|_\infty &= \|V^{\nu,\rho} - T^{\pi_n,\nu,\rho}V^{n-1} + T^{\pi_n,\nu,\rho}V^{n-1} - V^{\pi^n,\nu,\rho}\|_\infty \\
&= \|V^{\nu,\rho} - T^{\nu,\rho}V^{n-1} + T^{\pi_n,\nu,\rho}V^{n-1} - V^{\pi^n,\nu,\rho}\|_\infty \\
&\leqslant \|V^{\nu,\rho} - T^{\nu,\rho}V^{n-1}\|_\infty + \|T^{\pi_n,\nu,\rho}V^{n-1} - V^{\pi^n,\nu,\rho}\|_\infty \\
&= \|T^{\nu,\rho}V^{\nu,\rho} - T^{\nu,\rho}V^{n-1}\|_\infty + \|T^{\pi_n,\nu,\rho}V^{n-1} - T^{\pi_n,\nu,\rho}V^{\pi^n,\nu,\rho}\|_\infty \\
&\leqslant \gamma\|V^{\nu,\rho} - V^{n-1}\|_\infty + \gamma\|V^{n-1} - V^{\pi^n,\nu,\rho}\|_\infty
\end{aligned}
$$

where the first equality is by adding and substracting the vector $T^{\pi_n,\nu,\rho}V^{n-1}$, the second equality is by definition of $\pi^n$ for value iteration, the first inequality is by triangle inequality, the third equality is by $V^{\nu,\rho}$ and $V^{\pi^n,\nu,\rho}$ being the fixed points of $T^{\nu,\rho}$ and $T^{\pi^n,\nu,\rho}$ following Conditions 1 and 2, and the last inequality is from these operators being contractions. Now we also have

$$\|V^{n-1} - V^{\pi^n,\nu,\rho}\|_\infty = \|V^{n-1} - V^{\nu,\rho} + V^{\nu,\rho} - V^{\pi^n,\nu,\rho}\|_\infty \leqslant \|V^{n-1} - V^{\nu,\rho}\|_\infty + \|V^{\nu,\rho} - V^{\pi^n,\nu,\rho}\|_\infty$$

where the equality is by adding and substracting $V^{\nu,\rho}$ and the inequality follows the triangle inequality. Overall we have shown

$$\|V^{\nu,\rho} - V^{\pi^n,\nu,\rho}\|_\infty \leqslant \gamma\|V^{\nu,\rho} - V^{n-1}\|_\infty + \gamma(\|V^{n-1} - V^{\nu,\rho}\|_\infty + \|V^{\nu,\rho} - V^{\pi^n,\nu,\rho}\|_\infty)$$

which we can rearrange to obtain

$$\|V^{\nu,\rho} - V^{\pi^n,\nu,\rho}\|_\infty \leqslant \frac{2\gamma}{1-\gamma}\|V^{n-1} - V^{\nu,\rho}\|_\infty \ .$$

We therefore conclude that $\|V^{\pi^n,\nu,\rho} - V^{\nu,\rho}\|_\infty \leqslant \frac{2\gamma^n}{1-\gamma}\|V^{\pi^0,\nu,\rho} - V^{\nu,\rho}\|_\infty$.

2. Let $n \in \mathbb{N}$. We first show that $V^{\pi^n,\nu,\rho} \leqslant V^{\pi^{n+1},\nu,\rho}$. We have

$$T^{\pi^{n+1},\nu,\rho} V^{\pi^n,\nu,\rho} = T^{\nu,\rho} V^{\pi^{n+1},\nu,\rho} \geqslant T^{\pi^n,\nu,\rho} V^{\pi^n,\nu,\rho} = V^{\pi^n,\nu,\rho}$$

where the first equality is by definition of $\pi^{n+1}$, where the inequality is from the definition of the Bellman optimality operator $T^{\nu,\rho}$, and the second equality is from Condition 1. Therefore we obtain $T^{\pi^{n+1},\nu,\rho} V^{\pi^n,\nu,\rho} \geqslant V^{\pi^n,\nu,\rho}$. Since $T^{\pi^{n+1},\nu,\rho}$ is a monotone increasing contraction under the assumptions of Proposition 4.2, applying repeatedly the operator $T^{\pi^{n+1},\nu,\rho}$ to the component-wise inequality $T^{\pi^{n+1},\nu,\rho} V^{\pi^n,\nu,\rho} \geqslant V^{\pi^n,\nu,\rho}$ yields $V^{\pi^{n+1},\nu,\rho} \geqslant V^{\pi^n,\nu,\rho}$.

We now prove that $V^{\pi^n,\nu,\rho}$ converges to $V^{\nu,\rho}$ at a linear rate. We have

$$
\begin{aligned}
V^{\nu,\rho} - V^{\pi^{n+1},\nu,\rho} &= V^{\nu,\rho} - T^{\pi^{n+1},\nu,\rho} V^{\pi^{n+1},\nu,\rho} \\
&= V^{\nu,\rho} - T^{\pi^{n+1},\nu,\rho} V^{\pi^n,\nu,\rho} + T^{\pi^{n+1},\nu,\rho} V^{\pi^n,\nu,\rho} - T^{\pi^{n+1},\nu,\rho} V^{\pi^{n+1},\nu,\rho} \\
&\leqslant V^{\nu,\rho} - T^{\pi^{n+1},\nu,\rho} V^{\pi^n,\nu,\rho}
\end{aligned}
$$

where we use Condition 1 for the first equality, where we add and subtract the vector $T^{\pi^n,\nu,\rho} V^{\pi^{n+1},\nu,\rho}$ in the second equality, and where the inequality follows from $T^{\pi^{n+1},\nu,\rho} V^{\pi^n,\nu,\rho} \leqslant T^{\pi^{n+1},\nu,\rho} V^{\pi^{n+1},\nu,\rho}$, itself a consequence of $V^{\pi^n,\nu,\rho} \leqslant V^{\pi^{n+1},\nu,\rho}$ and the fact that $T^{\pi^{n+1},\nu,\rho}$ is a monotone contraction. Now we have

$$
\begin{aligned}
\|V^{\nu,\rho} - T^{\pi^{n+1},\nu,\rho} V^{\pi^n,\nu,\rho}\|_\infty &= \|T^{\nu,\rho} V^{\nu,\rho} - T^{\pi^{n+1},\nu,\rho} V^{\pi^n,\nu,\rho}\|_\infty \\
&= \|T^{\pi^\star,\nu,\rho} V^{\nu,\rho} - T^{\pi^{n+1},\nu,\rho} V^{\pi^n,\nu,\rho}\|_\infty \\
&\leqslant \|T^{\pi^\star,\nu,\rho} V^{\nu,\rho} - T^{\pi^\star,\nu,\rho} V^{\pi^n,\nu,\rho}\|_\infty \\
&\leqslant \gamma \|V^{\nu,\rho} - V^{\pi^n,\nu,\rho}\|_\infty
\end{aligned}
$$

where the equality uses Condition 2, the second equality uses the existence of a policy $\pi^\star \in \Pi_S$ such that $T^{\pi^\star,\nu,\rho} V^{\nu,\rho} = V^{\nu,\rho}$ (the attainability property from Proposition 3.7), the first inequality is from the definition of $\pi^{n+1}$, and the last inequality is from the operator $T^{\pi^\star,\nu,\rho}$ being a contraction. By induction we directly obtain that

$$\|V^{\nu,\rho} - V^{\pi^n,\nu,\rho}\|_\infty \leqslant \gamma^n \|V^{\nu,\rho} - V^{\pi^0,\nu,\rho}\|_\infty$$

which concludes the proof.

$\square$

It is also possible to obtain a convex program with $V^{\nu,\rho}$ as a unique solution, as we show in the next proposition. We need here that $\rho$ is a convex risk measure, in the sense defined in Appendix A: $\rho(\lambda \tilde{X} + (1-\lambda)\tilde{Y}) \leqslant \lambda \rho(\tilde{X}) + (1-\lambda)\rho(\tilde{Y})$ for any random variables $\tilde{X}, \tilde{Y}$ and scalar $\lambda \in [0,1]$. An example of convex risk measure is the essential supremum.

**Proposition G.1.** *Assume that $\rho$ is monotone, translation-invariant and convex. Then $V^{\nu,\rho}$ can be recovered as the unique solution to the following convex optimization program:*

$$\min \left\{ \sum_{s \in \mathcal{S}} V(s) \mid V \in \mathbb{R}^{\mathcal{S}}, V(s) \geqslant T^{\nu,\rho} V(s), \forall\, s \in \mathcal{S} \right\}.$$

*Proof.* Since $\rho$ is monotone and translation invariant, we know that $T^{\nu,\rho}$ is a monotone contraction from Proposition 3.7. From the contraction lemma (Lemma 3.1 in Grand-Clément & Petrik (2025)), $V^{\nu,\rho}$ is the unique fixed point of the optimization problem

$$\min \left\{ \sum_{s \in \mathcal{S}} V(s) \mid V \in \mathbb{R}^{\mathcal{S}}, V(s) \geqslant T^{\nu,\rho} V(s), \forall\, s \in \mathcal{S} \right\}.$$

It remains to show that this optimization problem is convex. By construction, $T^{\pi,P}$ is a monotone affine operator for each $\pi \in \Pi_S, P \in \Delta(\mathcal{S})^{\mathcal{S} \times \mathcal{A}}$. Therefore, $v \mapsto T^{\pi,\nu,\rho} v(s)$ is convex, since by definition $T^{\pi,\nu,\rho}$ is the composition of a convex operator $T^{\pi,P}$ with monotone convex random variable $\rho$ (see Section 3.2.4 in Boyd & Vandenberghe (2004)), and $v \mapsto T^{\nu,\rho} v(s)$ is convex as the pointwise supremum of convex functions (see Section 3.2.3 in Boyd & Vandenberghe (2004)). $\square$

**Evaluating the Bellman operators**    The efficient computations of the Bellman operators have been studied in some cases:

- *The case of robust MDPs.* In this case we have, for $v \in \mathbb{R}^{\mathcal{S}}$ and $s \in \mathcal{S}$,

$$T^{\pi,\nu,\rho}v(s) = \inf_{P \in \mathcal{P}} T^{\pi,P}v(s)$$

$$T^{\nu,\rho}v(s) = \sup_{\pi \in \Pi_S} \inf_{P \in \mathcal{P}} T^{\pi,P}v(s)$$

which shows that evaluating $T^{\pi,\nu,\rho}v$ requires solving linear programs over $\mathcal{P}$, and evaluating $T^{\nu,\rho}v$ requires solving saddle-point problems over $\Pi_S \times \mathcal{P}$. Under some additional assumptions on the uncertainty set $\mathcal{P}$, one can evaluate the Bellman operators in polynomial time, e.g., for sets based on weighted $\ell_1$-norm (Iyengar, 2005; Ho et al., 2018), $\ell_2$-norm (Iyengar, 2005), entropy and Kullback-Leibler divergence (Nilim & El Ghaoui, 2005), and $\ell_\infty$-norm (Givan et al., 2000).

- *The case of optimistic MDPs.* This case can be treated similarly to robust MDPs by noting that

$$\operatorname{ess\,sup}(\tilde{X}) = -\operatorname{ess\,inf}(-\tilde{X}).$$

- *The case of multi-model MDPs.* In this case we have $\rho(\tilde{X}) = \mathbb{E}^{\nu}(\tilde{X})$, and

$$T^{\pi,\nu,\rho}v(s) = \mathbb{E}^{\nu}\left(T^{\pi,\tilde{P}}v(s)\right) = T^{\pi,\mathbb{E}^{\nu}(\tilde{P})}v(s)$$

where the second equality is because $P \mapsto T^{\pi,P}v(s)$ is an affine function. This shows that evaluating the Bellman operator for $\rho = \mathbb{E}^{\nu}$ can be done in two steps: (i) computing the expected kernel $\mathbb{E}^{\nu}(\tilde{P})$ (ii) computing the nominal Bellman operator with the expected kernel.

- The case of more general risk measures may require resorting to numerical estimates. From the definition of the ambiguity-averse Bellman operators, evaluating $T^{\pi,\nu,\rho}v(s)$ requires evaluating $\rho(\tilde{X})$ for some random variable $\tilde{X}$.

  For instance, for $\mathsf{CVaR}$ one can compute $\mathsf{CVaR}(\tilde{X})$ by solving the optimization problem (10), which only involves a supremum over a single real variable. In general, one can result to sample-average approximation or Monte-Carlo methods to estimate $\mathsf{VaR}$ or $\mathsf{CVaR}$, and we refer to Chapters 5-6 in Shapiro et al. (2021) for some discussion on this and to Rockafellar & Uryasev (2000) for a seminal reference. It is worth noting that our main theorems (Theorem 4.4-Theorem 4.5) show that dynamic programming does not apply beyond $\operatorname{ess\,sup}, \operatorname{ess\,inf}, \mathbb{E}^{\nu}$, so that evaluating the Bellman operator in order to apply Value Iteration or Policy Iteration may not lead to meaningful optimization over the set of stationary policies.

**Approximate dynamic-programming.**    The convergence analyses of value and policy iteration in our setting rely on the same monotonicity, contraction, and translation-invariance properties of Bellman operators as in nominal MDPs, recovered in Proposition 3.7. Therefore, error-propagation arguments for approximate value or policy iteration could be adapted (in the lines of Scherrer et al. (2015)) when the Bellman operator is evaluated inexactly.

# H. Proofs of Section 4.2

In this appendix we provide the proof for the results stated in Theorem 4.4 and Theorem 4.5. As discussed in Section 4.2, some of these results are already known:

- For static kernels, the literature on s-rectangular robust MDPs (Wiesemann et al., 2013) and optimistic MDPs (Givan et al., 2000) show that Conditions 1 and 2 hold when $\rho \in \{\operatorname{ess\,inf}, \operatorname{ess\,sup}\}$.

- For resampled kernels, a recent result for s-rectangular robust MDPs (Proposition 2.2 in Wiesemann et al. (2013)) and a seminal result in DRMDPs (Theorem 4.1 and Theorem 4.2 in Xu & Mannor (2012)) show that Conditions 1 and 2 are implied by the fact that $\rho \in \{\operatorname{ess\,inf}, \operatorname{ess\,sup}, \mathbb{E}^{\nu}\}$.

We also recall that $\operatorname{ess\,sup}, \operatorname{ess\,inf}$ are not $W^1$-continuous (see Appendix A). Therefore, we are left with proving that:

- For proving Theorem 4.4, there remains to show that for $\rho$ monotone and law-invariant,

    1. for static kernels, each Condition 1 and Condition 2 individually implies $\rho \in \{\operatorname{ess\,inf}, \operatorname{ess\,sup}\}$
    2. for resampled kernels, each Condition 1 and Condition 2 individually implies $\rho \in \{\operatorname{ess\,inf}, \operatorname{ess\,sup}, \mathbb{E}^\nu\}$.

- For proving Theorem 4.5, we show that for $\rho$ $W^1$-continuous and law-invariant, each Condition 1 and Condition 2 individually implies that $\rho = \mathbb{E}^\nu$ for both the cases of static and resampled kernels. We then conclude since we know that $\rho = \mathbb{E}^\nu$ doesn't yield dynamic programming equation for static kernels, as highlighted in example E.1.

We provide the proof of Proposition 4.3 in Appendix H.1, the proof of Theorem 4.4 in Appendix H.2, and the proof of Theorem 4.5 in Appendix H.3. **In the remainder of this section, we use the notations and the rigorous probabilistic framework introduced in Appendix B**.

## H.1. Proof of Proposition 4.3

We start with the following lemma. Both equations can be derived from Condition 1 or Condition 2 applied to the single-policy MDP structure of Figure 3. We emphasize here that we only use the single-policy MDP structure of Figure 3 among all the possibilities in $\mathcal{M}_c$ (because both conditions Condition 1 and Condition 2 concern all the ambiguity-averse MDPs in $\mathcal{M}_c$). While the second equation will be useful later to establish that $\rho(0) = 0$, the first equation alone contains the key property needed to derive all the results of Proposition 4.3. We recall that $(\Omega_\mathsf{M}, \mathcal{F}_\mathsf{M}, \mathbb{P}_\mathsf{M})$ is the probability space associated with a specific ambiguity-averse MDP M, as defined right after Definition B.2.

**Lemma H.1.** *Let $\rho$ be a risk measure that is law invariant on $L_c^\infty(\mathbb{R})$. If $\rho$ satisfies Condition 1 or Condition 2 in either the case of static kernels or resampled kernels, then the following statements hold:*

1. *for any $\mathsf{M} \in \mathcal{M}$ and independent random variables $(\tilde{X}, \tilde{Y}, \tilde{Z}) \in L_c^\infty(\Omega_\mathsf{M}, \mathcal{F}_\mathsf{M}, \mathbb{P}_\mathsf{M}; \mathbb{R})^2 \times L_c^\infty(\Omega_\mathsf{M}, \mathcal{F}_\mathsf{M}, \mathbb{P}_\mathsf{M}; [0,1])$,*

$$\forall \gamma \in [0,1), \quad \rho\Big(\gamma\big(\tilde{Z}\tilde{X} + (1-\tilde{Z})\tilde{Y}\big)\Big) = \rho\Big(\gamma\big(\tilde{Z}\rho(\tilde{X}) + (1-\tilde{Z})\rho(\tilde{Y})\big)\Big).$$

2. $\forall C \in \mathbb{R}, \forall \gamma \in [0,1), \rho(C) = \rho(C + \gamma\rho(0))$.

*Proof.* Let $\rho$ be a risk measure that is law-invariant on $L_c^\infty(\mathbb{R})$ and that satisfies Condition 1 or Condition 2 in either the case of static kernels or resampled kernels and $\gamma \in [0,1)$. The statement is obvious for $\gamma = 0$, and we assume $\gamma \in (0,1)$ in the rest of this proof. We will consider an ambiguity-averse MDP with a single possible action, therefore there is a single possible policy (the one that consists in always taking the only possible action with probability 1), noted $\pi$, and Condition 1 or Condition 2 applied to this specific ambiguity-averse MDP yield the exact same conclusion.

Let $\mathsf{M} \in \mathcal{M}$ and $(\tilde{X}, \tilde{Y}, \tilde{Z}) \in L_c^\infty(\Omega_\mathsf{M}, \mathcal{F}_\mathsf{M}, \mathbb{P}_\mathsf{M}; \mathbb{R})^2 \times L_c^\infty(\Omega_\mathsf{M}, \mathcal{F}_\mathsf{M}, \mathbb{P}_\mathsf{M}; [0,1])$ be three independent random variables. Because $\tilde{X}, \tilde{Y}$ are bounded, there exists $a, b, c, d \in \mathbb{R}^2 \times (\mathbb{R}^\star)^2$ such that $\frac{\tilde{X}-a}{c} \in [0,1]$ a.s. and $\frac{\tilde{Y}-b}{d} \in [0,1]$ a.s. Our goal is now to define the ambiguity-averse MDP of Figure 3 in such a way that $\mathcal{L}(\tilde{X}') = \mathcal{L}(\frac{\tilde{X}-a}{c})$, $\mathcal{L}(\tilde{Y}') = \mathcal{L}(\frac{\tilde{Y}-b}{d})$ and $\mathcal{L}(\tilde{Z}') = \mathcal{L}(\tilde{Z})$, where $\mathcal{L}(\cdot)$ denotes the law of a random variable.

Let $\mathcal{S} = \{\mathsf{Start}, \mathsf{Left}, \mathsf{Right}, \mathsf{WinL}, \mathsf{LoseL}, \mathsf{WinR}, \mathsf{LoseR}\}$, $\mathcal{A} = \{1\}$, $\gamma \in (0,1)$ and $r$ a reward function such that $r(\mathsf{Left}, 1, s') = a$, $r(\mathsf{Right}, 1, s') = b$, $r(\mathsf{WinL}, 1, s') = \gamma^{-1}c$, $r(\mathsf{WinR}, 1, s') = \gamma^{-1}d$ and the reward associated with any other transition is equal to $0$. We now define the transition kernel law $\nu$ as

$$\nu = \mathcal{L}\left(\begin{pmatrix} 0 & \tilde{Z} & 1-\tilde{Z} & 0 & 0 & 0 & 0 \\ 0 & 0 & 0 & \frac{\tilde{X}-a}{c} & 1-\frac{\tilde{X}-a}{c} & 0 & 0 \\ 0 & 0 & 0 & 0 & 0 & \frac{\tilde{Y}-b}{d} & 1-\frac{\tilde{Y}-b}{d} \\ 0 & 0 & 0 & 0 & 1 & 0 & 0 \\ 0 & 0 & 0 & 0 & 1 & 0 & 0 \\ 0 & 0 & 0 & 0 & 0 & 0 & 1 \\ 0 & 0 & 0 & 0 & 0 & 0 & 1 \end{pmatrix}\right).$$

Because $\tilde{X}, \tilde{Y}, \tilde{Z}$ all have convex support, the law $\nu$ has convex support and by independence of $\tilde{X}, \tilde{Y}, \tilde{Z}$, $\nu$ also has a product structure. Consequently, having $\mu \in \Delta(\mathcal{S})$, the tuple $(\mathcal{S}, \mathcal{A}, r, \nu, \gamma, \mu)$ is a valid "convex, product" ambiguity-averse MDP that we note $\mathsf{M}' \in \mathcal{M}_c$.

The transition kernel sequence now depends on the sampling case:

- **Static transition kernel sequence:**
  Denoting by $\tilde{P}$ the static transition kernel sequence of the ambiguity-averse MDP M$'$, we define $\tilde{Z}' = \tilde{P}_{t=0}(\mathsf{Start}, 1, \mathsf{Left})$, $\tilde{X}' = \tilde{P}_{t=0}(\mathsf{Left}, 1, \mathsf{WinL})$ and $\tilde{Y}' = \tilde{P}_{t=0}(\mathsf{Right}, 1, \mathsf{WinR})$. By construction $\tilde{X}', \tilde{Y}', \tilde{Z}'$ are three independent random variables in $L_c^\infty(\Omega_{\mathsf{M}'}, \mathcal{F}_{\mathsf{M}'}, \mathbb{P}_{\mathsf{M}'}; [0,1])$ such that $\mathcal{L}(\tilde{X}') = \mathcal{L}(\frac{\tilde{X}-a}{c})$, $\mathcal{L}(\tilde{Y}') = \mathcal{L}(\frac{\tilde{Y}-b}{d})$ and $\mathcal{L}(\tilde{Z}') = \mathcal{L}(\tilde{Z})$. In particular, by independence, we also have $\mathcal{L}\left((c\tilde{X}' + a, d\tilde{Y}' + b, \tilde{Z}')\right) = \mathcal{L}\left((\tilde{X}, \tilde{Y}, \tilde{Z})\right)$.

The transition kernel of M$'$ is such that Start goes to Left with probability $\tilde{Z}'$ and to Right with probability $1 - \tilde{Z}'$. From Left, the transition to WinL happens with probability $\tilde{X}'$ and to LoseL with probability $1 - \tilde{X}'$. From Right, the transition to WinR happens with probability $\tilde{Y}'$ and to LoseR with probability $1 - \tilde{Y}'$. From WinL/WinR, the transition goes deterministically to LoseL/LoseR, and from LoseL/LoseR, the state self-loops.

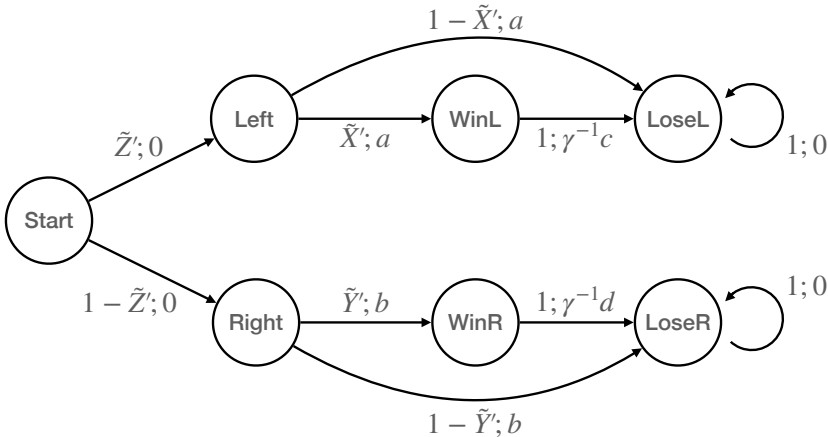

*Figure 3.* Ambiguity-averse MDP with $\mathcal{S} = \{\mathsf{Start}, \mathsf{Left}, \mathsf{Right}, \mathsf{WinL}, \mathsf{LoseL}, \mathsf{WinR}, \mathsf{LoseR}\}$, $\mathcal{A} = \{1\}$. The edges $(s, a, s')$ are labeled with pairs $(P(s, a, s'), r(s, a, s'))$.

On the one hand, because there is a single policy, the ambiguity-averse value function and the optimal ambiguity-averse value function are equal, and $V = V^{\pi,\nu,\rho} = V^{\nu,\rho}$ is given by

$$V(\mathsf{Start}) = \rho\left(\gamma\left(\tilde{Z}'(a + \gamma\tilde{X}' \cdot \gamma^{-1}c) + (1 - \tilde{Z}')(b + \gamma\tilde{Y}' \cdot \gamma^{-1}d)\right)\right)$$

$$= \rho\left(\gamma\left(\tilde{Z}\tilde{X} + (1 - \tilde{Z})\tilde{Y}\right)\right)$$

$$V(\mathsf{Left}) = \rho\left(a + \gamma\tilde{X}'\gamma^{-1}c + \gamma(1 - \tilde{X}') \cdot 0\right) = \rho\left(a + \tilde{X}'c\right) = \rho(\tilde{X})$$

$$V(\mathsf{Right}) = \rho\left(b + \gamma\tilde{Y}'\gamma^{-1}d + \gamma(1 - \tilde{Y}') \cdot 0\right) = \rho\left(b + \tilde{Y}'d\right) = \rho(\tilde{Y})$$

$$V(\mathsf{WinL}) = \rho\left(\gamma^{-1}c\right)$$

$$V(\mathsf{LoseL}) = \rho(0)$$

$$V(\mathsf{WinR}) = \rho\left(\gamma^{-1}d\right)$$

$$V(\mathsf{LoseR}) = \rho(0) \,,$$

where the last equalities in the first three lines come from $\mathcal{L}\left((c\tilde{X}' + a, d\tilde{Y}' + b, \tilde{Z}')\right) = \mathcal{L}\left((\tilde{X}, \tilde{Y}, \tilde{Z})\right)$ and $\rho$ is law-invariant.

On the other hand, the operator $T = T^{\pi,\nu,\rho} = T^{\nu,\rho}$ is defined by:

$$
\forall v \in \mathbb{R}^7, \quad T \left( \begin{pmatrix} v(\mathsf{Start}) \\ v(\mathsf{Left}) \\ v(\mathsf{Right}) \\ v(\mathsf{WinL}) \\ v(\mathsf{LoseL}) \\ v(\mathsf{WinR}) \\ v(\mathsf{LoseR}) \end{pmatrix} \right) = \begin{pmatrix} \rho\left( \gamma(\tilde{Z}'v(\mathsf{Left}) + (1 - \tilde{Z}')v(\mathsf{Right})) \right) \\ \rho\left( a + \gamma(\tilde{X}'v(\mathsf{WinL}) + (1 - \tilde{X}')v(\mathsf{LoseL})) \right) \\ \rho\left( b + \gamma(\tilde{Y}'v(\mathsf{WinR}) + (1 - \tilde{Y}')v(\mathsf{LoseR})) \right) \\ \rho\left( \gamma^{-1}c + \gamma v(\mathsf{LoseL}) \right) \\ \rho\left( \gamma v(\mathsf{LoseL}) \right) \\ \rho\left( \gamma^{-1}d + \gamma v(\mathsf{LoseR}) \right) \\ \rho\left( \gamma v(\mathsf{LoseR}) \right) \end{pmatrix} .
$$

Then, by Condition 1 or Condition 2, because $\mathsf{M}' \in \mathcal{M}_c$ (and $\pi \in \Pi_S$), we have $V = TV$:

$$
\begin{cases}
\rho\left( \gamma\left( \tilde{Z}\tilde{X} + (1 - \tilde{Z})\tilde{Y} \right) \right) = \rho\left( \gamma\left( \tilde{Z}'\rho(\tilde{X}) + (1 - \tilde{Z}')\rho(\tilde{Y}) \right) \right) \\
\rho\left( \tilde{X} \right) = \rho\left( a + \gamma\left( \tilde{X}'\rho\left( \gamma^{-1}c \right) + (1 - \tilde{X}')\rho(0) \right) \right) \\
\rho\left( \tilde{Y} \right) = \rho\left( b + \gamma\left( \tilde{Y}'\rho\left( \gamma^{-1}d \right) + (1 - \tilde{Y}')\rho(0) \right) \right) \\
\rho\left( \gamma^{-1}c \right) = \rho\left( \gamma^{-1}c + \gamma\rho(0) \right) \\
\rho(0) = \rho\left( \gamma\rho(0) \right) \\
\rho\left( \gamma^{-1}d \right) = \rho\left( \gamma^{-1}d + \gamma\rho(0) \right) \\
\rho(0) = \rho\left( \gamma\rho(0) \right)
\end{cases}
$$

The first equation gives the first part of the desired result after using $\mathcal{L}(\tilde{Z}') = \mathcal{L}(\tilde{Z})$ and law-invariance.

We also note that the fourth equation $\rho\left( \gamma^{-1}c \right) = \rho\left( \gamma^{-1}c + \gamma\rho(0) \right)$ holds for any $c \in \mathbb{R}$. Consequently, we also have $\forall C \in \mathbb{R}, \rho(C) = \rho(C + \gamma\rho(0))$.

- **Resampled kernels:** Denoting by $\tilde{P}$ the resampled transition kernel sequence of the ambiguity-averse MDP $\mathsf{M}' \in \mathcal{M}_c$, we define, for any $t \in \mathbb{N}$, $\tilde{Z}'_t = \tilde{P}_t(\mathsf{Start}, 1, \mathsf{Left})$, $\tilde{X}'_t = \tilde{P}_t(\mathsf{Left}, 1, \mathsf{WinL})$ and $\tilde{Y}'_t = \tilde{P}_t(\mathsf{Right}, 1, \mathsf{WinR})$. By construction, for any $t \in \mathbb{N}$, $\tilde{X}'_t, \tilde{Y}'_t, \tilde{Z}'_t$ are three independent random variables in $L^\infty_c(\Omega_{\mathsf{M}'}, \mathcal{F}_{\mathsf{M}'}, \mathbb{P}_{\mathsf{M}'}; [0,1])$ such that $\mathcal{L}(\tilde{X}'_t) = \mathcal{L}(\frac{\tilde{X}-a}{c})$, $\mathcal{L}(\tilde{Y}'_t) = \mathcal{L}(\frac{\tilde{Y}-b}{d})$ and $\mathcal{L}(\tilde{Z}'_t) = \mathcal{L}(\tilde{Z})$. In particular, by independence, for any $t_1, t_2, t_3 \in \mathbb{N}$ we also have $\mathcal{L}\left( (c\tilde{X}'_{t_1} + a, d\tilde{Y}'_{t_2} + b, \tilde{Z}'_{t_3}) \right) = \mathcal{L}\left( (\tilde{X}, \tilde{Y}, \tilde{Z}) \right)$.

Therefore, at time $t \in \mathbb{N}$, the transition kernel of $\mathsf{M}'$ is such that $\mathsf{Start}$ goes to $\mathsf{Left}$ with probability $\tilde{Z}'_t$ and to $\mathsf{Right}$ with probability $1 - \tilde{Z}'_t$. From $\mathsf{Left}$, the transition to $\mathsf{WinL}$ happens with probability $\tilde{X}'_t$ and to $\mathsf{LoseL}$ with probability $1 - \tilde{X}'_t$. From $\mathsf{Right}$, the transition to $\mathsf{WinR}$ happens with probability $\tilde{Y}'_t$ and to $\mathsf{LoseR}$ with probability $1 - \tilde{Y}'_t$. From $\mathsf{WinL/WinR}$, the transition goes deterministically to $\mathsf{LoseL/LoseR}$, and from $\mathsf{LoseL/LoseR}$, the state self-loops.

Now, $V = V^{\pi,\nu,\rho} = V^{\nu,\rho}$ from starting at time $t = 0$ in a state $s$ is given by

$$
\begin{aligned}
V(\mathsf{Start}) &= \rho\left( \gamma\left( \tilde{Z}'_0(a + \gamma\tilde{X}'_1 \cdot \gamma^{-1}c) + (1 - \tilde{Z}'_0)(b + \gamma\tilde{Y}'_1 \cdot \gamma^{-1}d) \right) \right) \\
&= \rho\left( \gamma\left( \tilde{Z}\tilde{X} + (1 - \tilde{Z})\tilde{Y} \right) \right) \\
V(\mathsf{Left}) &= \rho\left( a + \gamma\tilde{X}'_0\gamma^{-1}c + \gamma(1 - \tilde{X}'_0) \cdot 0 \right) = \rho\left( a + \tilde{X}'_0 c \right) = \rho(\tilde{X}) \\
V(\mathsf{Right}) &= \rho\left( b + \gamma\tilde{Y}'\gamma^{-1}d + \gamma(1 - \tilde{Y}'_0) \cdot 0 \right) = \rho\left( b + \tilde{Y}'_0 d \right) = \rho(\tilde{Y}) \\
V(\mathsf{WinL}) &= \rho\left( \gamma^{-1}c \right) \\
V(\mathsf{LoseL}) &= \rho(0) \\
V(\mathsf{WinR}) &= \rho\left( \gamma^{-1}d \right) \\
V(\mathsf{LoseR}) &= \rho(0)
\end{aligned}
$$

where the last equalities in the first three lines come from the law invariance of $\rho$ and from the fact that $\mathcal{L}\left((a + c\tilde{X}_1', b + d\tilde{Y}_1', \tilde{Z}_0')\right) = \mathcal{L}\left((\tilde{X}, \tilde{Y}, \tilde{Z})\right)$ and $\mathcal{L}\left((a + c\tilde{X}_0', b + d\tilde{Y}_0')\right) = \mathcal{L}\left((\tilde{X}, \tilde{Y})\right)$.

To explicitly describe the random return in the first line, we have that starting from position Start at time $t = 0$, the transition to Left happens with probability $\tilde{Z}_0'$ and to Right with probability $1 - \tilde{Z}_0'$. If the next state is Left, then at time $t = 1$, the transition to WinL happens with probability $\tilde{X}_1'$ and to LoseL with probability $1 - \tilde{X}_1'$, yielding a total return of $a + \gamma \tilde{X}_1' \cdot \gamma^{-1}c$. If the next state is Right, then at time $t = 1$, the transition to WinR happens with probability $\tilde{Y}_1'$ and to LoseR with probability $1 - \tilde{Y}_1'$, yielding a total return of $b + \gamma \tilde{Y}_1' \cdot \gamma^{-1}d$.

On the other hand, the operator $T = T^{\pi,\nu,\rho} = T^{\nu,\rho}$ is defined by:

$$\forall v \in \mathbb{R}^7, \quad T\left(\left(\begin{array}{c} v(\mathsf{Start}) \\ v(\mathsf{Left}) \\ v(\mathsf{Right}) \\ v(\mathsf{WinL}) \\ v(\mathsf{LoseL}) \\ v(\mathsf{WinR}) \\ v(\mathsf{LoseR}) \end{array}\right)\right) = \left(\begin{array}{c} \rho\left(\gamma(\tilde{Z}_0'v(\mathsf{Left}) + (1 - \tilde{Z}_0')v(\mathsf{Right}))\right) \\ \rho\left(a + \gamma(\tilde{X}_0'v(\mathsf{WinL}) + (1 - \tilde{X}_0')v(\mathsf{LoseL}))\right) \\ \rho\left(b + \gamma(\tilde{Y}_0'v(\mathsf{WinR}) + (1 - \tilde{Y}_0')v(\mathsf{LoseR}))\right) \\ \rho\left(\gamma^{-1}c + \gamma v(\mathsf{LoseL})\right) \\ \rho\left(\gamma v(\mathsf{LoseL})\right) \\ \rho\left(\gamma^{-1}d + \gamma v(\mathsf{LoseR})\right) \\ \rho\left(\gamma v(\mathsf{LoseR})\right) \end{array}\right).$$

Then, by Condition 1 or Condition 2, because $\mathsf{M}' \in \mathcal{M}_c$ (and $\pi \in \Pi_S$), we have $V = TV$:

$$\begin{cases} \rho\left(\gamma\left(\tilde{Z}\tilde{X} + (1 - \tilde{Z})\tilde{Y}\right)\right) = \rho\left(\gamma(\tilde{Z}_0'\rho(\tilde{X}) + (1 - \tilde{Z}_0')\rho(\tilde{Y}))\right) \\ \rho(\tilde{X}) = \rho\left(a + \gamma(\tilde{X}_0'\rho\left(\gamma^{-1}c\right) + (1 - \tilde{X}_0')\rho(0))\right) \\ \rho(\tilde{Y}) = \rho\left(b + \gamma(\tilde{Y}_0'\rho\left(\gamma^{-1}d\right) + (1 - \tilde{Y}_0')\rho(0))\right) \\ \rho\left(\gamma^{-1}c\right) = \rho\left(\gamma^{-1}c + \gamma\rho(0)\right) \\ \rho(0) = \rho\left(\gamma\rho(0)\right), \\ \rho\left(\gamma^{-1}d\right) = \rho\left(\gamma^{-1}d + \gamma\rho(0)\right) \\ \rho(0) = \rho\left(\gamma\rho(0)\right) \end{cases}$$

The first equation gives

$$\rho\left(\gamma\left(\tilde{Z}\tilde{X} + (1 - \tilde{Z})\tilde{Y}\right)\right) = \rho\left(\gamma(\tilde{Z}_0'\rho(\tilde{X}) + (1 - \tilde{Z}_0')\rho(\tilde{Y}))\right)$$
$$= \rho\left(\gamma(\tilde{Z}\rho(\tilde{X}) + (1 - \tilde{Z})\rho(\tilde{Y}))\right),$$

after using the law invariance of $\rho$ and from the fact that $\mathcal{L}(\tilde{Z}_0') = \mathcal{L}(\tilde{Z})$.

As in the case of static kernels, the fourth equation implies that $\forall C \in \mathbb{R}, \rho(C) = \rho(C + \gamma\rho(0))$.

$\square$

We will use the next technical lemma.

**Lemma H.2.** *Let $f : \mathbb{R} \to \mathbb{R}$ a non-constant function such that*

$$\forall (x, y, z) \in \mathbb{R}^2 \times [0, 1], \forall \gamma \in [0, 1), \quad f\left(\gamma(zx + (1 - z)y)\right) = f\left(\gamma(zf(x) + (1 - z)f(y))\right) \tag{22}$$

$$\forall C \in \mathbb{R}, \forall \gamma \in [0, 1) \quad f(C) = f(C + \gamma f(0)) \tag{23}$$

*Then $f$ is the identity: $f(x) = x, \forall x \in \mathbb{R}$.*

*Proof.* Let $f$ be a non-constant function that satisfies Equations (22)-(23). We first have that $f(0) = 0$. Indeed, if $f(0) \neq 0$, then for any $C \in \mathbb{R}$, Equation (23) directly implies that $f$ is constant on any closed interval of length $|f(0)/2| > 0$ and then, splitting the real line with points $(n\frac{f(0)}{2})_{n \in \mathbb{Z}}$, we get that $f$ is constant on $\mathbb{R}$, which contradicts the assumption.

We show the following auxiliary result: $\forall a < b, f(a) = f(b) \implies f$ is constant on $(a, b)$. Let $a < b$ such that $f(a) = f(b)$, and let $x, y \in (a, b)$. There exists $\gamma \in [0, 1)$ such that $x, y \in [\gamma a, \gamma b]$. By (22),

$$
\begin{aligned}
f(x) = f\Big(\gamma\big(zb + (1-z)a\big)\Big) &= f\Big(\gamma\big(zf(b) + (1-z)f(a)\big)\Big) \\
&= f\Big(\gamma f(a)\Big) \\
&= f\Big(\gamma\big(z'f(b) + (1-z')f(a)\big)\Big) \\
&= f\Big(\gamma\big(z'b + (1-z')a\big)\Big) = f(y) \; ,
\end{aligned}
$$

where $z = \frac{x/\gamma - a}{b - a} \in [0, 1]$ and $z' = \frac{y/\gamma - a}{b - a} \in [0, 1]$. So $f$ is constant on $(a, b)$.

We now prove by contradiction that $f$ is injective. Let $a < b$ such that $f(a) = f(b)$. We show that $f$ is constant on $\mathbb{R}$. Let $x < y \in \mathbb{R}$ such that $y - x = (b - a)/4$. We define $u = 4x - a = 4y - b \in \mathbb{R}$. Then, by (22) with $\gamma = z = 1/2$,

$$
\begin{aligned}
f(x) = f\left(\frac{1}{4}a + \frac{1}{4}u\right) &= f\left(\frac{1}{4}f(a) + \frac{1}{4}f(u)\right) \\
&= f\left(\frac{1}{4}f(b) + \frac{1}{4}f(u)\right) \\
&= f\left(\frac{1}{4}b + \frac{1}{4}u\right) = f(y) \; .
\end{aligned}
$$

Therefore, using our auxiliary result, $f$ is constant on any open interval of length $|b - a|/4 > 0$, so $f$ is constant on $\mathbb{R}$, which contradicts the assumption.

Finally, by (22) with $\gamma = z = 1/2$ and $y = 0$, we have

$$
\forall x \in \mathbb{R}, f(x/4) = f\left(\frac{1}{4}x + \frac{1}{4}0\right) = f\left(f(x)/4 + f(0)/4\right) = f(f(x)/4) \; ,
$$

so by injectivity of $f$, we have $\forall x \in \mathbb{R}, x/4 = f(x)/4$ and $f$ is the identity. $\qquad\square$

We now prove a slightly more general result than Proposition 4.3:

**Proposition H.3.** *Let $\rho$ be a risk measure that is law-invariant and non-constant on $L_c^\infty(\mathbb{R})$. If $\rho$ satisfies Condition 1 or Condition 2 in either the case of static kernels or resampled kernels, then the following statements hold:*

1. *$\rho$ coincides with the identity on constant random variables:*   $\forall x \in \mathbb{R}, \quad \rho(x) = x$

2. *$\rho$ is positive homogeneous on $L_c^\infty(\mathbb{R})$:*   $\forall \tilde{X} \in L_c^\infty(\mathbb{R}), \forall \alpha \geqslant 0, \quad \rho(\alpha \tilde{X}) = \alpha\rho(\tilde{X})$

3. *$\rho$ is additive independent on $L_c^\infty(\mathbb{R})$:*

   *for any $\mathsf{M} \in \mathcal{M}$ and independent $\tilde{X}, \tilde{Y} \in L_c^\infty(\Omega_\mathsf{M}, \mathcal{F}_\mathsf{M}, \mathbb{P}_\mathsf{M}; \mathbb{R}), \quad \rho(\tilde{X} + \tilde{Y}) = \rho(\tilde{X}) + \rho(\tilde{Y}) \; .$*

4. *If $\rho$ is also monotone on $L_c^\infty(\mathbb{R})$, then,*

   *for any $\mathsf{M} \in \mathcal{M}$ and independent $\tilde{X}, \tilde{Y} \in L_c^\infty(\Omega_\mathsf{M}, \mathcal{F}_\mathsf{M}, \mathbb{P}_\mathsf{M}; \mathbb{R}_+), \quad \rho(\tilde{X}\tilde{Y}) = \rho(\tilde{X})\rho(\tilde{Y}) \; .$*

*Proof.* Let $\rho$ be a risk measure that is law-invariant and non-constant on $L_c^\infty(\mathbb{R})$, and that satisfies Condition 1 or Condition 2 in either the case of static kernels or resampled kernels.

By the conclusion of Lemma H.1 we have, for any $\mathsf{M} \in \mathcal{M}$ and independent random variables $(\tilde{X}, \tilde{Y}, \tilde{Z}) \in L_c^\infty(\Omega_\mathsf{M}, \mathcal{F}_\mathsf{M}, \mathbb{P}_\mathsf{M}; \mathbb{R})^2 \times L_c^\infty(\Omega_\mathsf{M}, \mathcal{F}_\mathsf{M}, \mathbb{P}_\mathsf{M}; [0, 1])$,

$$
\forall \gamma \in [0, 1), \quad \rho\Big(\gamma\big(\tilde{Z}\tilde{X} + (1 - \tilde{Z})\tilde{Y}\big)\Big) = \rho\Big(\gamma\big(\tilde{Z}\rho(\tilde{X}) + (1 - \tilde{Z})\rho(\tilde{Y})\big)\Big) . \tag{24}
$$

and $\forall C \in \mathbb{R}, \forall \gamma \in [0,1), \rho(C) = \rho(C + \gamma \rho(0))$ .

We first show that $\rho_{|\mathbb{R}} : x \in \mathbb{R} \mapsto \rho(x)$, the restriction of $\rho$ to $\mathbb{R}$, is non-constant. In fact, because $\rho$ is non-constant on $L_c^\infty(\mathbb{R})$, there exists $\tilde{W} \in L_c^\infty(\mathbb{R})$ such that $\rho(\tilde{W}) \neq \rho(0)$. Then, applying (24) with $\tilde{Y} = 0$, $\tilde{Z} = 1$, $\gamma = 1/2$ and $\tilde{X} = 2\tilde{W}$, we have $\rho(1/2 \cdot \rho(2\tilde{W})) = \rho(1/2 \cdot 2\tilde{W}) = \rho(\tilde{W}) \neq \rho(0)$ so $\rho(2\tilde{W})/2$ is a real number whose image by $\rho$ is different from $\rho(0)$ and $\rho_{|\mathbb{R}}$ is non-constant.

By Lemma H.2 applied to $\rho_{|\mathbb{R}}$, we now know that $\forall x \in \mathbb{R}, \rho(x) = x$.

Then, Equation (24) with $\tilde{Z} = \lambda$ for $\lambda \in [0,1]$ now becomes, for any $\mathsf{M} \in \mathcal{M}$ and independent random variables $\tilde{X}, \tilde{Y} \in L_c^\infty(\Omega_\mathsf{M}, \mathcal{F}_\mathsf{M}, \mathbb{P}_\mathsf{M}; \mathbb{R})$,

$$\forall \lambda \in [0,1], \forall \gamma \in [0,1), \quad \rho\Big(\gamma\big(\lambda\tilde{X} + (1-\lambda)\tilde{Y}\big)\Big) = \gamma\big(\lambda\rho(\tilde{X}) + (1-\lambda)\rho(\tilde{Y})\big) . \tag{25}$$

Taking $\tilde{Y} = 0$ and $\lambda = 1$ in Equation (25) we obtain that,

$$\forall \tilde{X} \in L_c^\infty(\mathbb{R}), \forall \gamma \in [0,1), \quad \rho(\gamma\tilde{X}) = \gamma\rho(\tilde{X}) ,$$

which is still true for $\gamma = 1$. Now take $\alpha \geqslant 1$, we have $0 \leqslant \frac{1}{\alpha} \leqslant 1$ so that $\frac{1}{\alpha}\rho(\alpha\tilde{X}) = \rho(\frac{1}{\alpha}\alpha\tilde{X}) = \rho(\tilde{X})$, i.e. we have shown that $\rho$ is positively homogeneous on $L_c^\infty(\mathbb{R})$:

$$\forall \tilde{X} \in L_c^\infty(\mathbb{R}), \forall \alpha \geqslant 0, \quad \rho(\alpha\tilde{X}) = \alpha\rho(\tilde{X}) .$$

Then, taking $\gamma = \lambda = 1/2$ in Equation (25) and using positive homogeneity, we obtain that $\rho$ is additive for independent random variables:

$$\text{for any } \mathsf{M} \in \mathcal{M} \text{ and independent } (\tilde{X}, \tilde{Y}) \in L_c^\infty(\Omega_\mathsf{M}, \mathcal{F}_\mathsf{M}, \mathbb{P}_\mathsf{M}; \mathbb{R}), \quad \rho(\tilde{X} + \tilde{Y}) = \rho(\tilde{X}) + \rho(\tilde{Y}) .$$

Finally, if $\rho$ is also monotone, using Equation (24) with $\tilde{Y} = 0, \lambda = 1$ and $\gamma = 1/2$, we have that for any $\mathsf{M} \in \mathcal{M}$ and independent random variables $(\tilde{X}, \tilde{Z}) \in L_c^\infty(\Omega_\mathsf{M}, \mathcal{F}_\mathsf{M}, \mathbb{P}_\mathsf{M}; \mathbb{R}_+) \times L_c^\infty(\Omega_\mathsf{M}, \mathcal{F}_\mathsf{M}, \mathbb{P}_\mathsf{M}; [0,1])$:

$$\rho\Big(\frac{1}{2}\tilde{Z}\tilde{X}\Big) = \rho\Big(\frac{1}{2}\tilde{Z}\rho(\tilde{X})\Big) = \rho(\tilde{X})\rho\Big(\frac{1}{2}\tilde{Z}\Big) ,$$

where we have used positive homogeneity and the fact that $\rho(\tilde{X}) \geqslant \rho(0) = 0$ by monotonicity. Then, by positive homogeneity again, we obtain

$$\rho(\tilde{Z}\tilde{X}) = \rho(\tilde{Z})\rho(\tilde{X}) ,$$

which is still true if $(\tilde{X}, \tilde{Z}) \in L_c^\infty(\Omega_\mathsf{M}, \mathcal{F}_\mathsf{M}, \mathbb{P}_\mathsf{M}; \mathbb{R}_+)^2$ by positive homogeneity applied to $\tilde{Z}$. $\qquad\square$

## H.2. Proof of Theorem 4.4

We now provide the proof for Theorem 4.4. The proof relies on the following representation theorem from Mu et al. (2024), which we state here with the notations and terminology of our paper.

**Theorem H.4** (Theorem 1 in Mu et al. (2024))**.** *If a risk-measure $\Phi\colon L^\infty(\mathbb{R}) \to \mathbb{R}$ coincides with the identity for constants and is law-invariant, monotone and additive independent on $L^\infty$, then there exists a unique Borel probability measure $\mu$ on the extended real line $\overline{\mathbb{R}}$ such that for every $\tilde{X} \in L^\infty(\mathbb{R})$*

$$\Phi(\tilde{X}) = \int_{\overline{\mathbb{R}}} \mathsf{ERM}_a(\tilde{X}) \, d\mu(a).$$

**Remark H.5.** Monotonicity in Mu et al. (2024) is defined with respect to the first-order stochastic dominance but, as argued before Theorem 1 in Mu et al. (2024), this is equivalent to our notion of monotonicity because of law-invariance.

For the purpose of this work, we need a version of Theorem H.4 for risk measures defined on bounded random variables *with a convex support $L_c^\infty(\mathbb{R})$* (whereas Theorem H.4 only applies to risk measures on $L^\infty(\mathbb{R})$). This is done in the next lemma, which states that an additive independent law-invariant risk measure defined on $L_c^\infty(\mathbb{R})$ can be extended to a law-invariant risk measure $\mathring{\rho}$ defined on $L^\infty(\mathbb{R})$, which preserves its main properties (e.g., additive independence and monotonicity).

The key insight is that any random variable with nonconvex support can be represented as the difference of two random variables with convex support by adding an independent uniform noise. This property allows us to define the extension $\mathring{\rho}$ of any random variable with not-necessarily convex support as the difference of the images by $\rho$ of two random variables with convex support and this way preserve the algebraic properties (monotonicity, additive independence, positive homogeneity) required by the representation theorem. We recall that $(\Omega_{\mathsf{M}}, \mathcal{F}_{\mathsf{M}}, \mathbb{P}_{\mathsf{M}})$ is the probability space associated with a specific ambiguity-averse MDP $\mathsf{M} \in \mathcal{M}$, as defined right after Definition B.2.

**Lemma H.6.** *Let $\rho$ be a risk measure that is law-invariant on $L_c^\infty(\mathbb{R})$. Assume additionally that $\rho$ is additive independent on $L_c^\infty(\mathbb{R})$:*

$$\text{for any } \mathsf{M} \in \mathcal{M} \text{ and independent } \tilde{X}, \tilde{Y} \in L_c^\infty(\Omega_{\mathsf{M}}, \mathcal{F}_{\mathsf{M}}, \mathbb{P}_{\mathsf{M}}; \mathbb{R}), \quad \rho(\tilde{X} + \tilde{Y}) = \rho(\tilde{X}) + \rho(\tilde{Y}) .$$

*Then, there exists risk measure $\mathring{\rho}$ that is law-invariant on $L^\infty(\mathbb{R})$, that coincides with $\rho$ on $L_c^\infty(\mathbb{R})$ (i.e., $\forall X \in L_c^\infty(\mathbb{R})$, $\mathring{\rho}(X) = \rho(X)$) and such that the following statements hold:*

1. *$\mathring{\rho}$ is additive independent on $L^\infty(\mathbb{R})$:*

$$\text{for any } \mathsf{M} \in \mathcal{M} \text{ and independent } \tilde{X}, \tilde{Y} \in L^\infty(\Omega_{\mathsf{M}}, \mathcal{F}_{\mathsf{M}}, \mathbb{P}_{\mathsf{M}}; \mathbb{R}), \quad \mathring{\rho}(\tilde{X} + \tilde{Y}) = \mathring{\rho}(\tilde{X}) + \mathring{\rho}(\tilde{Y}) .$$

2. *if $\rho$ is positive homogeneous on $L_c^\infty(\mathbb{R})$, then $\mathring{\rho}$ is positive homogeneous on $L^\infty(\mathbb{R})$:*

$$\left[ \forall \tilde{X} \in L_c^\infty(\mathbb{R}), \forall \alpha \geqslant 0, \quad \rho(\alpha \tilde{X}) = \alpha \rho(\tilde{X}) \right] \implies \left[ \forall \tilde{X} \in L^\infty(\mathbb{R}), \forall \alpha \geqslant 0, \quad \mathring{\rho}(\alpha \tilde{X}) = \alpha \mathring{\rho}(\tilde{X}) \right] .$$

3. *if $\rho$ is monotone on $L_c^\infty(\mathbb{R})$, then $\mathring{\rho}$ is monotone on $L^\infty(\mathbb{R})$:*

$$\left[ \text{for any } \mathsf{M} \in \mathcal{M} \text{ and } \tilde{X}, \tilde{Y} \in L_c^\infty(\Omega_{\mathsf{M}}, \mathcal{F}_{\mathsf{M}}, \mathbb{P}_{\mathsf{M}}; \mathbb{R}) \text{ s.t. } \tilde{X} \geqslant \tilde{Y} \text{ a.s.,} \quad \rho(\tilde{X}) \geqslant \rho(\tilde{Y}) \right]$$
$$\implies \left[ \text{for any } \mathsf{M} \in \mathcal{M} \text{ and } \tilde{X}, \tilde{Y} \in L^\infty(\Omega_{\mathsf{M}}, \mathcal{F}_{\mathsf{M}}, \mathbb{P}_{\mathsf{M}}; \mathbb{R}) \text{ s.t. } \tilde{X} \geqslant \tilde{Y} \text{ a.s.,} \quad \mathring{\rho}(\tilde{X}) \geqslant \mathring{\rho}(\tilde{Y}) \right] .$$

*Proof of Lemma H.6.* Let $\rho$ be a risk measure that is law-invariant and additive independent on $L_c^\infty(\mathbb{R})$. Let $\mathsf{M}_0 \in \mathcal{M}$ an ambiguity-averse MDP such that $(\Omega_{\mathsf{M}_0}, \mathcal{F}_{\mathsf{M}_0}, \mathbb{P}_{\mathsf{M}_0})$ is non-atomic (for example the one of Example E.1).

For any $\tilde{X} \in L^\infty(\mathbb{R})$, we define $l_X = \operatorname{ess\,sup}(\tilde{X}) - \operatorname{ess\,inf}(\tilde{X})$. By non-atomicity $(\Omega_{\mathsf{M}_0}, \mathcal{F}_{\mathsf{M}_0}, \mathbb{P}_{\mathsf{M}_0})$, there exists $\tilde{X}', \tilde{U}_{\tilde{X}} \in L^\infty(\Omega_{\mathsf{M}_0}, \mathcal{F}_{\mathsf{M}_0}, \mathbb{P}_{\mathsf{M}_0}; \mathbb{R})$ such that $\mathcal{L}(\tilde{X}') = \mathcal{L}(\tilde{X})$, $\tilde{U}_{\tilde{X}} \sim \operatorname{Unif}([0, l_X])$ and $\tilde{X}', \tilde{U}_{\tilde{X}}$ are independent. We call $\tilde{U}_{\tilde{X}}$ a noise for $\tilde{X}'$.

We now observe that the support of $\tilde{X}' + \tilde{U}_{\tilde{X}}$ is the Minkowski sum of $\operatorname{supp}(\tilde{X}') \subseteq [\operatorname{ess\,inf}(\tilde{X}), \operatorname{ess\,sup}(\tilde{X})]$ and of $[0, l_X]$, so it is a closed interval of $\mathbb{R}$. Therefore, we can now define

$$\mathring{\rho}(\tilde{X}) := \rho(\tilde{X}' + \tilde{U}_{\tilde{X}}) - \rho(\tilde{U}_{\tilde{X}}) . \tag{26}$$

Note that by definition, $\mathring{\rho}$ is well defined by law-invariance of $\rho$ and it is itself a law-invariant risk measure on $L^\infty(\mathbb{R})$ (as a difference of two law-invariant risk measures) that coincides with $\rho$ on $L_c^\infty(\mathbb{R})$ by additive independence of $\rho$ on $L_c^\infty(\mathbb{R})$.

We also have, for any noise $\tilde{V} \in L^\infty(\Omega_{\mathsf{M}_0}, \mathcal{F}_{\mathsf{M}_0}, \mathbb{P}_{\mathsf{M}_0}; \mathbb{R})$ independent of $\tilde{X}'$ and $\tilde{U}_{\tilde{X}}$ with convex support that includes $[0, l_X]$,

$$\mathring{\rho}(\tilde{X}) = \rho(\tilde{X}' + \tilde{V}) - \rho(\tilde{V}) . \tag{27}$$

Indeed, $\tilde{U}_{\tilde{X}}, \tilde{V}, \tilde{X}' + \tilde{U}_{\tilde{X}}, \tilde{X}' + \tilde{V}$ and $\tilde{X}' + \tilde{U}_{\tilde{X}} + \tilde{V}$ all have convex support, so,

$$\begin{aligned}
\rho(\tilde{X}' + \tilde{V}) - \rho(\tilde{V}) &= \rho((\tilde{X}' + V) + \tilde{U}_{\tilde{X}}) - \rho(\tilde{U}_{\tilde{X}}) - \rho(\tilde{V}) \\
&= \rho((\tilde{X}' + \tilde{U}_{\tilde{X}}) + \tilde{V}) - \rho(\tilde{V}) - \rho(\tilde{U}_{\tilde{X}}) \\
&= \rho(\tilde{X}' + \tilde{U}_{\tilde{X}}) - \rho(\tilde{U}_{\tilde{X}}) \\
&= \mathring{\rho}(\tilde{X}) .
\end{aligned}$$

1. Let $\mathsf{M} \in \mathcal{M}$ and two independent random variables $\tilde{X}, \tilde{Y} \in L^\infty(\Omega_\mathsf{M}, \mathcal{F}_\mathsf{M}, \mathbb{P}_\mathsf{M}; \mathbb{R})$. There exists independent random variables $\tilde{X}', \tilde{Y}', \tilde{U}_{\tilde{X}}, \tilde{U}_{\tilde{Y}} \in L^\infty(\Omega_{\mathsf{M}_0}, \mathcal{F}_{\mathsf{M}_0}, \mathbb{P}_{\mathsf{M}_0}; \mathbb{R})$ such that $\mathcal{L}(\tilde{X}') = \mathcal{L}(\tilde{X})$, $\mathcal{L}(\tilde{Y}') = \mathcal{L}(\tilde{Y})$ and $\tilde{U}_{\tilde{X}}, \tilde{U}_{\tilde{Y}}$ are noises for $\tilde{X}'$ and $\tilde{Y}'$. We now have,

$$\mathring{\rho}(\tilde{X}) + \mathring{\rho}(\tilde{Y}) = \rho(\tilde{X}' + \tilde{U}_{\tilde{X}}) + \rho(\tilde{Y}' + \tilde{U}_{\tilde{Y}}) - \rho(\tilde{U}_{\tilde{X}}) - \rho(\tilde{U}_{\tilde{Y}})$$
$$= \rho(\tilde{X}' + \tilde{Y}' + \tilde{U}_{\tilde{X}} + \tilde{U}_{\tilde{Y}}) - \rho(\tilde{U}_{\tilde{X}} + \tilde{U}_{\tilde{Y}})$$
$$= \mathring{\rho}(\tilde{X} + \tilde{Y}) \,,$$

where the last equality comes from using Equation (27) because $\tilde{U}_{\tilde{X}} + \tilde{U}_{\tilde{Y}}$ has support exactly $[0, l_X + l_Y]$ and is independent of $\tilde{X} + \tilde{Y}$.

Therefore, $\mathring{\rho}$ is additive independent on $L^\infty(\mathbb{R})$.

2. Assume that $\rho$ is positive homogeneous on $L_c^\infty(\mathbb{R})$.

Then, for any $\tilde{X} \in L^\infty(\mathbb{R})$ and $\alpha \geqslant 0$, there exists independent $\tilde{X}', \tilde{U}_{\tilde{X}} \in L^\infty(\Omega_{\mathsf{M}_0}, \mathcal{F}_{\mathsf{M}_0}, \mathbb{P}_{\mathsf{M}_0}; \mathbb{R})$ such that $\mathcal{L}(\tilde{X}') = \mathcal{L}(\tilde{X})$ and $\tilde{U}_{\tilde{X}}$ is a noise for $\tilde{X}'$. We now have

$$\alpha\mathring{\rho}(\tilde{X}) = \alpha\rho(\tilde{X}' + \tilde{U}_{\tilde{X}}) - \alpha\rho(\tilde{U}_{\tilde{X}})$$
$$= \rho(\alpha\tilde{X}' + \alpha\tilde{U}_{\tilde{X}}) - \rho(\alpha\tilde{U}_{\tilde{X}})$$
$$= \mathring{\rho}(\alpha\tilde{X}) \,,$$

where the last equality comes from using Equation (27) because $\alpha\tilde{U}_{\tilde{X}}$ has support exactly $[0, \alpha l_X]$ and is independent of $\alpha\tilde{X}'$.

Therefore, $\mathring{\rho}$ is positive homogeneous on $L^\infty(\mathbb{R})$.

3. Assume that $\rho$ is monotone on $L_c^\infty(\mathbb{R})$.

Let $\mathsf{M} \in \mathcal{M}$ and two random variables $\tilde{X}, \tilde{Y} \in L^\infty(\Omega_\mathsf{M}, \mathcal{F}_\mathsf{M}, \mathbb{P}_\mathsf{M}; \mathbb{R})$ such that $\tilde{X} \geqslant \tilde{Y}$ almost surely. There exists random variables $\tilde{X}', \tilde{Y}', \tilde{U} \in L^\infty(\Omega_{\mathsf{M}_0}, \mathcal{F}_{\mathsf{M}_0}, \mathbb{P}_{\mathsf{M}_0}; \mathbb{R})$ such that $\mathcal{L}\left((\tilde{X}', \tilde{Y}')\right) = \mathcal{L}\left((\tilde{X}, \tilde{Y})\right)$, $\tilde{U} \sim \text{Unif}([0, \max(l_X, l_Y)])$ and $\tilde{U}$ is independent to both $\tilde{X}'$ and $\tilde{Y}'$. In particular, because their joint law is identical, we have $\tilde{X}' \geqslant \tilde{Y}'$ and adding $\tilde{U}$ we get $\tilde{X}' + \tilde{U} \geqslant \tilde{Y}' + \tilde{U}$ almost surely. As, $\tilde{X}' + \tilde{U}$ and $\tilde{Y}' + \tilde{U}$ have convex supports in $\mathbb{R}$ and $\mathring{\rho}$ coincides with $\rho$ on $L_c^\infty(\mathbb{R})$, by monotonicity of $\rho$ on $L_c^\infty(\mathbb{R})$, we have

$$\mathring{\rho}(\tilde{X}' + \tilde{U}) = \rho(\tilde{X}' + \tilde{U}) \geqslant \rho(\tilde{Y}' + \tilde{U}) = \mathring{\rho}(\tilde{Y}' + \tilde{U}) \,.$$

Now, subtracting by $\rho(\tilde{U})$ on both side and applying Equation (27), we get $\mathring{\rho}(\tilde{X}) \geqslant \mathring{\rho}(\tilde{Y})$.

Therefore, $\mathring{\rho}$ is monotone on $L^\infty(\mathbb{R})$.

$\square$

By Lemma H.6, we know that a version of Theorem H.4 holds for risk measures on $L_c^\infty(\mathbb{R})$, that is, if $\rho$ is a risk measure defined on $L_c^\infty(\mathbb{R})$ that is monotone, additive independent, coincides with the identity on constant random variable, and law-invariant, then there exists a unique Borel probability measure $\mu$ on $\overline{\mathbb{R}}$ such that

$$\forall \tilde{X} \in L_c^\infty(\mathbb{R}), \quad \rho(\tilde{X}) = \int_{\overline{\mathbb{R}}} \mathsf{ERM}_a(\tilde{X}) d\mu(a) \,.$$

We are now ready to prove Theorem 4.4.

*Proof of Theorem 4.4.* Let $\rho$ be a risk measure that is law-invariant, non-constant and monotone on $L_c^\infty(\mathbb{R})$ and that satisfies Condition 1 or Condition 2 in either the case of static kernels or resampled kernels.

By the conclusion of Proposition 4.3, $\rho$ coincides with the identity on constants, is positive homogeneous and additive independent on $L_c^\infty(\mathbb{R})$. Also, by monotonicity of $\rho$ on $L_c^\infty(\mathbb{R})$, using the multiplicativity statement proved as Item 4 of Proposition H.3, for any $\mathsf{M} \in \mathcal{M}$ and independent $\tilde{X}, \tilde{Y} \in L_c^\infty(\Omega_\mathsf{M}, \mathcal{F}_\mathsf{M}, \mathbb{P}_\mathsf{M}; \mathbb{R}_+)$, we have $\rho(\tilde{X}\tilde{Y}) = \rho(\tilde{X})\rho(\tilde{Y})$.

Then, using Lemma H.6, there exists $\mathring{\rho}$ a risk measure that is law-invariant on $L^\infty(\mathbb{R})$, coincides with $\rho$ on $L_c^\infty(\mathbb{R})$ and is positive homogeneous, additive independent and monotone on $L^\infty(\mathbb{R})$. In particular, $\mathring{\rho}$ also coincides with the identity on constants.

Let now $\mathsf{M}_0 \in \mathcal{M}$ an ambiguity-averse MDP such that $(\Omega_{\mathsf{M}_0}, \mathcal{F}_{\mathsf{M}_0}, \mathbb{P}_{\mathsf{M}_0})$ is non-atomic (for example the one of Example E.1). Because $\mathring{\rho}$ is monotone, additive independent and law invariant on $L^\infty(\mathbb{R})$ and it coincides with the identity on constants, the specific risk measure

$$\mathring{\rho}_{\mathsf{M}_0} : L^\infty(\Omega_{\mathsf{M}_0}, \mathcal{F}_{\mathsf{M}_0}, \mathbb{P}_{\mathsf{M}_0}; \mathbb{R}) \to \mathbb{R}$$
$$\tilde{X} \mapsto \mathring{\rho}(\tilde{X})$$

is a risk-measure on a non-atomic probability space that is law-invariant, monotone, additive independent, and that coincides with the identity on constants. Therefore, by Theorem 1 in Mu et al. (2024), there exists a unique Borel probability measure $\mu$ on $\overline{\mathbb{R}}$ such that

$$\forall \tilde{X} \in L^\infty(\Omega_{\mathsf{M}_0}, \mathcal{F}_{\mathsf{M}_0}, \mathbb{P}_{\mathsf{M}_0}; \mathbb{R}), \quad \mathring{\rho}_{\mathsf{M}_0}(\tilde{X}) = \int_{\overline{\mathbb{R}}} \mathsf{ERM}_a(\tilde{X}) d\mu(a) ,$$

where $\mathsf{ERM}_a(\tilde{X}) = \frac{1}{a} \log \mathbb{E}(\exp(a\tilde{X}))$ if $a \in \mathbb{R}^\star$, $\mathsf{ERM}_0(\tilde{X}) = \mathbb{E}(\tilde{X})$, $\mathsf{ERM}_{+\infty}(\tilde{X}) = \operatorname{ess\,sup}(\tilde{X})$ and $\mathsf{ERM}_{-\infty}(\tilde{X}) = \operatorname{ess\,inf}(\tilde{X})$.

Then, we fix $\lambda > 0$ and define the dilatation map $T_\lambda : \overline{\mathbb{R}} \to \overline{\mathbb{R}}$ such that $T_\lambda(a) = \lambda a$ if $a \in \mathbb{R}$ and $T_\lambda(+\infty) = +\infty$, $T_\lambda(-\infty) = -\infty$. Noting that for any $a \in \overline{\mathbb{R}}$ and $\tilde{X} \in L^\infty(\Omega_{\mathsf{M}_0}, \mathcal{F}_{\mathsf{M}_0}, \mathbb{P}_{\mathsf{M}_0}; \mathbb{R})$, we have $\mathsf{ERM}_a(\lambda\tilde{X}) = \lambda\mathsf{ERM}_{T_\lambda(a)}(\tilde{X})$, we obtain by positive homogeneity of $\mathring{\rho}$,

$$\forall \tilde{X} \in L^\infty(\Omega_{\mathsf{M}_0}, \mathcal{F}_{\mathsf{M}_0}, \mathbb{P}_{\mathsf{M}_0}; \mathbb{R}), \quad \mathring{\rho}_{\mathsf{M}_0}(\tilde{X}) = \frac{1}{\lambda}\mathring{\rho}_{\mathsf{M}_0}(\lambda\tilde{X}) = \frac{1}{\lambda} \int_{\overline{\mathbb{R}}} \mathsf{ERM}_a(\lambda\tilde{X}) d\mu(a)$$
$$= \int_{\overline{\mathbb{R}}} \mathsf{ERM}_{T_\lambda(a)}(\tilde{X}) d\mu(a)$$
$$= \int_{\overline{\mathbb{R}}} \mathsf{ERM}_a(\tilde{X}) d(T_\lambda \# \mu)(a) ,$$

where $T_\lambda \# \mu$ is the pushforward measure of $\mu$ by $T_\lambda$. By uniqueness of the representation of $\mathring{\rho}_{\mathsf{M}_0}$, we have $\mu = T_\lambda \# \mu$ for any $\lambda > 0$.

We now assume for contradiction that $\mu((0, +\infty)) > 0$. Then, there exists $0 < a_1 < a_2 < +\infty$ such that $\mu((a_1, a_2)) > 0$. Taking $\lambda > a_2/a_1$, we have that the intervals $(\lambda^n a_1, \lambda^n a_2)$ for $n \in \mathbb{N}$ are pairwise disjoint and each has the same positive measure under $\mu$, contradicting the finiteness of $\mu$.

The same argument leads to $\mu((-\infty, 0)) = 0$, finally showing that $\mu$ is supported on $\{0, -\infty, +\infty\}$.

Consequently, noting $a = \mu(\{0\})$, $b = \mu(\{-\infty\})$ and $c = \mu(\{+\infty\})$, we have $a + b + c = 1$ and

$$\forall \tilde{X} \in L^\infty(\Omega_{\mathsf{M}_0}, \mathcal{F}_{\mathsf{M}_0}, \mathbb{P}_{\mathsf{M}_0}; \mathbb{R}), \quad \mathring{\rho}_{\mathsf{M}_0}(\tilde{X}) = a \cdot \mathbb{E}(\tilde{X}) + b \cdot \operatorname{ess\,inf}(\tilde{X}) + c \cdot \operatorname{ess\,sup}(\tilde{X}) .$$

We now show that $a = 1, b = 1$ or $c = 1$ using that for any independent random variables $\tilde{X}', \tilde{Y}' \in L_c^\infty(\Omega_{\mathsf{M}_0}, \mathbb{P}_{\mathsf{M}_0}; \mathbb{R}_+)$,

$$\mathring{\rho}_{\mathsf{M}_0}(\tilde{Z}\tilde{X}) = \rho(\tilde{Z}\tilde{X}) = \rho(\tilde{Z})\rho(\tilde{X}) = \mathring{\rho}_{\mathsf{M}_0}(\tilde{Z})\mathring{\rho}_{\mathsf{M}_0}(\tilde{X}) .$$

By non-atomicity, for any $0 \leqslant x_1 \leqslant x_2$ and $0 \leqslant z_1 \leqslant z_2$, there exists independent random variables $\tilde{X}', \tilde{Z}' \in L_c^\infty(\Omega_{\mathsf{M}_0}, \mathcal{F}_{\mathsf{M}_0}, \mathbb{P}_{\mathsf{M}_0}; \mathbb{R})$ such that $\tilde{X} \sim \mathrm{Unif}([x_1, x_2])$ and $\tilde{Z} \sim \mathrm{Unif}([z_1, z_2])$. In particular, because $L_c^\infty(\Omega_{\mathsf{M}_0}, \mathcal{F}_{\mathsf{M}_0}, \mathbb{P}_{\mathsf{M}_0}; \mathbb{R}_+)$, we have $\mathring{\rho}(\tilde{X}\tilde{Z}) = \mathring{\rho}(\tilde{X})\mathring{\rho}(\tilde{Z})$.

First, noting $u = a/2 + b$ and $v = a/2 + c$, we have

$$\mathring{\rho}_{\mathsf{M}_0}(\tilde{X}) = a\frac{x_1 + x_2}{2} + bx_1 + cx_2 = ux_1 + vx_2 ,$$
$$\mathring{\rho}_{\mathsf{M}_0}(\tilde{Z}) = a\frac{z_1 + z_2}{2} + bz_1 + cz_2 = uz_1 + vz_2 ,$$
$$\mathring{\rho}_{\mathsf{M}_0}(\tilde{X}\tilde{Z}) = a\frac{x_1 z_1 + x_1 z_2 + x_2 z_1 + x_2 z_2}{4} + bx_1 z_1 + cx_2 z_2 .$$

so the equality $\mathring{\rho}_{M_0}(\tilde{X}\tilde{Z}) = \mathring{\rho}_{M_0}(\tilde{X})\mathring{\rho}_{M_0}(\tilde{Z})$ gives

$$(ux_1 + vx_2)(uz_1 + vz_2) = a\frac{x_1z_1 + x_1z_2 + x_2z_1 + x_2z_2}{4} + bx_1z_1 + cx_2z_2$$

$$u^2x_1z_1 + uv(x_1z_2 + x_2z_1) + v^2x_2z_2 = \left(u - \frac{a}{4}\right)x_1z_1 + \frac{a}{4}(x_1z_2 + x_2z_1) + \left(v - \frac{a}{4}\right)x_2z_2 \ .$$

Therefore, because this equality holds for any $0 \leqslant x_1 \leqslant x_2$ and $0 \leqslant z_1 \leqslant z_2$, we have the system of equations:

$$\begin{cases} u^2 = u - \dfrac{a}{4} \\ uv = \dfrac{a}{4} \\ v^2 = v - \dfrac{a}{4} \end{cases} \ .$$

From the first and third equations, we have that $u$ and $v$ are roots of the equation $X^2 - X + a/4 = 0$.

If they are equal, then $u = v = 1/2$ since $u + v = a + b + c = 1$. Then, the second equation gives $a = 1$.

If $u \neq v$, we immediately have $a < 1$ as $a = 1$ implies $u = v$. We now prove by contradiction that $a = 0$. In fact, if $a \neq 0$, we have $1 - a \in (0, 1)$ and noting $w = \min(u, v)$, we have that $w = \frac{1 - \sqrt{1-a}}{2} < \frac{a}{2}$ which contradicts the non-negativity of $b$ or $c$. Finally $a = 0$ so $u = b$ and $v = c$ are the two roots of $X^2 - X = 0$ and $\{b, c\} = \{0, 1\}$.

Because $a = 1, b = 1$ or $c = 1$, we now have

$$\begin{cases} \forall \tilde{X} \in L^\infty(\Omega_{M_0}, \mathcal{F}_{M_0}, \mathbb{P}_{M_0}; \mathbb{R}), \quad \mathring{\rho}_{M_0}(\tilde{X}) = \mathbb{E}(\tilde{X}) \\ \qquad\qquad\qquad\qquad \text{or} \\ \forall \tilde{X} \in L^\infty(\Omega_{M_0}, \mathcal{F}_{M_0}, \mathbb{P}_{M_0}; \mathbb{R}), \quad \mathring{\rho}_{M_0}(\tilde{X}) = \operatorname{ess\,inf}(\tilde{X}) \ , \\ \qquad\qquad\qquad\qquad \text{or} \\ \forall \tilde{X} \in L^\infty(\Omega_{M_0}, \mathcal{F}_{M_0}, \mathbb{P}_{M_0}; \mathbb{R}), \quad \mathring{\rho}_{M_0}(\tilde{X}) = \operatorname{ess\,sup}(\tilde{X}) \end{cases}$$

By law-invariance of $\mathring{\rho}$ on $L^\infty(\mathbb{R})$ and non-atomicity of $(\Omega_{M_0}, \mathcal{F}_{M_0}, \mathbb{P}_{M_0})$, it directly implies:

$$\begin{cases} \forall \tilde{X} \in L^\infty(\mathbb{R}), \quad \mathring{\rho}(\tilde{X}) = \mathbb{E}(\tilde{X}) \\ \qquad\qquad \text{or} \\ \forall \tilde{X} \in L^\infty(\mathbb{R}), \quad \mathring{\rho}(\tilde{X}) = \operatorname{ess\,inf}(\tilde{X}) \ , \\ \qquad\qquad \text{or} \\ \forall \tilde{X} \in L^\infty(\mathbb{R}), \quad \mathring{\rho}(\tilde{X}) = \operatorname{ess\,sup}(\tilde{X}) \end{cases}$$

so the desired result holds because $\rho$ and $\mathring{\rho}$ coincide on $L_c^\infty(\mathbb{R})$:

$$\begin{cases} \forall \tilde{X} \in L_c^\infty(\mathbb{R}), \quad \rho(\tilde{X}) = \mathbb{E}(\tilde{X}) \\ \qquad\qquad \text{or} \\ \forall \tilde{X} \in L_c^\infty(\mathbb{R}), \quad \rho(\tilde{X}) = \operatorname{ess\,inf}(\tilde{X}) \ . \\ \qquad\qquad \text{or} \\ \forall \tilde{X} \in L_c^\infty(\mathbb{R}), \quad \rho(\tilde{X}) = \operatorname{ess\,sup}(\tilde{X}) \end{cases}$$

$\square$

### H.3. $W^1$-continuity and Proof of Theorem 4.5

In this section we derive results for the case of $W^1$-continuous functionals. We first define $W^1$-continuity.

**Definition H.7.** The Wasserstein-1 metric $W^1(\nu_1, \nu_2)$ between two laws $\nu_1, \nu_2$ is defined as

$$W^1(\nu_1, \nu_2) = \int_0^1 |F_{\nu_1}^{-1}(u) - F_{\nu_2}^{-1}(u)| \, du \ .$$

A law-invariant risk measure $\rho(\tilde{X})$ can be written $\rho(\tilde{X}) = \varrho(\mathcal{L}(X))$ for some $\varrho$ that maps laws to scalars, see Proposition theorem B.3. With this notation, the risk measure $\rho$ is $W^1$-continuous if for any law $\nu$ with compact supports and any sequence of laws with compact supports $(\nu_n)_{n\in\mathbb{N}}$ we have

$$\lim_{n\to+\infty} W^1(\nu_n, \nu) = 0 \Rightarrow \lim_{n\to+\infty} \varrho(\nu_n) = \varrho(\nu) .$$

Intuitively, $W^1$-continuity corresponds to a continuity in the quantiles of the law. Table 2 summarizes the $W^1$-continuity properties of common risk measures. Theorem 4.5 provides a complete characterization of $W^1$-continuous, law-invariant risk measures that satisfy Condition 1 or Condition 2. We first prove that if $\rho$ is a $W^1$-continuous law-invariant risk measure that satisfies Condition 1 or Condition 2 in either the case of static kernels or resampled kernels, then $\rho = \mathbb{E}$.

*Proof of Theorem 4.5.* Let $\rho$ a risk measure that is non-constant, $W^1$-continuous and law-invariant on $L_c^\infty(\mathbb{R})$ and that satisfies Condition 1 or Condition 2 in either the case of static kernels or resampled kernels.

By the conclusion of Proposition 4.3, $\rho$ coincides with the identity on constants and is positive homogeneous and additive independent on $L_c^\infty(\mathbb{R})$.

Let $\mathsf{M}_0 \in \mathcal{M}$ an ambiguity-averse MDP such that $(\Omega_{\mathsf{M}_0}, \mathcal{F}_{\mathsf{M}_0}, \mathbb{P}_{\mathsf{M}_0})$ is non-atomic (for example the one of Example E.1). Now, for any $\tilde{X} \in L_c^\infty(\mathbb{R})$, consider a sequence of i.i.d. random variables $(\tilde{X}_i)_{i\in\mathbb{N}} \in L^\infty(\Omega_{\mathsf{M}_0}, \mathcal{F}_{\mathsf{M}_0}, \mathbb{P}_{\mathsf{M}_0}; \mathbb{R})^{\mathbb{N}}$ such that all the $\tilde{X}_i$ have the same law as $\tilde{X}$. By additive independence and positive homogeneity on $L_c^\infty(\mathbb{R})$, we have

$$\forall n \in \mathbb{N}, \quad \rho\left(\frac{1}{n}\sum_{i=1}^{n}\tilde{X}_i\right) = \frac{1}{n}\sum_{i\leq n}\rho(\tilde{X}_i) = \rho(\tilde{X}) .$$

Denoting by $\nu_n$ the law of the sample mean $\frac{1}{n}\sum_{i=1}^{n}\tilde{X}_i$ and by $\nu = \delta_{\mathbb{E}(\tilde{X})}$ the Dirac measure concentrated in $\mathbb{E}(\tilde{X})$, we now show that $W^1(\nu_n, \nu) \to_n 0$. Indeed,

$$\begin{aligned}
W^1(\nu_n, \nu) &= \int_0^1 \left| F_{\nu_n}^{-1}(u) - F_\nu^{-1}(u) \right| du \\
&= \int_0^1 \left| F_{\nu_n}^{-1}(u) - \mathbb{E}(\tilde{X}) \right| du \\
&= \mathbb{E}\left( \left| \frac{1}{n}\sum_{i=1}^{n}\tilde{X}_i - \mathbb{E}(\tilde{X}) \right| \right) ,
\end{aligned}$$

where we have used in first equality that $\nu$ is a Dirac measure and in the second equality that if $\tilde{U}$ is a random variable uniformly distributed on $(0,1)$, then $F_{\nu_n}^{-1}(\tilde{U})$ has the same distribution as $\frac{1}{n}\sum_{i=1}^{n}\tilde{X}_i$. Then, by boundedness of $\frac{1}{n}\sum_{i=1}^{n}\tilde{X}_i$, the Dominated Convergence Theorem gives

$$\lim_{n\to\infty} \mathbb{E}\left( \left| \frac{1}{n}\sum_{i=1}^{n}\tilde{X}_i - \mathbb{E}(\tilde{X}) \right| \right) = \mathbb{E}\left( \lim_{n\to\infty} \left| \frac{1}{n}\sum_{i=1}^{n}\tilde{X}_i - \mathbb{E}(\tilde{X}) \right| \right) = \mathbb{E}[0] = 0 ,$$

where the second equality comes from the fact that $\left| \frac{1}{n}\sum_{i=1}^{n}\tilde{X}_i - \mathbb{E}(\tilde{X}) \right| \to 0$ almost surely by the Strong Law of Large Numbers.

Therefore, using $W^1$-continuity,

$$\begin{aligned}
\rho(\tilde{X}) &= \lim_{n\to+\infty} \rho\left( \frac{1}{n}\sum_{i=1}^{n}\tilde{X}_i \right) \\
&= \rho\left( \lim_{n\to+\infty} \frac{1}{n}\sum_{i=1}^{n}\tilde{X}_i \right) \\
&= \rho(\mathbb{E}(\tilde{X})) = \mathbb{E}(\tilde{X})
\end{aligned}$$

where the last equality uses that $\rho(c) = c$ for $c \in \mathbb{R}$, as shown in Proposition 4.3. $\square$

We can now conclude by noting that it is known that $\rho = \mathbb{E}$ satisfies both Conditions 1 and 2 for resampled kernels (this corresponds to distributionally robust MDPs with a set of distribution equal to the singleton $\{\nu\}$ (Xu & Mannor, 2012)) but not for static kernels (as shown in Example 3.4-3.8).

## I. Extensions of Conditions 1 and 2 to the case of two risk measures

**Proposition I.1.** *Let $\rho_1, \rho_2$ be two risk measures for ambiguity-averse MDPs that are non-constant and law-invariant on $L_c^\infty(\mathbb{R})$ and such that one of the two conditions is satisfied in either the case of static kernels or resampled kernels:*

*1. $V^{\pi,\nu,\rho_1}$ is a fixed point of $T^{\pi,\nu,\rho_2}$ for any instance $\mathsf{M}_\nu \in \mathcal{M}_c$ and stationary policy $\pi \in \Pi_\mathsf{S}$, i.e.,*

$$V^{\pi,\nu,\rho_1} = T^{\pi,\nu,\rho_2} V^{\pi,\nu,\rho_1}, \quad \forall \mathsf{M}_\nu \in \mathcal{M}_c, \forall \pi \in \Pi_\mathsf{S} \ .$$

*2. $V^{\nu,\rho_1}$ is a fixed point of $T^{\nu,\rho_2}$ for any instance $\mathsf{M}_\nu \in \mathcal{M}_c$, i.e.,*

$$V^{\nu,\rho_1} = T^{\nu,\rho_2} V^{\nu,\rho_1}, \quad \forall \mathsf{M}_\nu \in \mathcal{M}_c \ .$$

*Then $\forall X \in L_c^\infty(\mathbb{R}), \rho_1(X) = \rho_2(X)$.*

*Proof.* We first show the result for static kernels. We heavily use the intermediary results of the proof of Theorem 4.4 here (see Appendix H). Keeping the same notations and random variables as in Lemma H.1, the MDP of Figure 3 and law-invariance lead to the following system for any $\gamma \in (0,1), \mathsf{M} \in \mathcal{M}$, independent random variables $(\tilde{X}, \tilde{Y}, \tilde{Z}) \in L_c^\infty(\Omega_\mathsf{M}, \mathcal{F}_\mathsf{M}, \mathbb{P}_\mathsf{M}; \mathbb{R})^2 \times L_c^\infty(\Omega_\mathsf{M}, \mathcal{F}_\mathsf{M}, \mathbb{P}_\mathsf{M}; [0,1])$ and $(a,c) \in \mathbb{R} \times \mathbb{R}^\star$ such that $\frac{\tilde{X}-a}{c} \in [0,1]$ almost surely,

$$\rho_1\left(\gamma\left(\tilde{Z}\tilde{X} + (1-\tilde{Z})\tilde{Y}\right)\right) = \rho_2\left(\gamma\left(\tilde{Z}\rho_1(\tilde{X}) + (1-\tilde{Z})\rho_1(\tilde{Y})\right)\right) , \tag{28}$$

$$\rho_1\left(\tilde{X}\right) = \rho_2\left(a + \gamma\left(\frac{\tilde{X}-a}{c}\rho_1\left(\gamma^{-1}c\right) + (1-\frac{\tilde{X}-a}{c})\rho_1(0)\right)\right) , \tag{29}$$

$$\rho_1\left(\gamma^{-1}c\right) = \rho_2\left(\gamma^{-1}c + \gamma\rho_1(0)\right) . \tag{30}$$

As in the beginning of the proof of Theorem 4.3, $\rho_1$ being non constant on $L_c^\infty(\mathbb{R})$ imposes $\rho_2$ to be non constant on $\mathbb{R}$ through (28). Consequently, (30) imposes $\rho_1(0) = 0$ and that $\forall x \in \mathbb{R}, \rho_1(x) = \rho_2(x)$. Then, by Lemma H.2, $\forall x \in \mathbb{R}, \rho_1(x) = \rho_2(x) = x$ and Equation (29) gives the desired result.

To prove the results for the case of resampled kernels, we can proceed similarly and use the same MDP instance as in Figure 3 and use law-invariance. $\qquad\square$

