# OpenReview forum: "Dynamic Programming for Epistemic Uncertainty in Markov Decision Processes"
_ICML.cc/2026/Conference — ICML 2026 spotlight_

### Official Review · Reviewer_XpK6 · 2026-03-05

**Soundness:** 2
**Presentation:** 2
**Significance:** 3
**Originality:** 3
**Overall Recommendation:** 5
**Confidence:** 3

**Summary:**

The paper deals with models of decision making under both aleatoric and epistemic uncertainty. It introduces a notion of ambiguity-averse MDPs, in which the transition probabilities are random variables and the player aims to optimize some risk measure over state values in a Markov chain induced by the policy. The paper shows that an appropriate choice of risk measure can recover many standard MDP models of epistemic uncertainty, such as RMDPs or multi-model MDPs. The key question addressed by the paper is the following: for which models captured by the ambiguity-averse MDPs are the values of policies (incl. optimal values) the fixed points of standard state-wise Bellman equation? (Without any form of state augmentation?) For which of them can the solution to the Bellman equation be computed by value and policy iteration?

The paper shows that (under appropriate and mild conditions on the risk measure) the algorithmic solution exists whenever the model admits the Bellman equations. Crucially, it shown that (again depending on the properties of the risk measure and whether static or dynamic kernels are considered), the only ambiguity-averse conformant models admitting the Bellman equations are the worst-case and best-case RMDPs and nominal MDPs (without epistemic uncertainty). The key novelty is the negative result, since Bellman the existence equations for RMDPs is known. The result rests on the recently published representation theorem for relevant risk measures, which shows that each such measure can be decomposed into a mixture of entropic risk measures.

**Compliance With Llm Reviewing Policy:**

Affirmed.

**Final Justification:**

I thank the authors for the clarifying discussion. I continue to support the acceptance of the paper.

**Key Questions For Authors:**

[Q1] Please comment on the relationship between MEMDPs and MM-MDPs.

[Q2] l. 1858: Why does monotonicity imply \rho(XY)=\rho(X)\rho(Y) for X,Y independent?

**Limitations:**

Yes.

**Strengths And Weaknesses:**

**Significance**
Decision models with epistemic uncertainty are currently a hot topic in planning and formal methods communities and play an increasingly important role in RL, particularly in the domain of safe RL. While the result is of mostly theoretical interest (the lack of state-wise Bellman equations for many relevant domains and the adoption of state-augmentation techniques are widely accepted, particularly in formal methods), having a strict impossibility result is nonetheless interesting.

**Presentation**
The paper is rather well written, it is clear what the results are and why they are important. The paper nicely presents the context of the work and the background on models it aims to unify.

What is a bit unfortunate is that for a theory paper, there is little in terms of proof intuitions. Indeed, the main results are only presented in the penultimate two pages. I understand that the proofs are somewhat dense and cannot fit the main text, but in its current form, the appendix looks more like a separate paper than a support for the main text. I believe that, e.g., the proof of of the key Theorem 4.4 could be distilled into a sketch of its main ideas (use the representation theorem -> dilation trick to show that the mixing measure is zero almost everywhere -> analyze uniform random variables to obtain equations showing that the distribution is Dirac on 0/min/max).

**Soundness**
The mathematical formalization is sound. I checked the proof of Theorem 4.4 (without its supporting Lemmas) and it seemed sound, modulo some questions provided below. I am reasonably confident about the validity of results, though I did not have the time capacity to verify all the details. In this sense, the results seem more suitable for a journal publication or a theory/formal method conference.

**Originality**
I am not aware of any previous impossibility results of this kind in the context of epistemic-aware models. The results seem novel and the methods interesting.

**Additional comments**
- I would suggest renaming “ambiguity-averse” to “ambiguity-aware”. We are not strictly speaking using the model to reduce ambiguity about the environment, just to handle such ambiguity.
- Regarding multi-model MDPs: In the formal methods community, there is a similar formalism called “Multi-Environment MDPs (MEMDPs)”, studied since 2010s. The main difference is that MEMDP formulations typically do not assume prior weights over the environments, rather, the environment choice is adversarial (which makes the formalism similar to non-rectangular RMDPs with static kernels, though here the uncertainty sets are non-convex and, indeed, discontinuous). However, MEMDP semantics with priors over the environments were also studied. It would be good to incorporate this formalism into the overview and discuss its relationship to MM-MDPs.

**Minor**
- l. 1902: there seem to be extra primes in X,Y
- l. 1908: The sentence “In particular…” seems incomplete

---

> ### Author Rebuttal · Authors · 2026-03-30
>
> We thank the reviewer for their positive evaluation and for finding our mathematical formalization and framework sound and well-presented. We also appreciate their encouraging feedback. We answer the remarks and remaining questions below, and we will revise our manuscript accordingly :
>
> ### Key Ideas of the proofs
>
> Following your suggestion to provide better accessibility to our technical results, we will expand the proof sketches in the revised manuscript to offer more high-level intuition. Specifically, we will add the three stage logic description for bridging Proposition 4.3 and Theorem 4.4 that you have outlined. We will also expand on the intuition for the proof of Theorem 4.5 (impossibility results for W1-continuous).
>
> ### Multiple-Environment MDPs and Multi-model MDPs
>
> Thanks for raising this interesting connection.
>
> As you mentioned, when MEMDPs [1] operate with a fully adversarial environment choice (no prior weights), the framework is structurally similar to non-rectangular Robust MDPs with static kernels (with non-convex distributions). Indeed, the objective in [1] is to guarantee performances against all possible environments, exactly as in Robust MDPs.
>
> When the MEMDP assumes a true, known distribution over the possible environments [2,3], it is related to Multi-Model MDP with static kernels [4,5] and concurrent MDPs [6,7]. Following your suggestion, we will use the extra space to provide more discussion on this model in the main body.
>
>
> ### Details on l.1858
>
> The fact that monotonicity implies homogeneity here ($ \rho(XY)=\rho(X)\rho(Y)$ for $X,Y$ independent and non-negative) follows from the fourth point of Proposition H.3 (which is a slightly modified statement of Proposition 4.3 through the addition of this fourth point). The specific part of the proof is at l.1724-1734.
>
> The simple idea behind this fourth point is the following : assume that $X,Y$ is non-negative and that $\rho$ monotone. Then $\rho(X)\geq0$. By positive homogeneity we get that $\rho(XY)=\rho(\rho(X)Y)=\rho(X)\rho(Y)$.
>
> Accordingly, we will add a reference to the fourth point of Proposition H.3 at l.1858.
>
> ## Conclusion:
> We hope that we addressed your main comments. We will improve the paper based on your inputs. We are looking forward to your new feedback and evaluation/score, and we remain at your disposal through the rebuttal process in case you have any other questions.
>
> [1] Raskin, Jean-François, and Ocan Sankur. Multiple-environment Markov decision processes. arXiv preprint arXiv:1405.4733. 2014
>
> [2] Chatterjee, K., Chmelík, M., Karkhanis, D., Novotný, P., and Royer, A. Multiple-environment Markov decision processes: Efficient analysis and applications. In Proceedings of the International Conference on Automated Planning and Scheduling (Vol. 30, pp. 48-56). 2020
>
> [3] Bordais, Benjamin, and Jean-François Raskin. Multi-Environment MDPs with Prior and Universal Semantics. arXiv preprint arXiv:2602.10938. 2026
>
> [4] Le Tallec, Y. Robust, risk-sensitive, and data-driven control of Markov decision processes. PhD thesis, Massachusetts Institute of Technology, 2007.
>
> [5] Steimle, L. N., Kaufman, D. L., and Denton, B. T. Multimodel Markov decision processes. IISE Transactions, 53 (10):1124–1139, 2021.
>
> [6] Buchholz, P. and Scheftelowitsch, D. Computation of weighted sums of rewards for concurrent MDPs. Mathematical Methods of Operations Research, 89(1):1–42, 2019.
>
> [7] Buchholz, P. and Scheftelowitsch, D. Concurrent MDPs with finite Markovian policies. In International Conference on Measurement, Modelling and Evaluation of Computing Systems, pp. 37–53. Springer, 2020.

---

> > ### Author Rebuttal · Reviewer_XpK6 · 2026-04-02
> >
> > I thank the authors for their response. My follow-up question is relatively minor: Do the MEMDPs (either with prior or adversarial semantics) fall in the scope of your impossibility result? (For adversarial - RMDP - semantics, you assume convexity on line 214). I think both a positive and negative answer to this could be pointed out in the paper. There are no known DP approaches to MEMDPs, so a positive answer would resolve the question whereas a negative one would point out a possible direction for future work.

---

> > > ### Author Response · Authors · 2026-04-03
> > >
> > > Thanks for the follow-up question. The class of MEMDPs, both with prior and adversarial semantics, falls under the impossibility results from Section 4.2 (Theorem 4.4 and Theorem 4.5).
> > > * MEMDPs with prior $\nu$ over some set of transition probabilities can be modeled as ambiguity-averse MDPs using $\rho = \mathbb{E}^{\nu}$. This construction is made explicit in Section 3.1 and Appendix C for multi-model MDPs ; we will mention explicitly that it still holds for MEMDPs with prior semantic.
> > > * MEMDPs with adversarial semantic (adversarial selection of the transition probabilities in an uncertainty set $\mathcal{P}$) also fall under our impossibility results, in the following sense:
> > >   - First, the fact that robust MDPs can be written as ambiguity-averse MDPs is always true even if we do not assume convex compact uncertainty set (we do not need this assumption in our reformulation Section 3.1 and Appendix C). However, if $\mathcal{P}$ is not convex, then Condition 1 and Condition 2 may not hold in the first place, and this is already known (see e.g. Wiesemann et al. 2013 or Si et al. 2023). This is why we used the convexity assumption in C1 and C2.
> > >   - Second, Additionally, assume that one were to write some generalizations of Condition 1 and Condition 2 to general (not necessarily convex) uncertainty sets, and let us call these new conditions as C1’ (“DP for policy evaluation holds across ambiguity-averse MDP instances with general uncertainty set $\mathcal{P}={\sf supp}(\nu)$”) and C2’  (“DP for Bellman optimality equations holds across ambiguity-averse MDP instances with general uncertainty set $\mathcal{P}={\sf supp}(\nu)$”). Then in particular C1’ and C2’ would imply that DP holds for *convex* uncertainty sets ${\sf supp}(\nu)$, (since C1’ and C2’ assume DP for all ${\sf supp}(\nu)$), hence our Condition 1 and Condition 2 hold, and our impossibility results (as written in section 4.2) apply verbatim.
> > >
> > > #### Conclusion :
> > >
> > > We hope that we have addressed your question above and we thank you for taking the time to read our response. In the revised manuscript we will significantly extend on the connection with MEMDPs. In the course of writing this paper, we did our best to review the (many!) variants of epistemic uncertainty MDPs existing in the literature and to provide a unifying perspective and a rigorous framework to study them together; we believe that such a complete picture on the topic was missing in the literature, and we believe that this is also one of our main contributions.

---

### Official Review · Reviewer_gA9D · 2026-03-09

**Soundness:** 4
**Presentation:** 3
**Significance:** 4
**Originality:** 4
**Overall Recommendation:** 5
**Confidence:** 3

**Summary:**

This paper studies MDPs under epistemic uncertainty where the transition kernel is modeled as a random variable drawn from a distribution. Policies are evaluated using a risk functional applied to the resulting return distribution. The paper characterizes when dynamic programming remains valid under this formulation and identifies classes of risk functionals that are compatible with Bellman-type recursions. The framework also connects several existing models of uncertainty in MDPs, including Bayesian and robust formulations.

**Compliance With Llm Reviewing Policy:**

Affirmed.

**Key Questions For Authors:**

### Questions for the Authors

1. Practical implications of the framework

The paper provides a clear theoretical characterization of risk functionals that admit Bellman-type recursions. However, the practical advantages of this formulation relative to existing uncertainty-aware MDP models remain somewhat unclear.

Could the authors elaborate on scenarios where this framework would lead to different policies or decisions compared to classical Bayesian or robust MDP formulations? Clarifying such situations would help better understand the practical relevance of the theory.


2. Interpretation of the transition resampling assumption

The framework assumes that the transition kernel is resampled from the distribution at each step, which differs from the classical Bayesian MDP setting where the kernel is fixed but unknown.

Could the authors clarify how this modeling assumption should be interpreted in practice, and whether it leads to qualitatively different policies compared to the classical Bayesian formulation? A clearer discussion of this distinction would improve the interpretability of the framework.

**Limitations:**

yes

**Strengths And Weaknesses:**

## Strengths

**1. Clean theoretical reframing**

The paper introduces a simple but effective reframing of epistemic uncertainty by modeling transition probabilities as random variables. This leads to a clean analytical structure and enables strong theoretical results characterizing which risk functionals are compatible with dynamic programming. In particular, the work clearly separates risk functionals that admit Bellman recursions from those that do not. The theoretical development is well organized and preserves compatibility with classical DP algorithms such as value iteration.

**2. Unification of uncertainty-aware MDP models**

The proposed framework provides a unified perspective on several existing approaches to uncertainty in MDPs. With appropriate choices of the transition kernel distribution, the formulation recovers Bayesian MDPs, deterministic kernels (classical MDPs), and connections to robust MDP formulations. This unification helps clarify the relationships among these models within a single framework.

---

## Mild Concerns and Questions

**1. Practical impact**

While the theoretical characterization is clear and interesting, the practical implications are somewhat less evident. The paper focuses primarily on structural results, and it would be helpful to further clarify the practical advantages of the framework compared with existing uncertainty-aware MDP formulations.

**2. Modeling assumption**

The framework assumes that the transition kernel is resampled from the distribution at each step, which differs from the classical Bayesian MDP setting where the kernel is fixed but unknown. It would be useful to discuss more explicitly how this modeling choice affects the interpretation of the resulting policies and whether it leads to differences relative to the classical Bayesian formulation.

---

> ### Author Rebuttal · Authors · 2026-03-30
>
> We thank the reviewer for their positive evaluation and their appreciation of our theoretical reframing and unification of MDP under parameter uncertainty models. We also value your constructive questions regarding the practical implications of our work and the nuances of our modeling assumptions and we will revise our manuscript accordingly :
>
> ### Practical Implications
>
> While the main focus of our work is methodological, we would like to emphasize that our contributions include some actionable takeaways for the community. In particular, on a practical level, our strong impossibility results definitively prove that one cannot formulate standard dynamic programming equations for law-invariant risk measures other than the expectation, essential infimum, and essential supremum. Therefore, this deters any further efforts toward obtaining standard Bellman equations for other risk measures. Consequently, it forces and directs practical algorithmic development toward alternatives, such as state augmentation [1] or direct policy gradient methods [2]. Furthermore, the general framework we define can be reused as a foundation to develop these non-DP methods.
>
> We will rewrite part of the introduction and conclusion to better emphasize the practical aspects of our work.
>
> ### Static/Resampled kernels
>
> We would like to clarify that our framework does model the setting where the transition kernel is fixed but unknown. This is precisely what we define and analyze as the **static** kernel model (where the kernel is sampled once at $t=0$) ; we refer to Definition 3.2 in the manuscript. Note that our main impossibility results (Theorem 4.4) explicitly distinguish between the case of static kernels and resampled kernels.
>
>
> ###  Relation with Bayesian MDPs
>
> We answer both your questions about Bayesian MDPs here.
>
> The fundamental difference between our static ambiguity-averse MDPs and classical Bayesian risk approaches [3,4] lies in how the objective is evaluated. Classical Bayesian models often evaluate uncertainty step-by-step, linking trajectory randomness (aleatoric risk) and parameter estimator randomness at each transition. In fact, the parameter randomness in Bayesian MDP only arises from the fact that the fixed parameter P is unknown and being estimated through realisations of trajectories (aleatoric risk). In contrast, our approach strictly isolates epistemic uncertainty: we first take the standard expectation over the random trajectories to evaluate the return, and then apply the risk measure to the distribution of the transition parameters.
>
> Our model yields policies that are structurally distinct from classical Bayesian MDPs [3,4]. While Bayesian approaches typically average out parameter estimators uncertainty to focus on trajectory randomness, an optimal policy in our framework explicitly hedges against the epistemic uncertainty of the transition kernels.
>
> We agree that discussing this distinction explicitly will improve the interpretability of the framework and will add more discussion on the relation between our framework and Bayesian MDPs in the revised manuscript.
>
> ### Relation with robust MDPs
>
> Because we treat the transition kernel $\tilde{P}$ as a random variable, our model is a direct generalization of robust MDPs. This is proved in Proposition C.2 (static) and Proposition C.3 (resampled) ; we will emphasize this more in the main body.
>
> ## Conclusion
>
> We hope that we addressed your main comments. We will improve the paper based on your inputs. We are looking forward to your new feedback and evaluation/score, and we remain at your disposal through the rebuttal process in case you have any other questions.
>
> [1] Chow, Y., Tamar, A., Mannor, S., and Pavone, M. Risk sensitive and robust decision-making: a CVAR optimization approach. Advances in Neural Information Processing Systems, 28, 2015.
>
> [2] Wang, Y. and Zou, S. Policy gradient method for robust reinforcement learning. In International Conference on Machine Learning, pp. 23484–23526. PMLR, 2022.
>
> [3] Lin, Y., Ren, Y., and Zhou, E. Bayesian risk Markov decision processes. Advances in Neural Information Processing Systems, 35:17430–17442, 2022.
>
> [4] Shapiro, A., Zhou, E., Lin, Y., and Wang, Y.. Episodic Bayesian optimal control with unknown randomness distributions. Operations Research. 2025

---

> > ### Author Rebuttal · Reviewer_gA9D · 2026-04-03
> >
> > The authors have adequately addressed my concerns during the rebuttal.
> > My overall assessment remains unchanged, and I support acceptance.

---

### Official Review · Reviewer_jck2 · 2026-03-11

**Soundness:** 4
**Presentation:** 4
**Significance:** 4
**Originality:** 3
**Overall Recommendation:** 5
**Confidence:** 4

**Summary:**

This paper present a unified framework for MDPs with uncertain transition probabilities where the transition are modeled as random variables and policies are evaluated via risk measures applied to return. They establishes conditions were dynamic programming principles hold.

**Compliance With Llm Reviewing Policy:**

Affirmed.

**Key Questions For Authors:**

One natural question would be: can you say anything about approximate DP, where you guarantee only error on the Bellman equation?

**Limitations:**

yes

**Strengths And Weaknesses:**

The work provide a comprehensive framework with rigorous derivation of relatively expected results.  Theorems 4.4 and 4.5 are strong impossibility results. However, to me they underline more that the "law invariance" in this setting can be too important, as there are alternative ways (like Ruszczynski nested risk measure framework).

The main weakness might be originality in some sense. To me the results are not so surprising, and there are multiple work by A. Shapiro that foreshadow (but do not completely cover) what is presented here. (As an aside there might be some missing reference on that. Unfortunately I can't pinpoint them precisely at the moment, but have for sure see him present results claiming that only essup / expectation are working - maybe in Risk-averse formulations of Stochastic Optimal Control and Markov Decision Processes).

On the other hand it is a good unifying framework paper that helps toward clarifying the field.

Small pet peeve of mine: I suggest to not use the CVAR definition in a setting where you may want to discuss conditional-Conditional Value at Risk. There are alternative name in the litterature (Average VaR, Tail VaR, Superquantile...) that escape this double use of conditional.

---

> ### Author Rebuttal · Authors · 2026-03-30
>
> We thank the reviewer for their time in reviewing our paper and for highlighting that our work provides a rigorous comprehensive framework with strong impossibility results. We also appreciate the reviewer’s constructive feedback, which we answer below :
>
> ### Beyond Law-invariance
>
> We completely agree that dropping the law-invariance assumption likely leads to alternative risk measures, such as nested risk measures [1]. To our knowledge, nested risk-measures have only been defined in the **risk-averse setting** in which they recover dynamic programming equations by design (by repeatedly applying the risk measure backward through time), whereas our paper focuses on the **ambiguity-averse** setting. As mentioned in the paper, we believe such methods are a promising direction to go beyond law-invariance in the ambiguity-averse setting.
>
> While exploring non-law-invariant measures beyond the nested case is an interesting direction, how to construct a formal and proper framework for such risk-measures is unclear. Specifically, for an arbitrary non-nested, non-law-invariant measure, it is not immediately clear how to define a well-posed Bellman operator (or even how to estimate it). Similarly, we are not aware of how one would define an optimal policy for such a scenario (as the value of policies would depend on the exact realisation of the universe).
> We highlight that our current framework already covers many practically relevant models, and we leave for future work going beyond non-law-invariant risk-measures.
>
> Covering the full generic possible risk-measures is a challenging interesting question and we will better highlight that in the conclusion.
>
> ### Originality and links with Shapiro’s work
>
> We thank you for bringing up this point, as it allows us to clarify a crucial distinction. Given the nature of Shapiro’s main seminal contributions, we believe it is highly likely that the results you are recalling belong to the risk-averse MDP literature [2,3]. In contrast, we would like to emphasize our work focuses on the *ambiguity-averse setting*.
>
> There is a fundamental difference between these two problems:
> - Risk-Averse MDPs: In this setting, the transition probabilities and rewards are fixed and known, and the decision-maker uses a risk measure to hedge against the variability of the return over the random trajectories (aleatoric risk). In this context, time-consistency essentially forces the risk measure to collapse to the entropic risk measure or the expectation [2].
> - Ambiguity-Averse MDPs (Our framework): In our model, we first take the standard expectation over the trajectories, and then we apply the risk measure to hedge against the uncertainty in the transition parameters themselves (epistemic risk).
> The DP in our setting emerge from the interaction between the risk measure, the expectation operator (over trajectories), and the discount factor.
>
> Consequently, our impossibility results are mathematically distinct from the time-consistency results in risk-averse control [2]. We will discuss in more detail the existing work on time-consistency for risk-averse MDPs directly in the discussion section (Section 4) and the conclusion. We are also happy to include in the manuscript more discussion on some references from Shapiro that the reviewer can provide.
>
> ### CVaR definition
>
> We understand your concern regarding the potentially confusing terminologies "conditional risk measures"  vs. “conditional value at risk”. We will add a paragraph in Section 2 (when we introduce risk measures) to make sure this distinction is explicit to the reader.
>
> ### Approximate DP
>
> We thank the reviewer for raising this natural question. We believe that our results on the convergence rates of value iteration and policy iteration can be generalized to handle the case of approximate evaluation of the Bellman operators, similarly as in the analysis from the seminal paper on approximate policy iteration for MDPs [4]. In particular, the analysis from [4] only uses the properties of the Bellman operator that we recover in our Proposition 3.7. We will add a discussion on this in section 4.
>
> ## Conclusion
> We hope that we addressed your main comments, will improve the paper based on your inputs and are looking forward to your new feedback and evaluation/score. We remain at your disposal through the rebuttal process.
>
> [1] Ruszczyński, A. Risk-averse dynamic programming for Markov decision processes. Mathematical programming, 125(2):235–261, 2010.
>
> [2] Shapiro, A. Time consistency of dynamic risk measures. Operations Research Letters, 0(6):436–439, November 2012. ISSN 0167-6377. doi: 10.1016/j.orl.2012.08.007.
>
> [3] Shapiro, A. and Li, Y. Risk-averse formulations of Stochastic Optimal Control and Markov Decision Processes. arXiv preprint arXiv:2505.16651, 2025
>
> [4] Scherrer, B. et al. (2015). Approximate modified policy iteration and its application to the game of Tetris. The Journal of Machine Learning Research, 16(1), 1629-1676.

---

> > ### Author Rebuttal · Reviewer_jck2 · 2026-04-03
> >
> > The authors have adequately addressed my concerns during the rebuttal. My overall assessment remains unchanged, and I support acceptance.

---

### Official Review · Reviewer_gTcm · 2026-03-12

**Soundness:** 4
**Presentation:** 4
**Significance:** 3
**Originality:** 3
**Overall Recommendation:** 5
**Confidence:** 4

**Summary:**

The paper presents an “ambiguity-averse” MDP model that treats the transition probabilities as random variables and measures the resulting random return via risk measures, and provides dynamic programming (DP)-based analyses. This class encompasses many robust-type and percentile optimization models. The authors show that for law-invariant risk measures, the DP principles require the risk measures to be the essential infimum or essential supremum or the expectation.

**Compliance With Llm Reviewing Policy:**

Affirmed.

**Final Justification:**

The authors have adequately addressed my questions. I support the acceptance of this work.

**Key Questions For Authors:**

1. To what extent, can the impossibility result (Theorem 4.5) be relaxed if one considers a more general risk measure without law-invariance?

2. To what extent, can the theory be generalized? What parts of the theory remain valid when considering more general measurable state/action spaces or average cost (reward) criteria? If generalization is not possible, what is the main bottleneck? A discussion on the potential for or limits of such extensions would be beneficial for the readers.

**Limitations:**

Yes

**Strengths And Weaknesses:**

Strengths
- The paper provides an elegant formulation and theory, offering a unified perspective on various robust MDP-type models and results.
- The authors identify useful necessary conditions for law-invariant risk measures to satisfy the DP principles (Proposition 4.3) and provide further characterizations (Theorem 4.4 and 4.5). Notably, they establish the equivalence between the validity of Condition 1 (DP for policy evaluation) and Condition 2 (DP for policy optimization) as well as the impossibility result (Theorem 4.5) under law-invariance.
- The presentation is clear and careful, despite a few minor typos (e.g., "iteration" on p. 2 should be corrected to "value iteration").

Weaknesses
- The algorithms and the convergence rate (Proposition 4.2) are somewhat standard and offer limited new insight. A discussion on tractability relative to various risk measures (beyond Appendix G) would be more useful.
- (Very minor) The illustrative examples were not particularly useful. Presenting the key ideas behind the proofs instead might be more helpful to the readers.

---

> ### Author Rebuttal · Authors · 2026-03-30
>
> We thank the reviewer for taking the time to review our paper, for their positive evaluation, and for recognizing our formulation as an elegant and unifying perspective on various MDP models. We also appreciate the reviewer’s constructive feedback. We answer your remarks and questions below, and we will revise our manuscript accordingly :
>
> ### Tractability
>
> We agree that emphasizing computational tractability across various MDP models adds valuable insights to the paper. Following your suggestion, we will update the main body with a discussion on these aspects and enrich Appendix G with specific complexity bounds and relevant methods to estimate the Bellman operator.
>
> ### Key Ideas of the proofs
>
> We also believe that the proof techniques behind Theorem 4.4 are novel in the field of MDP / ambiguity-averse MDPs. To provide better accessibility to our technical results, we will expand the proof sketches in the revised manuscript to offer more high-level intuition. Specifically, we will detail the three-stage logic in the path from Proposition 4.3 to Theorem 4.4 (invoking the representation Theorem, using the Dilatation trick and plugging in uniform random variables to obtain 2nd degree polynomials equalities that lead to the fact that $\mu$ is a Dirac on either $+\infty, -\infty$ or $0$).
>
> ### Beyond Law-Invariance
>
> The impossibility result relies fundamentally on the law-invariance assumption and does not hold without it. In the risk-averse setting (with P fixed and known), a primary counterexample of non-law-invariant risk measures that satisfy *risk-averse* DP is the class of nested risk measures [1], which can directly satisfy risk-averse DP by design (by repeatedly applying the risk measure backward through time). As mentioned in the paper, we believe such methods are a promising direction to go beyond law-invariance *in the ambiguity-averse setting*.
>
> While exploring non-law-invariant measures beyond the nested case is an interesting direction, how to construct a formal and proper framework for such risk-measures is unclear. Specifically, for an arbitrary non-nested, non-law-invariant measure, it is not immediately clear how to define a well-posed Bellman operator (or even how to estimate it). Similarly, we are not aware of how one would define an optimal policy for such a scenario (as the value of policies would depend on the exact realisation of the universe).
>
> We highlight that our current framework already covers many practically relevant models, and we leave for future work going beyond non-law-invariant risk-measures.
>
> Covering the full generic possible risk-measures is a challenging interesting question and we will better highlight that in the conclusion.
>
> ### Extensions
> **Measurable State/Action Spaces**: Our main impossibility result (Theorem 4.4) extends to the case of more general infinite state/action spaces in the following sense. If one asks that Condition 1 and Condition 2 hold for all MDP instances with measurable state and action spaces, then in particular Condition 1 and Condition 2 hold for all MDP instances with *finite* state and action spaces, and our main impossibility result (Theorem 4.4) can be applied verbatim.
>
> **Average/Total Return Criteria**: We believe our main results could likely be extended to average or total cost criteria.
> However, analyzing these criteria would require a different technical setting: one would first need to rigorously define the set of MDPs for which the average reward is well-defined. Additionally, the optimality equations for the average return criterion are fundamentally different from the optimality equations for the discounted return. Indeed, for the average return, in general (multichain) instances the optimality equations consist in a pair of equations involving the gain vector and the bias vector, and one would need to rewrite Condition 1 - Condition 2 (from our paper) accordingly. We will add a short discussion on this and list it as a promising next step.
>
> We view these two extensions as a highly promising direction for future work and we will add a discussion outlining the potential feasibility of these extensions in the main body.
>
>
> ## Conclusion
> We hope that we addressed your main comments. We will improve the paper based on your inputs. We are looking forward to your new feedback and evaluation/score, and we remain at your disposal through the rebuttal process in case you have any other questions.
>
> [1] Ruszczyński, A. Risk-averse dynamic programming for Markov decision processes. Mathematical programming, 125(2):235–261, 2010.

---

> > ### Author Rebuttal · Reviewer_gTcm · 2026-04-03
> >
> > Thank you for the detailed response. I do not have any further question.

---

### Decision · Program_Chairs · 2026-04-30

**Decision:**

Accept (spotlight)

**Comment:**

The paper introduces ambiguity-averse Markov decision processes (MDPs) for modeling MDPs with uncertain transition probabilities, thereby adding epistemic uncertainty to the existing aleatoric uncertainty when analyzing random returns of policies. This formalism, combined with different risk measures over the returns, unifies several prior models of MDPs with epistemic uncertainty. Studying the fundamental concepts of value functions and Bellman operators in this broader formalism, the paper establishes conditions for the risk measures that lead to dynamic programming-based solutions.

All the reviewers support the acceptance of the paper. They particularly appreciate its building of an elegant, unified framework, encapsulating several existing views of uncertain MDPs, and its establishment of a theoretical understanding for the computation and optimization of different risk measures for uncertain MDPs. The reviewers believe that providing more intuition to the structure and steps of the proofs, as well as expanding on the practical implications of the positive and negative results, would be helpful. The authors are encouraged to follow up on this feedback, as they suggested in their response, to the extent possible. The authors' response sufficiently addressed the technical questions of the reviewers. Overall, the clean, abstract, and principled perspective of the paper in studying uncertain MDPs would be valuable to the community.